# Emergence of oncofetal plasticity is ubiquitous in early colorectal cancers

Julian R. Buissant des Amorie[1,2,8,9], Joris H. Hageman[1,2,8,9], Sascha R. Brunner[1,2,8,9], Suzanne E. M. van der Horst[1,2,8,9], Maria C. Puschhof[1,2], Arne van Hoeck[1,2], Inge van Lierop[1,2,8], Sjors Middelkamp[1,2,8], Lisa van der Schee[3], Sven van Kempen[3], Folkert Morsink[3], Robin Geene[1,4], Sander Mertens[1,2,8], David S. Cavigelli[1,2,8], Ingrid Verlaan-Klink[1,2,8], Lianne J. Kraaier[5], Jorieke Salij[6], Renate Bezemer[6], Onno Kranenburg[6], Miangela M. Laclé[3], Leon M. G. Moons[7] & Hugo J. G. Snippert[1,2,8✉]

Metastasis formation is classically considered a late-stage event in colorectal cancer evolution. Yet the time and spatial patterning by which metastatic competence is acquired remain poorly understood[1,2]. Here we show that metastasis-associated oncofetal cell states already emerge at the earliest stages of colorectal cancer, concurrent with invasive front formation. However, although necessary for metastasis, we detect them ubiquitously among early non-metastatic cancers, highlighting extra bottlenecks such as immune evasion. To understand how oncofetal cells first emerge, we generated multiregional organoid models that reflect successive tumour progression stages within individual early-stage colorectal cancers. Whole-genome sequencing and growth factor-dependency assays exclude tumour cell-intrinsic acquired traits. By contrast, single-cell spatial atlases of the tumour microenvironment before and after malignant transformation revealed stereotypic patterning of fibroblast subtypes resembling normal tissue architecture, resulting in distinct regional microenvironments. At the onset of malignant growth into the submucosa, the first cancer-associated fibroblasts to appear strongly resemble submucosal trophocytes and colocalize with oncofetal cell states at invasive fronts. Functionally, fibroblast–organoid cocultures confirm that these trophocyte-like cancer-associated fibroblasts induce plastic transitioning to oncofetal states. Thus, interactions between tumour and submucosal fibroblasts directly following malignant transformation dictate the timing and location at which oncofetal plasticity first occurs during colorectal cancer progression.

Colorectal cancer (CRC) serves as a prototype cancer to study tumour progression, with well-characterized signalling pathways affected by driver mutations. These drivers are typically acquired early during tumour progression[3–6], whereas genetic drivers of metastasis formation have not been identified[7]. In fact, invasive phenotypes are polyclonal[8] and further subclonal mutations seem largely unrelated to tumour cell phenotypes[9]. Moreover, from a clinical perspective, a substantial fraction of the genes that predict high risk of CRC relapse are expressed by cells of the tumour microenvironment (TME)[10–12]. These notions indicate a dominant role for the TME in the acquisition of metastatic capacity[13].

At the same time, the induction of regenerative fetal-like states within cancer cells is essential for successful metastatic seeding[14,15]. Although the molecular mechanisms underlying this oncofetal plasticity are under intensive investigation, it remains poorly understood when,

where and how these phenotypes first arise during tumour progression[1,2]. Indeed, metastases are generally associated with late-stage cancers, yet metastatic capacity can be acquired early. Notably, around 10% of CRCs are already metastatic at an early stage directly following the formation of invasive fronts (T1 stage)[16,17]. Moreover, evolutionary reconstructions of late-stage metastatic CRC indicated that metastatic seeding had typically started early, years before diagnosis[6].

Unfortunately, evolving cellular behaviour and phenotypes during early tumour stages are challenging to study, particularly in humans[18]. Traditionally, there has been a strong sampling bias towards late-stage cancers, as early stages often go unnoticed. Moreover, longitudinal sampling at sequential stages of tumour progression to extract a temporal account of key events is practically infeasible for most cancer types. Last, unlike adenoma initiation and late-stage cancer growth with metastatic spread, there is a lack of experimental models to study

[1]Center for Molecular Medicine, University Medical Center Utrecht, Utrecht, the Netherlands. [2]Oncode Institute, Utrecht, the Netherlands. [3]Department of Pathology, University Medical Center Utrecht, Utrecht, the Netherlands. [4]Utrecht Sequencing Facility, University Medical Center Utrecht, Utrecht, the Netherlands. [5]Independent Researcher, Utrecht, the Netherlands. [6]Utrecht Platform for Organoid Technology, Utrecht, the Netherlands. [7]Department of Gastroenterology and Hepatology, University Medical Center Utrecht, Utrecht, the Netherlands. [8]Present address: Princess Máxima Center for Pediatric Oncology, Utrecht, the Netherlands. [9]These authors contributed equally: Julian R. Buissant des Amorie, Joris H. Hageman, Sascha R. Brunner, Suzanne E. M. van der Horst. ✉e-mail: h.j.g.snippert-2@prinsesmaximacentrum.nl

the interactions between tumour cells and stroma during the transition from precancer to cancer following malignant transformation. As a consequence, the time, patterning and mechanisms by which metastatic competence[14,19,20] first emerges during the evolutionary timeline of CRCs have yet to be elucidated.

To investigate the initiation of metastatic competence in CRC, we here characterize tissue architecture and spatial cellular heterogeneity in early-stage CRCs, and complement these datasets with experimental, multiregional organoid models to disentangle tumour cell-intrinsic and extrinsic factors inducing metastasis-associated phenotypes.

## Mapping phenotypes in early colon cancer

To characterize diverging phenotypes succeeding malignant transformation, we performed spatial transcriptomics (Nanostring GeoMx) on early-stage CRCs. For this, we selected 19 stage T1 tumours, as these represent the earliest stage of invasion, with invasive tumour cells that have penetrated through the muscularis mucosae, but are still confined to the submucosa (Fig. 1a and Extended Data Fig. 1a, see 'Data availability' for an interactive dashboard). To generate a comprehensive dataset encompassing complete tumour architecture, we placed regions of interest (ROIs) along the sequential stages of tumour progression, that is, tumour-adjacent normal tissue, adenomatous tumour component, tumour core and invasive front (Fig. 1b and Extended Data Fig. 1b). Next, we obtained epithelium-specific and microenvironment-specific expression profiles per ROI (Fig. 1c), assessed with either a cancer-specific transcriptome atlas (CTA, ten patients) with high sensitivity for individual genes or a whole-transcriptome atlas (WTA, nine patients) with high utility for gene set analyses.

Unsupervised clustering (Extended Data Fig. 1c,d) and variance partitioning (Fig. 1d) revealed the strongest separation between epithelial and stromal segments in both assays, with patient-specific variance most pronounced in the epithelial cancer segments (Fig. 1e and Extended Data Fig. 1e). This is probably a consequence of tumour-specific genetic backgrounds[21]. By contrast, the remodelling of the stroma is more uniform with smaller patient-specific effects during tumour progression (Fig. 1e and Extended Data Fig. 1c–e).

Molecular classification of CRC by consensus molecular subtypes (CMS) is frequently used to predict disease progression, with CMS4 tumours, characterized by high stromal content, having particularly poor prognosis[22]. We investigated variability between different histopathological regions with regards to CMS and confirmed that CMS can be region-specific within a single tumour[9,23] (Extended Data Fig. 1f,g). Notably, regions classified as CMS4 showed higher amounts of stromal nuclei, underlining that this bulk transcriptomic classification is predominantly determined by stromal content (Extended Data Fig. 1h). To circumvent this, we also predicted epithelial classifications (iCMS)[24]. As expected, this analysis showed strong concordance of iCMS between segments within individual T1 CRCs (Extended Data Fig. 1i).

Next, we performed differential expression analysis between the tumour cell segments (Pan-Cytokeratin, PanCK+) of the four histopathological regions using the sensitive CTA dataset (Fig. 1f). Tumour cells of the invasive front showed high expression of markers associated with regenerative and fetal cell states, such as *MMP7*, *LAMC2* and *ANXA1* (refs. 14,25,26). The tumour core, however, showed expression of cell cycle-associated genes such as *MYC* and *PCNA*. To examine transcriptional programs in tumour cells, we extracted epithelial gene signatures of the core and invasive front (earlyCRCcore and earlyCRCinv, Supplementary Table 2) from the WTA dataset. Tumour core-specific expression patterns correlated with classical intestinal (cancer) stem cell[26–29] and proliferative[29–31] signatures. By contrast, we observed strong overlap between the invasive front signature and regenerative[14,25,26,31–33] and fetal-like[34] signatures, which are associated with plastic cell states

crucial for drug resistance and metastatic capacity[14,15,20,31,35,36] (Fig. 1g and Extended Data Fig. 1j).

Among these metastasis-associated signatures were the recently documented High Relapse Cell (HRC) signatures (coreHRC and epithelial High Relapse (epiHR))[14]. HRCs are a dynamic cell state originally detected in late-stage CRCs. They are enriched in invasive fronts and are the source of metastatic relapse in mice, whereas their abundance predicts the risk of relapse in patients. Detailed examination of our spatial datasets reveals HRC signature expression in invasive fronts of most T1 CRCs, despite half of them being non-metastatic (Fig. 1h). Confinement of HRC phenotypes to the invasive front of early-stage CRC was confirmed with immunofluorescence against laminin subunit gamma 2 (LAMC2) (Fig. 1i and Extended Data Fig. 1k), one of the most representative HRC markers[14] (Extended Data Fig. 2a–f). To substantiate the surprising finding that metastasis-associated transcriptional programs arise early during cancer progression, we validated their near-uniform presence in the invasive fronts of 232 T1 CRC specimen collected on tissue microarrays[37] (Fig. 1j,k). Moreover, 5-year clinical follow-up information from these tumours corroborated our finding that early expression of HRC signatures does not automatically equate to metastatic success (Fig. 1k). To understand this apparent mismatch, we analysed the HRC expression program in published single-cell RNA sequencing (scRNA-seq) datasets and observed that this program remains largely stable in oncofetal cells during disease progression (Extended Data Fig. 2g). Instead, the proportion of oncofetal cells increases in advancing tumours, probably as a consequence of expanding tumour mass and invasive front size (Extended Data Fig. 2g,h). Next, we re-analysed our spatial datasets of early-stage CRCs, which were selected to include non-metastatic (T1N0M0) and metastatic T1 CRCs (T1N1M0 and T1N0M1) (Fig. 1a). Investigating differences between these two groups revealed downregulation of immune-related expression programs in metastatic invasive fronts (Fig. 1l and Extended Data Fig. 1l), hinting at acquired immune evasion, a known hallmark of metastatic disease[33,38–40]. In conclusion, whereas metastatic burden is mostly a phenomenon of late-stage cancers, the emergence of metastasis-associated oncofetal phenotypes is not. Yet, metastatic spread among early-stage CRCs remains infrequent, highlighting further bottlenecks such as co-acquisition of immune-evasive properties.

## Non-genetic origin of invasive phenotypes

Investigating local phenotypic differences in early-stage CRC, we generated a multiregional organoid biobank (Fig. 2a and Supplementary Fig. 1). Using punch biopsies in surgically resected early-stage CRCs, a total of 73 organoid lines were derived from 16 patients with paired samples from normal, adenoma, tumour core and invasive front regions. Accuracy of regional biopsy identity was confirmed by histopathological examination of the remaining formalin-fixed paraffin-embedded (FFPE)-processed tumour (Fig. 2a,b). Success rate of organoid derivation from identity-confirmed biopsies was greater than 90%, with inaccessibility or absence of tumour regions being the main contributor for incomplete sampling (Fig. 2b). The biobank consists of adenocarcinomas, including two mucinous and one medullary carcinoma, the last of which has not been included in any CRC organoid biobank to our knowledge.

Whole-genome sequencing (WGS) of early passage organoid cultures from eight patients revealed that somatic mutations in canonical CRC drivers were already present at this early stage, including *APC*, *TP53*, *KRAS* and heterozygous loss of *SMAD4*, consistent with earlier observations[4,5,41] (Fig. 2c and Supplementary Table 4). Notably, in all patients the invasive fronts did not harbour any extra driver mutations compared with their corresponding tumour core (Fig. 2c,d and Extended Data Fig. 3a). Likewise, the karyotypes of early-stage CRC organoids showed typical patterns of chromosomal gains and losses known for CRCs[42] and were similar between paired organoids from the tumour core and the

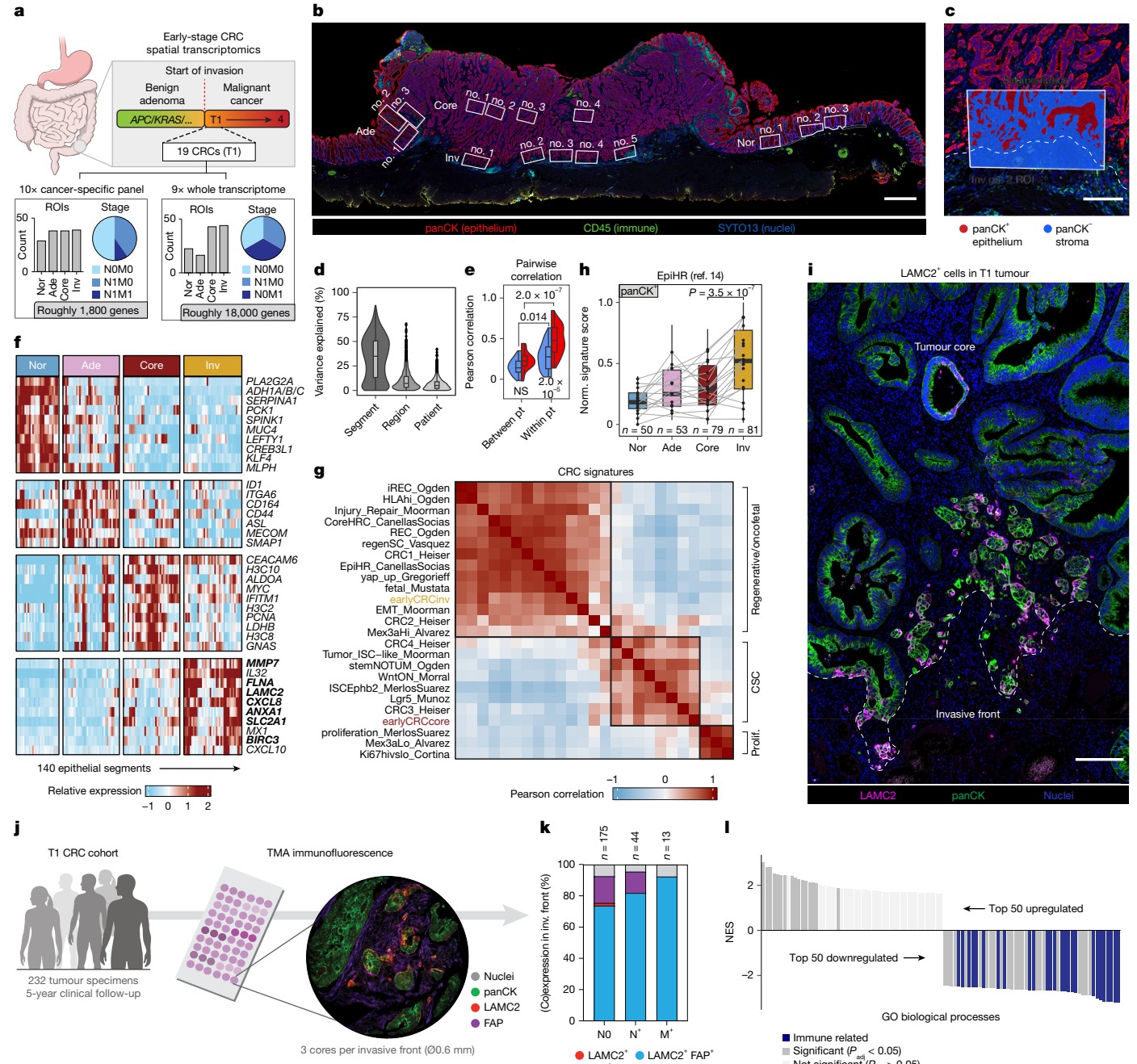

**Fig. 1 | Metastasis-associated signatures at the invasive front of early-stage colon cancer. a**, Spatial transcriptomics on 19 T1 CRCs with cancer-specific (CTA; 373 segments; 5 probes per gene) and whole-transcriptome (WTA; 281 segments; 1 probe per gene) probe panels. Lymph node (N) and distant metastasis status (M) are indicated. **b**, Example of ROI placement in T1 CRCs. **c**, Zoom-in of a representative ROI (invasive front no. 2 shown in **b**), showing epithelial and stromal segmentation for separate profiling. The dashed line shows the tumour border. **d**, Variance partitioning of a spatial transcriptomics dataset (WTA; $n = 2,000$ most variable genes; boxes, interquartile range; black bars, median; whiskers, 1.5× interquartile range). **e**, Violin plot of pairwise correlations between stromal–stromal (blue) or epithelial–epithelial (red) segment pairs within, or between, patients ($n = 104$ region comparisons across 9 patients; boxes, interquartile range; black bars, median; whiskers, 1.5× interquartile range; ANOVA $P = 2.31 \times 10^{-11}$, Tukey's honestly significant difference (HSD)). **f**, Heatmap showing relative expression ($\log_2$ fold change) of the top differentially expressed genes (lowest 10 $P_{adj}$ values with $P_{adj} < 0.05$; Wilcoxon rank-sum test with Bonferroni correction) in epithelial segments ($n = 140$) for each histopathological region (CTA). Bold font shows fetal markers. **g**, Correlation matrix of T1 tumour core (red font) and invasive front

(yellow font) signatures from this study versus published CRC signatures within tumour core ($n = 42$) and invasive front ($n = 43$) epithelial segments (WTA). **h**, EpiHR signature in epithelial segments of indicated histopathological regions of all 19 spatially profiled T1 CRCs. Connected points denote patient median. EpiHR signature was subset for genes probed in both the CTA and WTA probe panels (boxes, interquartile range; black bars, median; whiskers, 1.5× interquartile range; t-test). **i**, Immunofluorescence showing oncofetal marker LAMC2 at the invasive front of T1 CRC ($n = 2$ tumours). **j**, Immunofluorescence for LAMC2⁺ oncofetal tumour cells and FAP⁺ CAFs on tissue microarray of 232 T1 CRCs with 5-year clinical follow-up. **k**, Composition analysis of **j**. **l**, GSEA of epithelial invasive front segments from metastatic versus non-metastatic tumours. Significant immune-related gene ontology biological processes are highlighted in blue (permutation-based test, Benjamini–Hochberg corrected). Ade, adenoma; core, tumour core; CSC, cancer stem cell; GO, gene ontology; inv, invasive front; NES, normalized enrichment score; nor, normal; NS, not significant; pt, patient; TMA, tissue microarray. Scale bars, 1 mm (**b**); 200 μm (**c**,**i**). Illustration in **a** reproduced from NIH BioArt (https://bioart.niaid.nih. gov/bioart/212); illustrations in **j** adapted from NIH BioArt (https://bioart. niaid.nih.gov/bioart/214 and https://bioart.niaid.nih.gov/bioart/232).

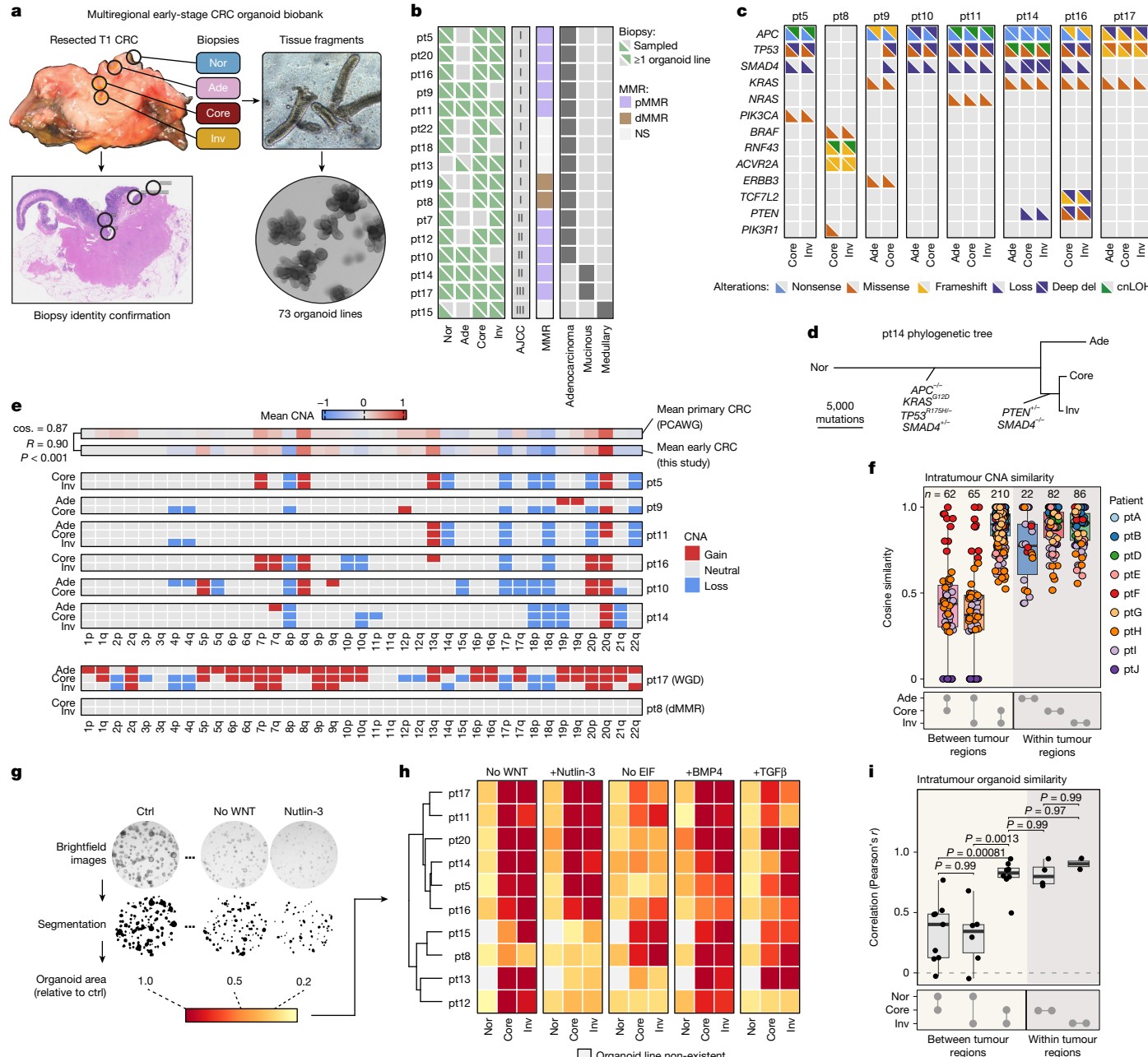

**Fig. 2 | Invasive front tumour cell phenotypes are not genetically driven.**
**a**, Overview of organoid derivation from regional punch biopsies (circles) in early-stage CRC. Biopsies are processed to tissue fragments and aliquoted for organoid culture and cryopreservation. Regional identity of biopsies was histologically confirmed on the sampled CRC (FFPE) by a pathologist (circles denote punch locations on haematoxylin and eosin). **b**, Overview of multiregional early-stage CRC organoid biobank. **c**, Driver landscape in early-stage CRC organoids. Driver genes with a prevalence of more than 4% in ref. 49 and a driver likelihood score of more than 0.8 are shown. **d**, Representative phylogenetic tree (patient 14) showing an evolutionary relationship between histopathological regions based on WGS data. Mutations and stage of acquisition are annotated. **e**, CNAs in early-stage CRC organoids. Top, mean CNA profile of our samples and PCAWG primary CRC reference dataset (dMMR and WGD excluded, statistical comparison is Pearson's correlation *r* with *P* value and cosine similarity). **f**, Boxplot showing pairwise cosine similarities of

inferred CNA profiles (inferCNV) within spatial transcriptomics dataset of early-stage CRCs. Points represent pairwise comparisons between regions (boxes, interquartile range; black bars, median; whiskers, 1.5× interquartile range). **g**, Quantification method for organoid outgrowth efficiency in various culture conditions using OrganoSeg[64]. Total organoid area relative to control medium after 9 days of outgrowth from single cells was assessed. **h**, Outgrowth efficiency of organoids from the early-stage CRC biobank in medium with indicated modifications. **i**, Pearson correlations of outgrowth efficiencies between organoid lines derived from the same tumour (*n* = 29 comparisons across regional organoid lines from 13 patients; boxes, interquartile range; black bars, median; whiskers, 1.5× interquartile range; ANOVA *P* < 0.0001, Tukey's HSD). AJCC, American Joint Committee on Cancer staging system; cnLOH, copy neutral loss of heterozygosity; ctrl, control; del, deletion; dMMR, deficient mismatch repair; ND, not determined; pMMR, proficient mismatch repair; WGD, whole-genome duplication.

invasive front (Fig. 2e). To corroborate these results, we derived karyotypes from our spatial transcriptomics dataset of early-stage CRCs. Again, despite regional divergence in tumour cell phenotypes, tumour

cells from the core and invasive front regions showed high similarity in their chromosome arm aneuploidy levels, whereas variation within the adenoma regions was larger[43] (Fig. 2f and Extended Data Fig. 3b,c).

To perform functional interrogation of regional acquired independence of external growth factors, we tested organoid sensitivity towards perturbations in WNT, P53, epidermal growth factor (EGF), BMP and TGFβ signalling pathways (Fig. 2g and Extended Data Fig. 3d,e). We assessed the total organoid area on brightfield images as a measure for outgrowth efficiency and observed a strong correlation in growth behaviour between paired organoids from core and invasive front (Fig. 2h,i), matching their identical driver landscape. Together, these experimental insights exclude tumour cell-intrinsic traits as drivers of phenotypic heterogeneity between tumour cores and invasive fronts of early-stage CRCs.

## Spatial fibroblast patterning in CRC

Next, we turned our attention to the TME as potential instigator of phenotypic plasticity in early CRC. Using scRNA-seq on aliquoted material from the same multiregional punch biopsies as used for organoid derivation, we characterized cell type composition of the diverse microenvironments in early-stage CRC (Fig. 3a,b, Extended Data Fig. 4a–d and Supplementary Fig. 2, see 'Data availability' for an interactive dashboard). We noticed extreme differences in spatial localization within the fibroblast compartment. Previously described stromal 1 fibroblasts[44,45] were restricted to normal tissue, whereas cancer-associated fibroblasts (CAFs) expressing *FAP* were exclusively found in cancer (Extended Data Fig. 4e,f). Furthermore, within cancer tissues we detected a fibroblast subtype (*SFRP2*[+], *GREM1*[+], *RSPO3*[+]) that resides exclusively at the invasive front and corresponds to so-called trophocytes (or S3 fibroblasts), which are known to localize around normal crypt bottoms where they support stem cell function[44,46] (Fig. 3b,c). Vice versa, telocytes (or S2 fibroblasts), a fibroblast subtype (*SOX6*[+], *BMP4*[+] and *WNT5A*[+]) residing towards crypt tops in normal colon tissue, were most abundant in the tumour core[44].

To relate transcriptional states in the TME to metastatic disease, we used the previously described TME-high relapse (TME-HR) signature and computed expression scores for each stromal cell type[14] (Fig. 3d). The TME-HR signature represents the stromal counterpart of the epiHR and predicts relapse. As expected, adaptive immune cell populations, such as CD8[+] T cells, presented with low TME-HR scores, suggesting inverse association with metastatic disease[14,47,48]. Moreover, deconvolution of our pseudo-bulk spatial transcriptomic datasets (Fig. 1) with the scRNA-seq-derived cell type signatures (Extended Data Fig. 4g), revealed decreased CD8[+] T cell signature expression in the invasive fronts of metastatic T1 CRCs (Fig. 3e). This is in line with the aforementioned downregulation of immune-related processes in metastatic tumour cells (Fig. 1l).

Next, we focused our attention on TME cell types that were positively associated with metastatic disease and could be involved in the induction of oncofetal phenotypes in early-stage CRC. The highest TME-HR scores were found in the CAFs and trophocytes of the invasive front. Further investigating the clinical relevance of these fibroblast subpopulations, we evaluated their prognostic value in a large cohort of CRCs[49]. Relative enrichment of both the invasive front CAF and trophocyte signatures coincided with shorter disease-free survival, even in CMS4 tumours (*n* = 239), which are known for high stromal content, aggressive nature and worst prognosis of the four molecular CRC subtypes (Fig. 3f and Extended Data Fig. 4h).

Investigating the origin and nature of the CAFs in early-stage CRC, we assembled a scRNA-seq cell–cell correlation matrix containing all fibroblasts from our cancer biopsies (Fig. 3g). In addition to fibroblast clusters resembling trophocytes and telocytes, this revealed three subclusters within the overall CAF population. Whereas one of the CAF subclusters showed clear enrichment of the telocyte signature (telocyte-like CAFs), the other two CAF subclusters were more similar to trophocytes, leading us to collectively refer to them as trophocyte-like CAFs (Fig. 3g and Extended Data Fig. 4i,j). Both trophocyte-like CAF

subclusters expressed the trophocyte markers *SFRP2* and *GREM1* in addition to fibroblast activation protein (FAP), but showed differential expression of the extracellular matrix remodellers *MMP1* and *MMP3* (Fig. 3h).

To assess the spatial patterning of the five fibroblast subclusters, we extracted their expression signatures (Supplementary Table 5) and determined fibroblast subtype enrichment in our spatial transcriptomic dataset of T1 CRCs (Fig. 1). Like trophocyte enrichment in invasive front biopsies, the trophocyte-like CAF signature was significantly enriched at the invasive front in the spatial transcriptomics dataset (Fig. 3i and Extended Data Fig. 5a). Moreover, immunofluorescence against SFRP2, a marker shared by trophocytes and trophocyte-like CAF populations, confirmed enrichment at the invasive front (Fig. 3j and Extended Data Fig. 5b). LAMC2[+] oncofetal cells colocalized with FAP[+] CAFs (Fig. 3k), supported by local co-enrichment of oncofetal and trophocyte-like CAF signatures in invasive front regions of the spatially examined T1 CRCs (Extended Data Fig. 5c).

Further increasing spatial resolution, we performed single-cell spatial transcriptomics (Nanostring CosMx) on a T1 CRC from our regional organoid biobank (pt5). Using our early-stage CRC scRNA-seq dataset as a reference, we could reliably assign cell type identities to roughly 630,000 single cells (Extended Data Fig. 6a,b, see 'Data availability' for an interactive dashboard). The ensuing single-cell spatial map corroborated the presence of oncofetal cells (HRC signature, roughly 5% of all tumour cells) at the invasive front (Fig. 3l and Extended Data Fig. 6b–e). Moreover, we again observed the regional patterning of the fibroblast subtypes, with telocytes restricted to the tumour core and trophocytes populating the submucosa beneath the invasive front (Fig. 3m and Extended Data Fig. 6f), reminiscent of normal tissue architecture (Extended Data Fig. 6a). Of interest, the trophocyte-like CAFs (MMP[−]) were highly abundant at a confined narrow belt at the stromal side of the invasive front interface, whereas the trophocyte-like CAF population that resided at the tumour side expressed MMPs (MMP[+]) and colocalized strongest with oncofetal cells (Fig. 3n and Extended Data Fig. 6g).

Taken together, the spatial patterning of fibroblast subtypes along the radial tumour axis in early-stage CRCs bears resemblance to fibroblast subtype zonation along the crypt axes in the normal gut. In particular, trophocytes and trophocyte-like CAFs are enriched at the invasive front of early-stage CRCs and are associated with oncofetal tumour cell states, high TME-HR signatures and poor patient outcome.

## Trophocyte-like CAFs induce plasticity

To functionally test whether CAFs from early-stage CRC can induce oncofetal tumour cell states we performed bulk RNA-seq on cocultures of our T1 organoids and fibroblast lines derived in parallel with the organoids (Fig. 4a and Extended Data Fig. 7a,b). Fibroblasts were cultured both on plastic (two-dimensional, 2D) and in Matrigel (three-dimensional, 3D), as the latter method can force invasion-associated phenotypes[50]. Differential expression analysis showed clear induction of oncofetal programs in cocultured tumour organoids, reminiscent of the invasive fronts in early-stage CRCs (Fig. 4b,c and Extended Data Fig. 7c,d). The strongest induction was observed in 3D coculture, where the fibroblasts phenocopied stromal expression patterns of invasive fronts the most, including the inflammatory CAF signature (Fig. 4d and Extended Data Fig. 7e). Thus, while reducing the complexity of all cellular interactions within the TME at the invasive front, our in vitro organoid–fibroblast cocultures recapitulate paracrine crosstalk between tumour cells and fibroblasts at the invasive front.

To identify ligands that induce oncofetal plasticity, we leveraged the fact that the 3D fibroblast culture conditions yielded stronger induction of the oncofetal program than the 2D condition. Three-dimensional fibroblasts showed elevated expression of *TGFB3* (Fig. 4e), which is also upregulated in trophocytes and trophocyte-like CAFs of the invasive

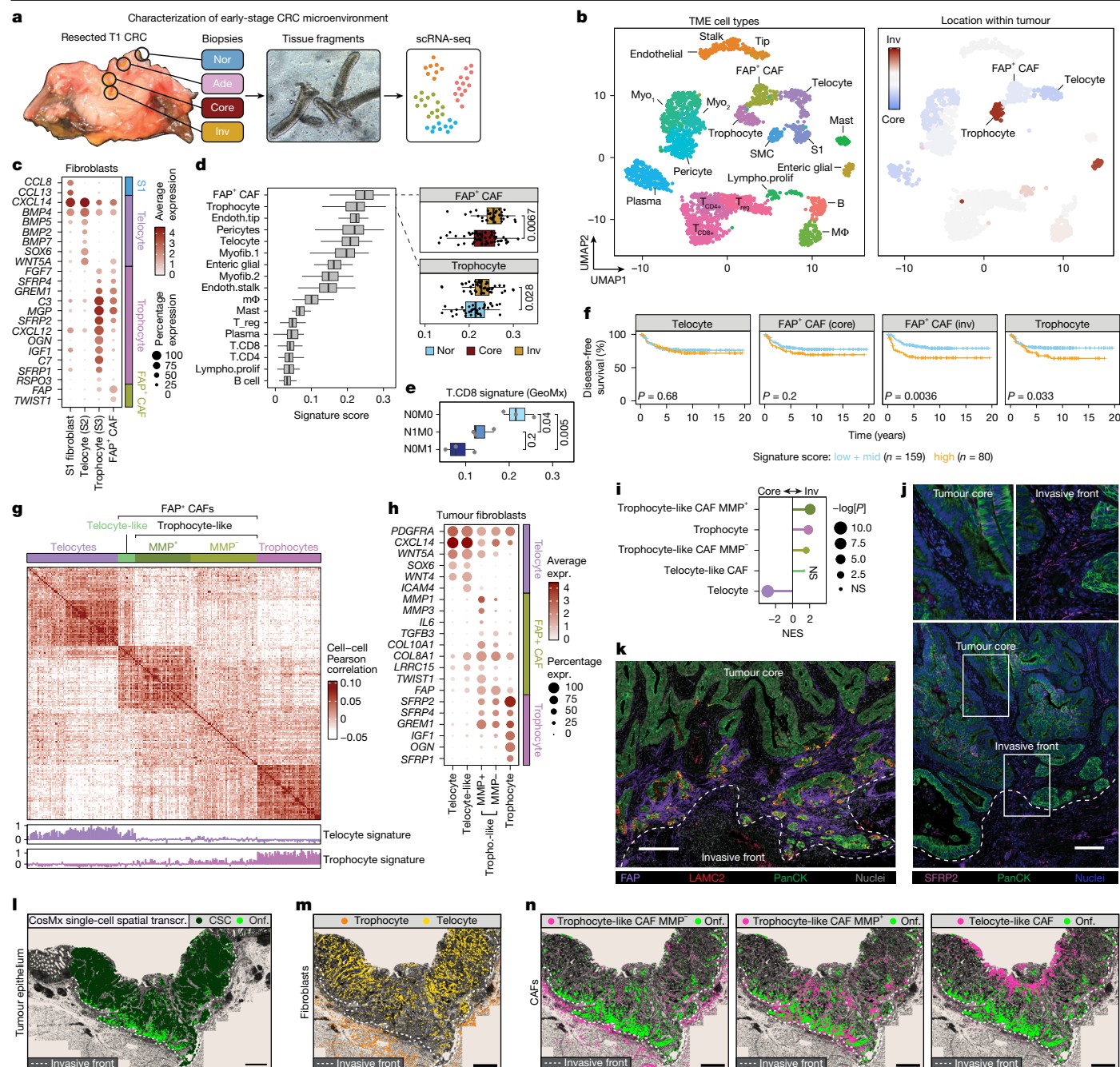

**Fig. 3 | See next page for caption.**

front in vivo (Extended Data Fig. 8a,b) and, together with *TGFB1*, predicted to be a potent regulator of oncofetal programs such as HRC in a cell–cell communication analysis using ligand–target inference (NicheNet)[26] (Extended Data Fig. 8c). In addition to TGFβ expression, 3D fibroblasts phenocopied in vivo trophocyte-like CAFs by expressing prostaglandin synthase 2 (*PTGS2*, also known as *COX2*) (Fig. 4e and Extended Data Fig. 8b), which is a rate-limiting enzyme for prostaglandin production[51]. Prostaglandins, absent in NicheNet, have been implicated in induction of regenerative and fetal states in epithelial cells[52,53] and thus also represent a potential regulator of oncofetal plasticity at the invasive front.

To experimentally test candidate stimuli of oncofetal plasticity (Extended Data Fig. 8c), we edited invasive front organoids of early-stage CRC with a CRISPR-mediated fluorescent knock-in reporter at the *EMP1* locus (Fig. 4f,g, Extended Data Fig. 8d and Supplementary Fig. 3),

the marker gene that was used to validate the essential role of HRCs in metastatic relapse[14]. Among the stromal cues tested, treatment with TGFβ and prostaglandin induced the strongest increase in the fraction of EMP1+ cells (Fig. 4h), yielding the highest and most consistent results when the two were combined (Fig. 4i and Extended Data Fig. 8e,f). By contrast, telocyte-secreted factors (BMP2, BMP4, WNT5a and CXCL14), inflammatory cytokines (tumour necrosis factor, interleukin-1b (IL-1b), IL-27, IL-36a, IL-36b) (Fig. 4h) or exposure to collagen (Extended Data Fig. 8g) did not induce EMP1+ cells. We validated oncofetal cell state induction by TGFβ and prostaglandins with RNA-seq (Fig. 4j,k) and immunofluorescence against LAMC2 (Extended Data Fig. 8h). In agreement with our earlier findings, we observed that the oncofetal program could be induced in both the invasive front and tumour core organoids (Fig. 4l and Extended Data Fig. 2e), underscoring that oncofetal plasticity is extrinsically induced.

**Fig. 3 | Trophocyte-like CAFs at the invasive front. a**, scRNA-seq-based cell type identification in early-stage CRC TME. **b**, UMAP of scRNA-seq data from regional biopsies of biobanked CRCs (pt5, pt11, pt13, pt14 and pt16 (Fig. 2b)). Left, cell type annotation. Right, relative contribution of core and invasive front biopsies per cluster. **c**, Marker gene expression level across fibroblast subtypes from early-stage CRCs and tumour-adjacent normal tissue. Colours, expression level; dot size, percentage of cells expressing transcript. **d**, TME-HR signature[14] scores across TME cell types in early-stage CRCs (n = 1,612 cells; 5 patients). Right panels, cells (black points) with highest TME-HR signature scores among CAFs and trophocytes originate from invasive front biopsies (boxes, interquartile range; black bars, median; whiskers, 1.5× interquartile range); t-test. **e**, CD8 T cell signature expression in tumours with indicated metastasis status from WTA spatial transcriptomics dataset of early-stage CRC (Fig. 1) (n = 9 patients; boxes, interquartile range; black bars, median; whiskers, 1.5× interquartile range; ANOVA P = 0.0015, Tukey's HSD test). **f**, Disease-free survival of patients with CMS4 (ref. 49) stratified by low or mid (bottom 66%) versus high (top 33%) expression of indicated fibroblast subtype signatures (top 100 region-specific markers, Supplementary Table 5) (log-rank test). **g**, scRNA-seq cell–cell correlation matrix of fibroblasts from core and invasive front biopsies. Three CAF subtypes (green shades) can be recognized within the FAP[+] CAF cluster. Colour scale represents cell–cell transcriptome similarity (Pearson correlation coefficient calculated over the 8,000 most variable genes). Bottom tracks, telocyte and trophocyte signature scores for each cell.

**h**, Expression level of markers among fibroblast subtypes. Colours, expression level; dot size, percentage of cells expressing transcripts. Benjamini–Hochberg. **i**, GSEA of GeoMx (WTA; n = 9 T1 CRCs; permutation test with FDR) bulk expression profiles between tumour core and invasive front using cell type-specific signatures from scRNA-seq. Lollypop length, normalized enrichment score; dot size, nominal probability (−log[P]). **j**, Immunofluorescence for trophocyte marker SFRP2 (magenta) at the invasive front of early-stage CRC (n = 4 tumours). Tumour cells in green (PanCK[+]), nuclei in blue (SYTO13). Top panels show zoom-ins of indicated regions. The dashed line shows the tumour border. **k**, Immunofluorescence against FAP (purple) and HRC marker LAMC2 (red) at the invasive front of early-stage CRC (n = 3 tumours). Tumour cells in green (PanCK), nuclei in grey (SYTO13). Costaining of PanCK and LAMC2 is shown in yellow. The white dashed line shows the tumour border. **l**, Single-cell spatial transcriptomics of T1 CRC specimen showing oncofetal cells (high expressors of High Relapse signature, bright green)[14] and WNT-driven cancer (stem) cells (dark green)[29] The dashed line shows the tumour border. **m**, As in **l**, but showing trophocytes (orange) and telocytes (yellow). The dashed line shows the tumour border. **n**, As in **l**, but showing FAP + CAF subtypes. B, B cell; IQR, interquartile range; lympho.prolif, proliferative lymphocytes; MΦ, macrophage; myo, myofibroblast; NES, normalized enrichment score; NS, not significant; onf, oncofetal; S1, stromal 1; SMC, smooth muscle cell; $T_{CD8+}$, CD8[+] T cell; $T_{CD4+}$, CD4[+] T cell; transcr., transcriptomics; $T_{reg}$, regulatory T cell. Scale bars, 200 μm (**j,k**); 1 mm (**l–n**).

Furthermore, major histocompatibility complex class I (MHCI) levels were not different between EMP1[+] and EMP1[−] cells (Extended Data Fig. 8i–k), supporting the notion that oncofetal plasticity seems uncoupled from the acquisition of immune-evasive properties.

Together, our in vitro experiments confirm that signalling gradients, established by the spatial patterning of fibroblast subtypes, contribute to the first induction and local confinement of oncofetal plasticity in invasive fronts at the earliest stages of CRC.

## Mapping the birth of CRC plasticity

To elucidate the timing and origin of the various CAF populations that we identified, we performed extra single-cell spatial transcriptomics on 11 early-stage CRC specimens that were captured at critical timepoints just before and after malignant transformation. Specifically, we analysed roughly 1.25 million cells (6,000 gene probe panel) at 3 pseudo-timed substages flanking the onset of malignancy, that is, before invasive front formation in the submucosa (intramucosal carcinoma, n = 3), immediately after (T1 sm1, n = 5) and once the invasive front is robustly established beyond the muscularis mucosae (T1 sm3, n = 3) (Fig. 5a,b and Extended Data Fig. 9a).

Detailed cell type assignment based on transcriptome (Fig. 5c,d), and confirmation with immunofluorescence (Fig. 5e,f and Extended Data Fig. 9b), indicated that these samples capture the precise moment at which LAMC2[+] oncofetal tumour cell states and FAP[+] CAFs emerge, with both cell types absent in intramucosal carcinomas and increasingly present in the invasive fronts from T1 sm1 to T1 sm3. Trophocyte-like CAFs were first found immediately after malignant transformation (T1 sm1), whereas the intramucosal carcinomas only contained tissue-resident trophocytes (Fig. 5c,d and Extended Data Fig. 9c). Within the trophocyte-like CAF population, the MMP[+] fibroblasts seemed to arise the latest, with high abundance in full-fledged fronts of T1 sm3 cancers (Fig. 5c,d and Extended Data Fig. 9c). Notably, both early CAF populations (MMP[−] and MMP[+]) were predominantly found in the spatial neighbourhood of oncofetal tumour cell states (Fig. 5g and Extended Data Fig. 9d), coinciding with local elevation of TGFβ and prostaglandin signalling (Fig. 5h and Extended Data Fig. 9e). Next, we used the many fibroblasts identified in these early CRC specimens (roughly 40,000) to study potential relationships and transitioning between the fibroblast subtypes. We observed a lot of cells constituting intermediates between

tissue-resident trophocytes and trophocyte-like CAFs (Extended Data Fig. 9f), suggestive of possible transitioning between these states.

The phenotypic resemblance of trophocyte-like CAFs to trophocytes, together with their emergence in the trophocyte-rich submucosa at the T1 sm1 stage, suggests that submucosal trophocytes are a major cellular source for CAFs and probably their first cells of origin. This model is further supported by trajectory inference analyses on published scRNA-seq datasets[45,54,55] of the CRC microenvironment (Fig. 5i,j and Extended Data Fig. 10a–d), which attribute the highest differentiation potential to tissue-resident trophocytes (Extended Data Fig. 10e) and confirm the differentiation trajectory from trophocytes to CAFs (Fig. 5i,j). Last, using this integrated scRNA-seq dataset, we observed the same diversity of fibroblasts subtypes in early-stage and late-stage CRC, while the relative proportions of fibroblast subtypes diverged (Fig. 5k and Extended Data Fig. 10f).

Overall, our data show that oncofetal tumour cell states emerge shortly after the dissolution of the muscularis mucosae, at the earliest stages of invasive front formation. Moreover, this early oncofetal plasticity coincides with and is driven by the transitioning of submucosal trophocytes towards CAF-like phenotypes (Extended Data Fig. 10g).

## Discussion

Although many resources are devoted to understanding and combating metastatic disease, it is striking how poorly we understand the timing and mechanisms by which metastatic competence first arises during tumour progression. Here we provide functional characterization of the earliest CRC stage in patients that is demarcated by invasive front formation following malignant transformation. In virtually all early-stage CRCs examined, we find clear evidence of cellular states that are classically associated with metastatic competence. Moreover, using multiregional paired organoid models we provide functional evidence that these de novo cell states in the invasive front are the result of oncofetal plasticity that is instigated by environmental factors.

The observation that early acquisition of oncofetal cell states is a commonality rather than an exception, paired with the fact that most early CRCs show no signs of metastatic seeding, indicates that oncofetal cells may be essential but not sufficient for metastatic success. Indeed, comparison of tumours with and without early metastatic

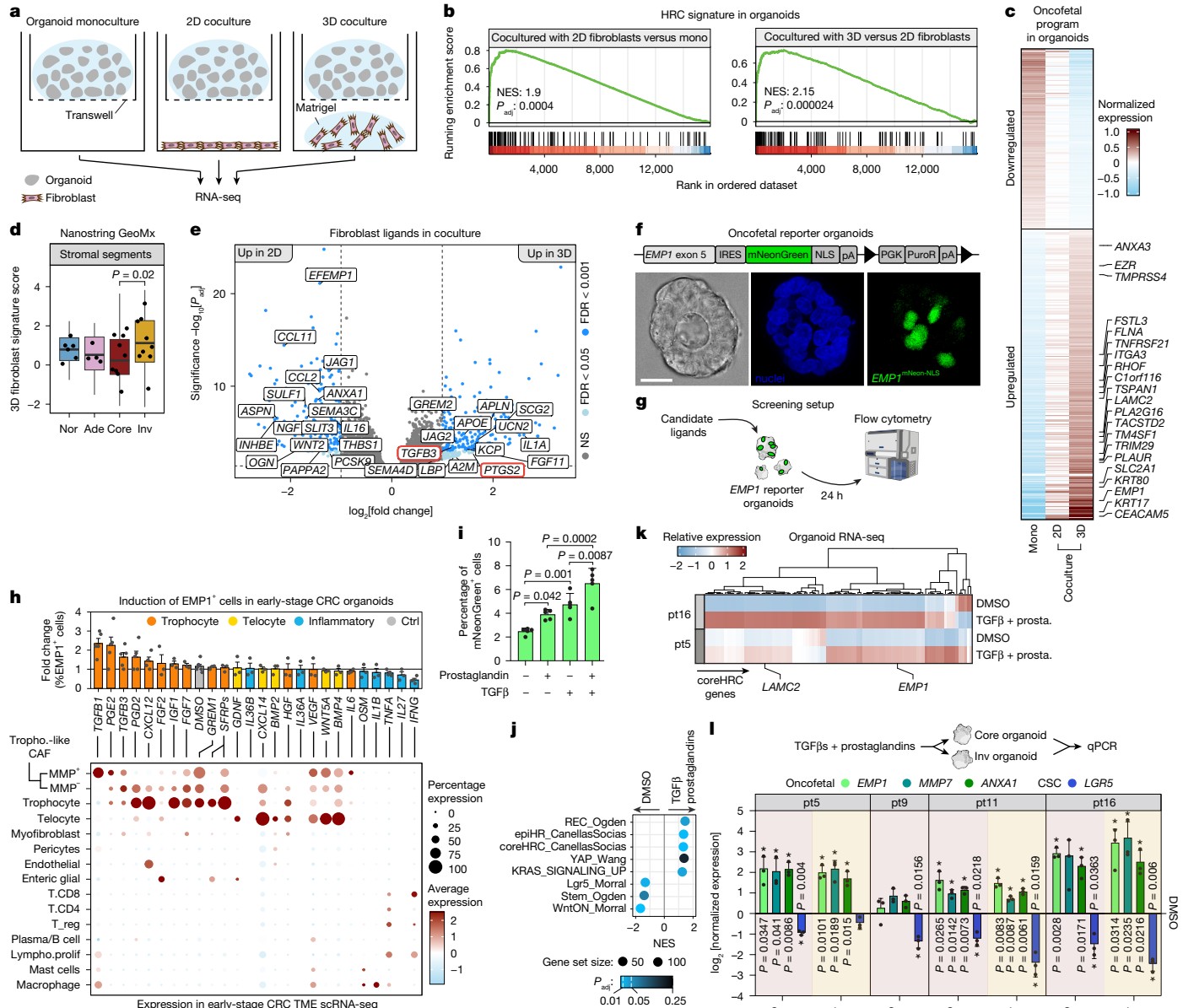

**Fig. 4 | Trophocyte-like CAFs induce oncofetal plasticity to EMP1+ tumour cell states. a**, Schematic of coculture experiment. **b**, GSEA of coreHRC signature in organoids across culture conditions (permutation test with FDR). Left, 2D (n = 6) versus monoculture (n = 7). Right, 3D (n = 6) versus 2D (n = 6). **c**, The log₂ fold change of differentially expressed genes in coculture. Oncofetal markers annotated. **d**, 3D fibroblasts signature (top 100 differentially expressed genes ranked by log₂ fold change, $P_{adj}$ < 0.01) expression in GeoMx regions (WTA; n = 9 patients). Boxes, interquartile range; black bars, median; whiskers, 1.5× interquartile range; t-test). **e**, Differentially expressed genes in 3D (n = 6) versus 2D (n = 6) cocultured fibroblasts. Ligand-mediated signalling genes are annotated (Wald test with Benjamini–Hochberg correction). **f**, Top, *EMP1* reporter knock-in schematic. Bottom, fluorescence image of *EMP1* reporter organoid. **g**, Organoid-based screen for inducers of oncofetal state. **h**, Top, fold change in EMP1-mNeon+ cells (%) versus control. Data points, independent experiments (exact n shown by number of data points; error bars, s.e.m.). Bottom, scRNA-seq dot plot of screened ligands in invasive front biopsies (Fig. 3)

(PGE2, PTGES; PGD2, PTGDS). **i**, Percentage of EMP1-mNeon+ cells across conditions. Prostaglandin, PGD2 + PGE2. TGFβ, TGFβ1 + TGFβ3 (n = 5; mean + s.d.; ANOVA, Bonferroni correction). **j**, GSEA of oncofetal and CSC signatures in organoids treated with TGFβ (TGFβ1 + TGFβ3) and prostaglandins (PGD2 + PGE2) (n = 2 patients; 3 replicates; permutation-based test, Benjamini–Hochberg corrected). **k**, Like **j**, but relative expression of coreHRC signature genes. **l**, qPCR of oncofetal markers and LGR5, following 24 h of treatment with PGD2, PGE2, TGFβ1 and TGFβ3 in 7 independent organoid lines from 4 early-stage CRCs. Values normalized to DMSO, mean + s.d. Points, independent measurements (n = 3; *P < 0.05, ratio paired t-test). Illustration in **g** adapted from NIH BioArt (https://bioart.niaid.nih.gov/bioart/160); illustration in **l** reproduced from NIH BioArt (https://bioart.niaid.nih.gov/bioart/661). DMSO, dimethylsulfoxide; IRES, internal ribosomal entry site; mono, monoculture; NLS, nuclear localization signal; pA, polyA; PuroR, Puromycin Resistance cassette. Scale bar, 20 μm (**f**).

seeding, hinted at co-acquisition of immune-evasive properties as an important further requirement[3,6]. The timing and mechanisms by which immune-evasive properties appear, and whether they continue to evolve while the cancer progresses to more advanced stages warrant further investigation. Likewise, whether the relative increase in oncofetal cells during disease progression is due to evolving TMEs or

cell-intrinsic acquired hypersensitivity for fetal-like reprogramming cues[35,39,56] needs to be explored.

Although previous research efforts underline the important role of CAFs in CRC[52,57–60], the time at which they emerge, their origin and how heterogeneity within the CAF population relates to function, have remained largely elusive. Our analysis of pseudo-timed substages

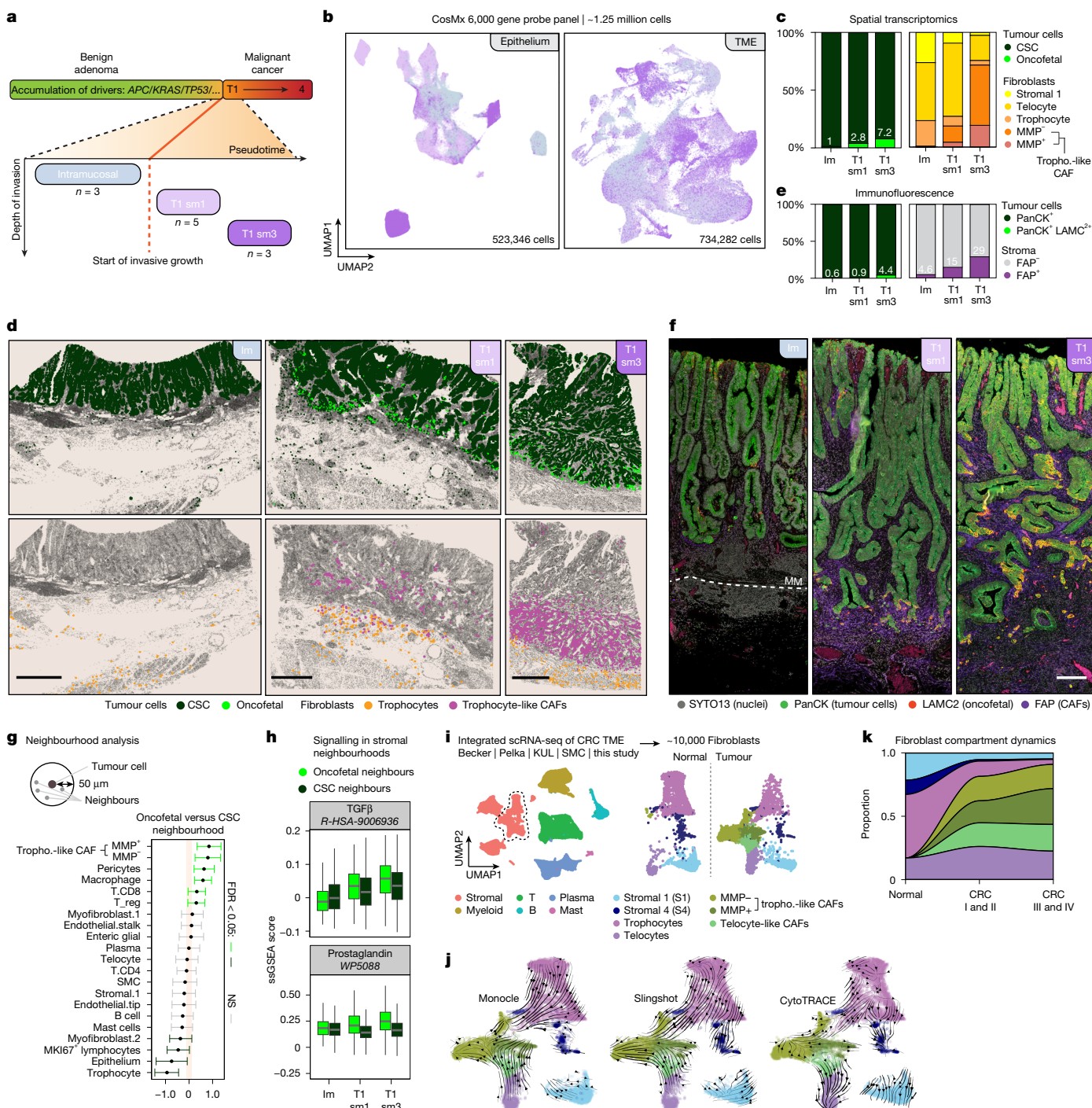

**Fig. 5 | Trophocyte-like CAFs and oncofetal plasticity co-emerge at the birth of malignancy. a**, Pseudo-longitudinal single-cell spatial transcriptomics of 11 early-stage CRCs (CosMx; 6,000 probe panel). **b**, UMAP of epithelial (left) and microenvironmental (right) compartments. **c**, Relative abundance (%) of tumour cell types and fibroblasts in single-cell spatial transcriptomics of intramucosal (n = 3), T1 sm1 (n = 5) and T1 sm3 (n = 3) specimens. **d**, Single-cell spatial plots showing localization of WNT-driven cancer (stem) cells (CSC, dark green), oncofetal cells (bright green, HRC signature), trophocytes (orange) and trophocyte-like CAFs (purple) in representative intramucosal, T1 sm1 and T1 sm3 specimens. **e**, Percentage of LAMC2+ oncofetal tumour cells (within PanCK+ epithelium) and FAP+ CAFs (within PanCK− stroma) just before (intramucosal carcinoma, n = 3) and shortly after (T1 sm1 n = 3 and T1 sm3 n = 3) malignant transformation. **f**, Immunofluorescence of tumour specimens just before and after malignant transformation. Nuclei (grey, SYTO13), epithelial cells (green, PanCK), oncofetal tumour cells (red, LAMC2) and CAFs (purple, FAP)

are stained. Orange, PanCK and LAMC2 co-expression. White dashed line, muscularis mucosae. **g**, Differential cell type composition of oncofetal and CSC neighbourhoods (50 µm radius; n = 11 patients). Whiskers, 95% credible interval, coloured by significance (FDR < 0.05). **h**, ssGSEA scores for TGFβ and prostaglandin signalling in stromal neighbourhoods of oncofetal and CSC tumour cells per tumour stage (n = 11 patients; boxes, interquartile range; grey bars, median; whiskers, 1.5× interquartile range). **i**, UMAP of integrated CRC TME scRNA-seq datasets[45,54,55] (n = 110 patients). Left, unsupervised clustering and cell type annotations. Right, zoom-in of fibroblasts, split over normal and tumour tissue. **j**, UMAP of fibroblasts from integrated scRNA-seq with streamlines showing inferred trajectories of Monocle, Slingshot and CytoTRACE pseudotime. **k**, Relative contribution of fibroblast subtypes to total fibroblast compartment in normal tissue, and early (stage I–II) and advanced (stage III–IV) CRCs. Im, intramucosal; MM, muscularis mucosae. Scale bars, 1 mm (**d**); 200 µm (**f**).

flanking malignant transformation indicates that trophocyte-like CAFs and oncofetal plasticity arise nearly simultaneously in space and time. Notably, the ability of trophocyte-like CAFs to induce these phenotypes, together with the observation that FAP[+] tumours without LAMC2 were more common than LAMC2[+] tumours without FAP (Fig. 1k $n = 35$ versus $n = 3$, respectively), suggest that trophocyte-like CAFs generally come first.

Heterogeneity and patterning of fibroblast subtypes orchestrate signalling gradients that regulate cell fate in the intestinal epithelium[44,46,61–63]. Similarly, we find a large diversity of fibroblast subtypes in early-stage CRCs, including tissue-resident fibroblasts such as trophocytes in the submucosa, and telocytes towards the luminal side of the cancer. Most striking is the diversity within the fibroblast population commonly referred to as FAP[+] CAFs. This CAF population shows subtypes that bear resemblance to the aforementioned trophocytes and telocytes. In addition to their phenotypic similarity, they show patterning within the tumour akin to the normal mucosa, with trophocyte-like CAFs concentrated at the tumour–stroma interface of the invasive front and telocyte-like CAFs residing in the tumour core towards the luminal side.

Transcriptomic relationships between fibroblast states and their successive appearance in the pseudo-timed spatial single-cell atlases capturing the onset of malignancy support a differentiation trajectory in which tissue-resident trophocytes transition towards trophocyte-like CAFs during the initial stages of invasive front formation. Whereas trophocytes seem the first to transition, our data indicate that telocytes may also be susceptible to transition to CAFs (telocyte-like CAFs). It is of high interest to study whether the same patterning of CAF subtypes observed in early-stage CRC remains intact in more advanced CRC stages and if the same signalling axes that govern fibroblast patterning in normal tissue, such as BMP signalling[63], are responsible for the patterning of fibroblast subtypes in cancer. Furthermore, it will be of importance to understand the signals that are involved in the first transformation of tissue-resident fibroblast populations towards CAF subtypes, as interference in this process may represent an indirect therapeutic strategy to affect tumour cell states and metastatic competence.

Practical limitations have hampered scientific progress in early-stage CRC. Organoid models, in conjunction with single-cell atlases describing in vivo cell states and architecture, provide functional resources to start understanding the first stages succeeding malignant transformation of precancer to cancer and the origin of metastatic disease.

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

## Methods

### Patients

This study was approved by the University Medical Centre (UMC) Utrecht ethical committee, carried out in accordance with the ethical guidelines and regulations and all patients provided written informed consent. FFPE specimens for immunohistochemistry and spatial transcriptomics were requested from and provided by the UMC Utrecht pathology department. Patient inclusion for the organoid biobank was managed by the Utrecht Platform for Organoid Technology (https://uport.umcutrecht.nl/researcher/en/). The biobank participants were 16 patients suspected of having early-stage CRC who underwent surgery for removal of the primary tumour, instead of endoscopic removal, owing to inaccessibility of the tumour. Clinical data from patients featured in this study can be found in Supplementary Table 1.

### GeoMx bulk spatial transcriptomics

Nanostring GeoMx experiments were conducted with the Utrecht Sequencing Facility (USEQ) and performed as previously described in ref. 65. In brief, 10 T1 CRCs (5× T1N0M0 and 5× T1N1M0) were analysed using the GeoMx CTA (Cancer Transcriptome Atlas) panel and 9 T1 CRCs (3× T1N0M0, 3× T1N1M0 and 3× T1N0M1) were analysed using the GeoMx WTA panel. The specimens analysed by CTA were selected such that risk factors, including lymphovascular invasion, tumour budding, location and morphology were similar between metastatic and non-metastatic primary tumours. Specimens were stained for PanCK (Novus Biologicals, NBP2-33200AF532, 2 µg ml⁻¹) to visualize epithelium, CD45 (Novus Biologicals, NBP2-34528AF594, 5 µg ml⁻¹) to visualize immune cells and SYTO13 (Invitrogen, S7575, 500 nM) to visualize nuclei. ROIs containing 100 to 1,000 nuclei were placed in 4 histopathological regions per tumour: normal tissue adjacent to the tumour, adenomatous tumour component, tumour core and invasive front. Invasive front ROIs were consistently placed, with epithelial tumour strands penetrating the supportive tissue for roughly three-quarters of the ROI edge perpendicular to the tumour border. After ROI placement, PanCK immunofluorescence was used to segment epithelial (PanCK⁺) and stromal (PanCK⁻) compartments for separate transcriptomic profiling. For the CTA cohort, CD45 negative and positive areas within the stromal compartment were profiled separately, but were summed during analysis for comparability with the WTA experiment. Standard quality control (unified quality control threshold) was applied to both experiments and can be viewed in Supplementary Reports 1 and 2. In total 426 (CTA) and 285 (WTA) ROIs were sampled across all specimens of which 373 and 281 ROIs were retained after quality control for the CTA and WTA experiments, respectively. At the gene level, 1,781 out of 1,812 and 18,441 out of 18,677 genes were retained after quality control for the CTA and WTA experiments, respectively. Probe counts were aggregated per gene target, Q3 normalized, batch corrected (with 'slide name' as the batch to be corrected for) and log₂ transformed. For all downstream analyses, sample pt17 (T1_NANO_013) was excluded, because it is classified as a T3 tumour. For variance partition analysis the VariancePartition[66] (v.1.38.1) R package was used. To compare different tissue regions within a specimen and across different specimens, we used a linear mixed model approach to model the normalized expression separately for epithelial and stromal segments: log₂(gene) ~ tissue region + (1 + tissue region | patient ID). For gene set enrichment analysis (GSEA), two methods were applied: preranked GSEA (fgsea[67] v.1.24.0) and single-sample GSEA (ssGSEA[68] implemented in GSVA v.1.46.0). Gene sets tested originated from MsigDB (https://www.gsea-msigdb.org/gsea/msigdb), from this study or from published literature (summarized in Supplementary Table 3).

### GeoMx CMS classification

Regions from the WTA cohort were used for CMS[19] and iCMS classification[20]. For CMS classification, raw transcript counts of adjacent PanCK⁺ and PanCK⁻ segments were summed per area of interest and thereafter summed by patient ID and tissue region. Patient F was excluded from this analysis, because the PanCK⁻ and PanCK⁺ segments were not located within the same areas of interest. These pseudo-bulk samples were used as input for CMScaller[19] (v.2.0.1), which was run with 'RNAseq = TRUE' alongside default parameters. Finally, the fraction of stromal nuclei for each area of interest was calculated. For iCMS classification CMScaller was run with raw PanCK⁺ gene counts only and 'RNAseq = TRUE'. CMS2 and iCMS3 Up gene sets[20] were used as templates to classify the segments.

### GeoMx CNA prediction

Copy number alteration (CNA) profiles of epithelial cells from the different histopathological regions were estimated using inferCNV (v.1.14.2; 'cutoff = 0.1'; using normal tissue as a reference group and excluding chromosome XY and mitochondrial genes). Chromosome arm gains and losses were defined as an average residual expression of more than 1.1 or less than 0.9 across all genes on that arm, respectively. Short arms of acrocentric (13p, 14p, 15p, 21p, 22p) and both arms of sex chromosomes were excluded. To calculate pairwise cosine similarities among ROIs from the same tumour, the average residual expression per chromosome arm was rounded to the nearest decimal.

### Immunohistochemistry of CRCs

Spatial transcriptomics findings were validated with immunohistochemistry labelling on consecutive slides of the selected T1 tumours. Here 5-µm thick FFPE-embedded tumour sections were mounted on glass slides and baked in at 60 °C for 1 h. Deparaffinization and rehydration was performed as follows: xylene (3 min, 1 change), 96% ethanol (3 min, 1 change), 70% ethanol (3 min, 1 change), rinse in deionized water and rinse in tap water. Heat-mediated antigen retrieval was performed for 20 min in 50 mM Tris/1 mM EDTA pH 9.4 buffer at 95 °C. The following primary antibodies were used: SFRP2 (PA5-29390, Invitrogen, 1:200), LAMC2 (AMAb91098, Atlas Antibodies, 1:500), PanCK (AlexaFluor 532 conjugated; NBP2-33200 Novus 1:500 and NBP3-08398 Novus 1:300) and DNA Syto 13 (S7575, Invitrogen, 1:10,000). The following secondary antibodies were used: Alexa 594 anti-rabbit (Invitrogen A11037; 2 µg ml⁻¹) and Alexa 594 anti-mouse (Invitrogen A11032; 2 µg ml⁻¹). Slides were scanned on the GeoMx Digital Spatial Profiler (Nanostring) with a ×20 0.45 numerical aperture objective and analysed using the QuPath (v.0.6.0) Instanseg extension[69]. In brief, we quantified all cells within the invasive front (1 mm deep, measured from tumour margin), irrespective of tumour width. Within invasive fronts, nuclei and epithelial cell bodies were segmented on the basis of Syto13 and PanCK pixel intensities, after which percentages of LAMC2⁺ cells (in epithelium) and the percentages of SFRP2⁺ and FAP⁺ cells (in stroma) were calculated. Quantifications were visualized with GraphPad Prism (v.10.4.1).

### T1 CRC tissue microarray

A cohort of 261 patients with non-pedunculated T1 CRC were selected from a Dutch multicentre CRC cohort study[37]. This case cohort consists of 50% random patients of a larger T1 cohort, supplemented with 50% of patients with an endpoint of interest (lymph node metastases and/or recurrence) as previously described. For each tumour specimen, three cores (Ø 0.6 mm) were punched out of both the tumour centre and invasive front and set into paraffin blocks using an automated tissue microarray. Tissue microarray blocks were cut into 4-µm thick sections and stained with antibodies against nucleus, PanCK, LAMC2 and FAP as described in the 'Immunohistochemistry of CRCs' section. After quality control, 232 tumours were analysed and quantified (175 N0M0, 44N⁺, 13M⁺), using QuPath (v.0.6.0) software for visualization and GraphPad Prism (v.10.4.1) for visualization.

### Organoid biobank

Organoid cultures were generated from punch biopsies (Ø roughly 3 mm) of fresh, surgically removed CRCs. Sampled histopathological

regions included: normal tissue adjacent to the tumour, adenomatous tumour component, tumour core (carcinoma) and the invasive front. After sampling of a fresh tumour by punch biopsies, the remaining tumour specimens were fixed, embedded in paraffin, sliced and stained with H&E for validation of accurate sampling by histopathological examination of the tissue surrounding the holes resulting from the punch biopsies. For organoid derivation, punch biopsies were minced with scissors and subjected to enzymatic digestion at 37 °C for 15–25 min with 1 mg ml$^{-1}$ collagenase (Sigma C9407) and 1 mg ml$^{-1}$ Dispase II (Gibco 11510536) in basal medium (advanced DMEM (Gibco) supplemented with 1% HEPES buffer (Gibco), 1% GlutaMAX (Gibco) and 1% Penicillin/ Streptomycin (Lonza)). The resulting tissue fragments were washed 3 times by means of centrifugation (500$g$, 4 min) and resuspension in 2 ml of basal medium, and then split into a fraction used for cryogenic preservation in Recovery Medium (Gibco, 11560446) and a fraction used for organoid derivation. The latter was resuspended in ice cold Matrigel (Corning) and plated in domes in prewarmed plastic culture plates. Following solidification (37 °C, 15 min) of the Matrigel, organoid culture medium (basal medium with 0.5 nM Wnt surrogate-FC fusion protein (U-Protein Express), 20% R-spondin conditioned medium (in-house production), 10% Noggin conditioned medium (in-house production), 1× B27 (Invitrogen), 1.25 mM $N$-acetylcysteine (Sigma-Aldrich), 50 ng ml$^{-1}$ recombinant human EGF (Invitrogen), 50 ng ml$^{-1}$ recombinant human insulin-like growth factor 1 (IGF1) (Biolegend), 50 ng ml$^{-1}$ recombinant human FGF2 (FGF-basic, Peprotech) and 500 nM A83-01 (Tocris)), supplemented with 100 µg ml$^{-1}$ Primocin (InvivoGen) and Rho-kinase inhibitor 10 µM Y-27632 (Gentaur), was added. Organoids were maintained in culture medium without Primocin at 37 °C with 5% CO$_2$ and passaged weekly by trypsinization (37 °C, 1–4 min, Trypsin-EDTA, Sigma T3924). After trypsinization for passaging, medium was supplemented with Y-27632 for 3 days. Cultures were regularly tested for mycoplasma contamination.

The availability of the organoid lines that have been generated in this study is restricted by the UMC Utrecht ethical committee. To receive these organoid lines, a request with the appropriate forms has to be made through this committee, which will determine whether the request corresponds with the informed consent of the patient.

### Organoid growth factor-dependency screens

To assess growth factor dependency of organoid lines, organoids were plated as single cells and cultured for 9 days in the presence or absence of indicated growth factors and inhibitors (Nutlin-3 (Sanbio 10004372), 5 ng ml$^{-1}$ recombinant human TGFB1 (Immunotools 11343160), 20 ng ml$^{-1}$ recombinant human BMP2 (Immunotools 11343273) and 20 ng ml$^{-1}$ recombinant human BMP4 (Immunotools 11345043), 1 µM afatinib (SelleckChem)). In brief, organoids were trypsinized with Trypsin-EDTA, filtered with a 30-µm cell strainer (Sysmex), seeded at 3,000 cells per condition in 10 µl Matrigel (Corning) drops on glass bottom 96-well angiogenesis culture and imaging plates (IBIDI), and overlayed with 70 µl of medium. Medium was refreshed on days 3 and 6 after seeding. Organoid growth was monitored by brightfield imaging using an EVOS imaging system (Invitrogen). To assess outgrowth efficiency per condition, the total organoid area on the brightfield images of the ninth day after seeding was determined. For this, images were segmented with OrganoSeg[64] software and analysed with a custom ImageJ/Fiji macro.

### WGS of organoids

For WGS, DNA was extracted from organoid cultures as early as possible (always before the eighth passage) using the DNA micro kit (Qiagen) according to the manufacturer's instructions. Truseq DNA nano WGS library preparation and sequencing (Illumina NovaSeq 6000 or X; 2× 150 bp; coverage 15–30×) were performed by the USEQ. Somatic variants were called using the nf-core implementation (oncoanalyser v.1.0.0: https://github.com/nf-core/oncoanalyser of the Hartwig Medical Foundation pipeline (https://github.com/hartwigmedical/ pipeline5). The pipeline was run in TUMOR_GERMLINE mode ('mode', 'wgts'). Relevant reference data, prebuilt indices and reference genome (Hartwig human reference GRCh38) were downloaded from the public repository before running the pipeline. *SMAD4* heterozygous loss was manually annotated based on CNA data of chromosome 18q.

For construction of phylogenetic lineage trees, short variants shared by many samples from the same patient were called and filtered using joint variant calling by GATK HaplotypeCaller (v.4.1.3, part of the NF-IAP pipeline; https://github.com/UMCUGenetics/NF-IAP). SMuRF (v.3.02, https://github.com/ToolsVanBox/SMuRF) was used to filter somatic variants (absent in the normal samples) from the multi-sample VCF files. High-confident somatic small variants with a variant allele frequency of more than 0.25 in at least 1 sample were included to generate a binary mutation table. The R package ape (v.5.8) was used to construct and visualize the lineage trees.

### Plate-based scRNA-seq

To characterize cell type composition in early-stage CRC, we performed scRNA-seq on tissue fragments of five CRCs that were cryopreserved in parallel to organoid establishment of the punch biopsies mentioned in the section 'Organoid biobank'. For this, tissue fragments were thawed, washed with basal medium and trypsinized to single-cell suspensions using TrypLE (Gibco 12604013) supplemented with 10 µM Y-27632 for 5 min at 37 °C. To distinguish epithelial, immune and stromal cell populations and sort equal amounts of these three populations, single-cell suspensions were stained with DRAQ7 (Invitrogen, 1:200), phycoerythrin anti-human CD326 (EpCAM) (324205 9C4, Biolegend, 1:200) and fluorescein isothiocyanate (FITC) anti-CD45 (368507 2D1, Biolegend, 1:200) in advanced DMEM/F12 for 30 min on ice. Viable single cells (DRAQ7$^-$) were sorted (BD FACSAria III) into 384-well cell-capture plates from Single Cell Discoveries, which contain a 50-nl droplet of well-specific barcoded primers and 10 µl of mineral oil (Sigma M8410). After sorting, plates were briefly centrifuged (500$g$) and then kept on dry ice until further storage at −80 °C. scRNA-seq was performed by Single Cell Discoveries according to an adapted version of the SORT-seq protocol[70] with primers described in ref. 71. Cells were heat-lysed at 65 °C followed by complementary DNA (cDNA) synthesis. After second-strand cDNA synthesis, all the barcoded material from one plate was pooled into one library and amplified using in vitro transcription. Following amplification, library preparation was performed following the CEL-Seq2 protocol[72] to prepare a cDNA library for sequencing using TruSeq small RNA primers (Illumina). The DNA library was sequenced by paired-end sequencing on an Illumina NextSeq 500, high output, with a 1× 75 bp Illumina kit (read 1, 26 cycles; index read, 6 cycles; read 2, 60 cycles).

### scRNA-seq analysis

For alignment of reads, an adapted version of the nf-core scrnaseq pipeline (v.2.4.0)[73] was used (https://github.com/gowanaka/nf-core-scrnaseq). In brief, STARsolo (v.2.7.10b) was used to align reads to a custom GRCh38 human reference transcriptome including External RNA Controls Consortium (ERCC) spike-ins. Following mapping, count matrices were generated with STARsolo (v.2.7.10b). Gene expression was analysed using Seurat (v.5.0.1)[74]. Cells with less than 25% mitochondrial content, less than 25% exogenous ERCC spike-in content, more than 1,000 transcript counts (nCount_RNA) and more than 500 unique detected genes (nFeature_RNA) were selected for downstream analysis. Mitochondrial transcript counts were removed before count normalization and scaling by the Seurat NormalizeData and ScaleData functions, respectively. Unsupervised clustering was used to cluster cells according to the standard Seurat workflow. Gene expression signature scores were calculated with the Seurat AddModuleScore function. Differential expression analysis was performed with the FindAllMarkers function.

## CosMx single-cell spatial transcriptomics

To map spatial distribution of cell types identified with scRNA-seq, we performed Nanostring CosMx single-cell spatial transcriptomics[75] on one T1 CRC included in the organoid biobank (pt5/ptD; Human CosMx Universal Cell Characterization Panel; 1,000 gene targets; Fig. 3) and 11 CRC specimens temporally surrounding the moment of malignant transformation (3× intramucosal carcinoma, 5× T1 sm1 and 3× T1 sm3; Human CosMx 6,000 Discovery Panel; 6,000 gene targets, Fig. 5). Slides were stained with segmentation markers (Human Universal Cell Segmentation Kit, RNA, Bruker Spatial Biology, 531-121500020) for nuclei (4,6-diamidino-2-phenylindole (DAPI)), cell membranes (CosMx Hs CD298/B2M Segmentation Marker Mix, Ch2 RNA), epithelial and immune cells (CosMx Hs PanCK/CD45 Marker Mix Ch3/Ch4, RNA, Bruker Spatial Biology) and macrophages (CosMx Hs CD68 A La Carte Marker, Ch5 RNA, Bruker Spatial Biology, 531-121500022, second experiment only). After filtering on the basis of standard quality control, cells were labelled according to predicted cell type using label transfer from the Seurat package (v.5.0.1)[74], with our early-stage CRC scRNA-seq dataset as a reference. Query and reference datasets were downsampled to only include overlapping gene targets before label transfer and both were normalized and scaled using the SCTransform method. Principal component analysis (PCA) was performed for the scRNA-seq data. FindTransferAnchor() and TransferData() were used to anchor the scRNA-seq PCA reference data to the CosMx query data and transfer cell type labels. After label transfer, raw CosMx data were normalized and scaled again using SCTransform. PCA was performed on normalized data. Uniform manifold approximation and projection (UMAP) (30 principal components, min.dist = 0.01) was used for dimensionality reduction. Nearest neighbour graphs were constructed using the first 30 principal components. Unsupervised clustering was performed using the Seurat default implementation of the Louvain algorithm (resolution 0.7). In plots where cell type labels are shown, only cells that were annotated with prediction.score.max ≥ 0.6 are shown.

Subclustering of FAP+ CAFs and epithelial clusters (pt5; Fig. 3) was performed using the Louvain algorithm with resolutions 0.2 and 0.05, respectively. FAP+ CAF subclusters were assigned to a CAF subtype on the basis of marker gene expression. The epithelial HRC subcluster was annotated on the basis of marker gene expression. Epithelial subclustering of the other 11 CRC specimens was restricted to the cancer epithelial clusters identified by means of clustering per specimen (resolution 0.7). We selected clusters with high HRC program expression within each specimen separately by reclustering cancer epithelium (resolution 0.7 and 0.2). We did not detect a HRC cluster in specimens T1_NANO_022 (incomplete invasive front), T1_NANO_030 and T1_NANO_031 (both intramuscosal carcinomas). For single-cell spatial plots of epithelial cells, cells were filtered by PanCK staining intensity (lowest tenth percentile excluded).

## Neighbourhood analysis

Profiling spatial context of cancer cells, we performed cellular neighbourhood analysis for the oncofetal and cancer stem cells of the 11 CRC specimens analysed with the Nanostring CosMx 6,000 gene panel. In brief, we ran RANN's nn2() function per sample to find the neighbours of a cancer cell within a 50-µm radius. The output cells × clusters matrix was used to count neighbouring cell types for composition analysis (sccomp[76]), sum expression profiles across all neighbours for neighbourhood differential expression analysis and to cluster cells on the basis of neighbour cell composition using $k$ means clustering ($k$ = 10).

## NicheNet analysis

NicheNet analysis was performed on the GeoMx WTA invasive front segments and CosMx 'niche3' cells (oncofetal niche) with nichenetr[77] (v.2.0.0; receivers = epithelial segments; senders = stromal segments).

Genes with expression below the 25th quantile across all sender or receiver segments were excluded. Ligands of interest were prioritized on the basis of cumulative interactive potential across all the coreHRC genes.

## Fibroblast immortalization and culture

Fibroblast lines were derived from early passage cultures of the punch biopsies that were used to establish organoids ('Organoid biobank' section). In brief, fibroblasts adhering to the plastic bottom of the organoid culture plates were maintained with DMEM supplemented with 10% fetal bovine serum (Bodinco) and 1% penicillin/streptomycin (Lonza) after organoid removal for passaging and subjected to simultaneous lentiviral transduction with hTERT (third-generation adaptation of Addgene no. 85140) and BMI1 (no. 12240) overnight[78]. Fibroblast lines were passaged weekly by trypsinization.

## Organoid–fibroblast cocultures

Organoids were cocultured with fibroblasts in a transwell setup (Polycarbonate Cell Culture Inserts with 0.4 µm pore size in a six-well plate format, ThermoFisher) for 48 h in growth factor depleted medium (basal medium, B27 (Invitrogen) and 1.25 mM $N$-acetylcysteine (Sigma-Aldrich)). Fibroblasts were trypsinized, counted and seeded as a single-cell suspension (300,000 cells per well) in fibroblast culture medium (above) on plastic or in 200 µl of Matrigel (Corning) 1 day before coculture to allow for adherence to the plastic substrate. To start coculture, 5-day old organoids were plated in 150 µl of Matrigel (Corning) on top of the transwell membranes. To harvest RNA, transwell culture inserts with organoids were removed and organoids and fibroblasts were lysed separately, followed by RNA extraction using the Nucleospin RNA isolation kit (Macherey-Nagel 740955), according to the manufacturer's instructions. To investigate matrix-induced and juxtacrine effects, organoids and fibroblasts were seeded simultaneously in collagen-Matrigel (25%/25%) (Collagen Type I Corning 354236) mixtures and cocultured in growth factor depleted medium for 2 days or 5 days before flow cytometric quantification of EMP1-mNeon+ cells.

## RNA-seq of organoid–fibroblast cocultures

RNA-seq library preparation was performed by the USEQ according to the Illumina TruSeq stranded PolyA protocol. Libraries were sequenced in two runs on an Illumina NextSeq 2000 (run 1: 20 samples, 2 × 50 bp paired-end sequencing, index 1: 17 cycles, read 1: 50 cycles, index 2: 8 cycles, read 2: 50 cycles and run 2: 11 samples, 1 × 50 bp single-end sequencing, index 1: 17 cycles, read 1: 50 cycles, index 2: 8 cycles). For alignment of reads, the nf-core RNA-seq pipeline (v.3.14.0) was used (https://doi.org/10.5281/zenodo.1400710, ref. 79) with the option 'star_salmon'. Briefly, FASTQ files underwent quality control (FastQC v.0.12.1), adaptors were trimmed (Trim Galore! v.0.6.7), reads were aligned to the GRCh38 human reference transcriptome (STAR v.2.7.9a) and a gene expression matrix was generated (Salmon v.1.10.1). Differential expression analysis at the gene and gene set level (ssGSEA/GSEA) was performed using DESeq2 (v.1.38.3). Genes that had at least a count of 10 in at least 4 samples were retained, VST normalized and a PCA was conducted. Organoid and fibroblast samples were batch corrected by sequencing run and Line_ID, respectively.

## Generation of *EMP1*<sup>mNeon</sup> organoid knock-in

*EMP1*^mNeon knock-in organoids (pt5 inv) were generated by in-trans paired Cas9 targeting as described in ref. 80. SpCas9 (Addgene no. 48139) locus-specific expression vectors were generated according to published protocols[81] (guide 5′-TCCTGAGAAAGAAATAAGGC-3′). The targeting vector was generated by introducing 449-nucleotide homology arms and flanking EMP1 guide sequences into a custom-made vector (IRES-mNeon-NLS-P2A-iCasp9-WPRE-pA-PGK-PuroR-pA; Addgene no. 251175) using golden gate assembly. For transfection, organoids were trypsinized to cell clumps containing roughly 5 cells (around

$1 \times 10^6$ cells in total) and coelectroporated with 4 µg of SpCas9 DNA and 11 µg of targeting vector using the NEPA21 Super Electroporator (Nepagene) following the conditions described in ref. 82. Electroporated cell clumps were plated in Matrigel overlaid with organoid culture medium supplemented with 10 µM Y-27632 Rho-kinase inhibitor for the first 3 days. Targeted cells were selected using 1 µg ml⁻¹ puromycin and maintained as polyclonal populations. To confirm EMP1-mNeon-NLS fluorescence and nuclear localization, live organoids were incubated with Hoechst 33342 (ThermoFisher Scientific 62249, 1:5,000, 30 min, 37 °C with 5% CO$_2$) to visualize nuclei and imaged with a Leica SP8 scanning confocal microscope using LAS X software (v.3.5.7.23225).

### EMP1$^{mNeon}$ organoid reporter-based screen

To screen for ligands that induce oncofetal tumour cell states, *EMP1*$^{mNeon}$ organoids were trypsinized (TrypLE), plated as single cells (300 cells per µl, filtered with a 40-µm strainer) and treated with candidate ligands 5 days after plating. Single candidate stimuli or combinations were added in growth factor-deprived medium (basal medium with B27 (Invitrogen) and 1.25 mM *N*-acetylcysteine (Sigma-Aldrich) after 2 washes with basal medium and included: TGFβ1 (5 ng ml⁻¹; Immunotools 11343160), TGFβ3 (5 ng ml⁻¹; Immunotools 11344483), PGE2 (10 µM; Tocris 2296), PGD2 (10 µM; Merck 538909), CXCL12 (40 ng ml⁻¹; Immunotools 11343363), FGF2 (50 ng ml⁻¹; Peprotech 100-18B), IGF1 (50 ng ml⁻¹; Biolegend 590904), FGF7 (50 ng ml⁻¹; Peprotech 100-19), GREM1 (100 ng ml⁻¹; Peprotech 120-42-50UG), SFRP1 (100 ng ml⁻¹; Peprotech 120-29), SFRP2 (100 ng ml⁻¹; Biotechne 1169-FR-025), GDNF (50 ng ml⁻¹; ThermoFisher 450-10-10UG), IL-36A (50 ng ml⁻¹; ThermoFisher 200-36A-2UG), IL-36B (50 ng ml⁻¹; ThermoFisher 200-36B-2UG), CXCL14 (40 ng ml⁻¹; Immunotools 11345190), BMP2 (20 ng ml⁻¹; Immunotools 11343273), BMP4 (20 ng ml⁻¹; Immunotools 11345043), hepatocyte growth factor (50 ng ml⁻¹; ThermoFisher 100-39-10UG), vascular endothelial growth factor (50 ng ml⁻¹; ThermoFisher 100-20-2UG), WNT5A (20 ng ml⁻¹; Biotechne 645-WN-010), IL-6 (100 ng ml⁻¹; Stem Cell Technologies 78050.1), OSM (50 ng ml⁻¹; R&D Systems 295-OM-010), IL-1B (20 ng ml⁻¹; ThermoFisher 200-01B-10UG), tumour necrosis factor (10 ng ml⁻¹; Knoll AG), IL-27 (100 ng ml⁻¹; ThermoFisher 200-38-2UG) and interferon-gamma (100 ng ml⁻¹, ThermoFisher 300-02-20UG). The percentage of *EMP1*$^{mNeon}$ positive cells among live cells was measured 24 h after addition of candidate stimuli as described below.

### Flow cytometry

Single-cell organoid suspensions were prepared by trypsinization with Trypsin-EDTA for 5 min at 37 °C. Flow cytometry measurements were performed on a BD FACSCelesta CellAnalyzer. Single live cells (DAPI⁻) were gated in the BV421 channel, mNeon and phycoerythrin fluorescence were measured in the FITC-A and PE-A channels, respectively. Gates were set on the basis of negative control samples, that is, parental organoid line or unstained cell suspensions. To separate organoid and fibroblast cells in juxtacrine cocultures (Extended Data Fig. 8g), cells were stained with phycoerythrin anti-human CD326 (EpCAM) (324205 9C4, Biolegend, 1:400). To measure MHCI levels (Extended Data Fig. 8i,j), cells were stained with phycoerythrin anti-human HLA A/B/C (311405 W6/32, Biolegend, 1:400). Flow cytometry data were analysed and visualized using BD FACSdiva software and the free online tool https://floreada.io.

### Immunofluorescence of organoids

Organoids form coculture experiments were immunostained for LAMC2 protein levels as described previously[83]. In brief, organoids were dislodged from Matrigel matrix domes by incubation in basal medium supplemented with 1 mg ml⁻¹ dispase for 30 min at 37 °C/5% CO$_2$ and pelleted after several washing cycles with basal medium. Organoids were fixed in 4% paraformaldehyde in PBS on ice for 45 min. Fixed organoids were transferred to repellent plates (Greiner Bio-One). Permeabilization, blocking and antibody incubation steps were performed with

organoid washing buffer (0.1% Triton X-100 in PBS and −0.2% wt/vol BSA) at 4 °C on a shaker. Primary antibodies used: LAMC2 (AMAb91098, Atlas Antibodies, 1:500) and beta-catenin (C2206, Sigma-Aldrich, 1:500). Secondary antibodies used: Alexa 647 anti-mouse (Invitrogen A21236; 1:500) and Alexa 568 anti-rabbit (Invitrogen A11011; 1:1,000) and Hoechst. Organoids were mounted in clearing solution (ddH$_2$O, 60% (vol/vol) glycerol and 2.5 M fructose) and imaged on a Zeiss LSM880 confocal laser scanning microscope at ×40 magnification. Images were processed in Fiji software. Hoechst was used as a nuclear marker and beta-catenin to mark cell boundaries, to allow for LAMC2 quantification at single-cell resolution. Statistical analysis was performed in GraphPad Prism (v.10.4.1).

### RNA-seq and qPCR of organoids treated with TGFβ and prostaglandins

Organoids were treated with a combination of TGFβ1 (5 ng ml⁻¹; Immunotools 11343160), TGFβ3 (5 ng ml⁻¹; Immunotools 11344483), PGE2 (10 µM; Tocris 2296) and PGD2 (10 µM; Merck 538909) in growth factor-deprived medium (basal medium with B27 (Invitrogen) and 1.25 mM *N*-acetylcysteine (Sigma-Aldrich) after 2 washes with basal medium, 5 days after trypsinization to single cells. After 24 h of induction, RNA was extracted using the Nucleospin RNA isolation kit (Macherey-Nagel 740955), according to the manufacturer's instructions. Library preparation (directional messenger RNA; poly-A enrichment) and sequencing (NovaSeq X Plus Series PE150) were performed at Novogene and data were analysed with DESeq2 (v.1.38.3) and clusterProfiler (v.4.8.3) with method fgsea (v.1.24.0). Differential expression analysis was corrected for Patient ID. For quantitative PCR (qPCR), cDNA was generated from RNA using the iScript cDNA Synthesis Kit (Bio-Rad). For qPCR, 20 ng of cDNA was mixed with 0.5 µM forward and reverse primer each and 5 µl of PowerTrack SYBR Green (Applied Biosystems) per well. qPCR was performed on the Bio-Rad CFX96 and results were analysed with Microsoft Excel (v.16.95) using the ΔΔCt method with *ACTB* and *PBGD* as reference genes. Sequences of primers used for qPCR can be found in Supplementary Table 8.

### Analysis of published scRNA-seq data

Published scRNA-seq data of human CRCs from refs. 45,55,54 (see 'Data availability' for accession codes) were integrated using the Seurat package (v.5.0.1)[74] in R (v.4.2.0) with harmony integration according to the standard workflow. Clusters were annotated using cell type annotations included with the published datasets and marker genes of the clusters. For trajectory inference analyses, Monocle3 (ref. 84) (v.1.4.26), CytoTRACE2 (ref. 85) (v.1.1.0) and Slingshot[86] (v.2.16.0) R packages were used to calculate single-cell potency and pseudotime scores. The CytoTRACEkernel from CellRank[87] (v.2.0.7) was used to compute a transition matrix and construct pseudotime-based streamline plots featured in Fig. 5j. The bottom 2% low density areas in the UMAP space were excluded from these analyses.

### Statistics and reproducibility

Statistical analysis was performed as noted in the figure legends using R (R base (v.4.2.0 or later), ggplot2 (v.3.5.1), ggpubr (v.0.6.0) Seurat (v.5.0.1)) and GraphPad Prism (v.10.4.1). Data distribution was assumed to be normal, but this was not formally tested. All statistical tests were two-tailed. Where not stated, $P < 0.05$ or false discovery rate (FDR) < 0.05 was deemed to be statistically significant. The Benjamini–Hochberg method was used to correct the *P* value for multiple testing. For comparisons between more than two sample groups, one-way analysis of variance (ANOVA) was performed, using Tukey's HSD for post hoc analysis. Data are presented as mean ± standard deviation, unless otherwise stated in the figure legend. For GSEA results, an FDR < 0.25 was deemed to be statistically significant in line with ref. 88. Representative images (Fig. 4f and Extended Data Figs. 7a and 9b) depict consistent results that were observed in at least two independent experiments.

## Reporting summary

Further information on research design is available in the Nature Portfolio Reporting Summary linked to this article.

## Data availability

WGS data of patient-derived organoids (EGAD50000002204), RNA-seq data of organoids and organoid–fibroblast cocultures (EGAD50000002202) and scRNA-seq data of early-stage CRCs (EGAD50000002203) are available through the European Genome–Phenome Archive (EGA) under accession number EGAS50000001532. An adapted version of the nf-core scrnaseq pipeline is available at GitHub (https://github.com/gowanaka/nf-core-scrnaseq). Bulk (Nanostring GeoMx) and single-cell (Nanostring CosMx) spatial transcriptomic data of T1 CRCs and processed expression data (RNA-seq of organoids and organoid–fibroblast cocultures and scRNA-seq of early-stage CRC biopsies) are available at Zenodo (https://doi.org/10.5281/zenodo.17671259)[89]. Expression data (scRNA-seq, Nanostring GeoMx CTA, Nanostring CosMx fig. 3) can be accessed through an interactive dashboard at https://snippertlab.nl/resources. Published scRNA-seq data of human CRCs were obtained from GSE144735 and GSE132465 (ref. 45); GSE201349 (ref. 55) and GSE178341 (ref. 54). Source data are provided with this paper.

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

**Acknowledgements** We thank all members of the Snippert laboratory for their support, discussions and reagents (in particular B. Ponsioen for immunofluorescent staining quantifications); Utrecht Platform for Organoid Technology (in particular A. Brousali, A. Snelting and J.-H. Venhuizen) for patient inclusion; the UMCU Flow Core Facility (in particular M. O. Nordkamp) for their assistance with cell sorting; L. Kleij for support with flow cytometry and microscopy experiments; I. Nijman and USEQ for performing GeoMx, CosMx, bulk RNA-seq, WGS, and for advice regarding experimental sequencing setup and imaging of immunofluorescent stainings. USEQ is subsidized by the University Medical Center Utrecht and The Netherlands X-omics Initiative (NWO project no. 184.034.019). We are grateful for all patients who participated in this study. This work has been supported by research grants from the ZonMw NWO VIDI program (grant no. 09150172010017 to H.J.G.S.), ERC consolidator (TRANSFORMATION, grant no. 101125393 to H.J.G.S.) and Dutch Cancer Society (grant no. EXPL2024/1 16037 to H.J.G.S.). This work is part of the Oncode Institute, which is partly financed by the Dutch Cancer Society.

**Author contributions** J.R.B.d.A., J.H.H., S.R.B. and S.E.M.v.d.H. contributed equally to this work. J.R.B.d.A., J.H.H., S.R.B. and S.E.M.v.d.H. analysed and interpreted the data. J.R.B.d.A., J.H.H., S.E.M.v.d.H. and I.V.-K. established and characterized the organoid biobank. J.R.B.d.A., J.H.H., I.v.L., D.S.C. and S. Mertens performed in vitro organoid and fibroblast experiments. J.H.H. designed and created organoid knock-in reporter line. S.E.M.v.d.H performed and quantified immunofluorescence on CRC specimens and organoids. S.R.B. analysed spatial transcriptomics and bulk RNA-seq data. J.R.B.d.A., J.H.H., S.E.M.v.d.H. and M.C.P. generated early-stage CRC scRNA-seq data. J.R.B.d.A. and J.H.H. analysed early-stage CRC scRNA-seq data. J.R.B.d.A. integrated public scRNA-seq data and analysed public bulk CRC expression data. L.J.K. constructed data dashboard. S.R.B., A.v.H. and S. Middelkamp analysed WGS data. R.G. provided support with sequencing, spatial transcriptomics and slide imaging. J.S., R.B. and O.K. obtained patient consent. L.v.d.S. selected tissue samples for spatial transcriptomics. S.v.K. and F.M. prepared tissue samples for spatial transcriptomics. M.M.L. provided histopathological examination. M.M.L., L.M.G.M. and H.J.G.S. conceived the study. H.J.G.S. supervised the study. J.R.B.d.A., J.H.H. and S.R.B. visualized data. J.R.B.d.A., J.H.H. and H.J.G.S. wrote the paper with contributions and approval from all authors. H.J.G.S. and L.M.G.M. acquired funding.

**Competing interests** S.E.M.v.d.H., M.M.L., L.M.G.M. and H.J.G.S. are inventors on a patent related to biomarkers of metastatic disease. The other authors declare no competing interests.

**Additional information**
**Correspondence and requests for materials** should be addressed to Hugo J. G. Snippert.

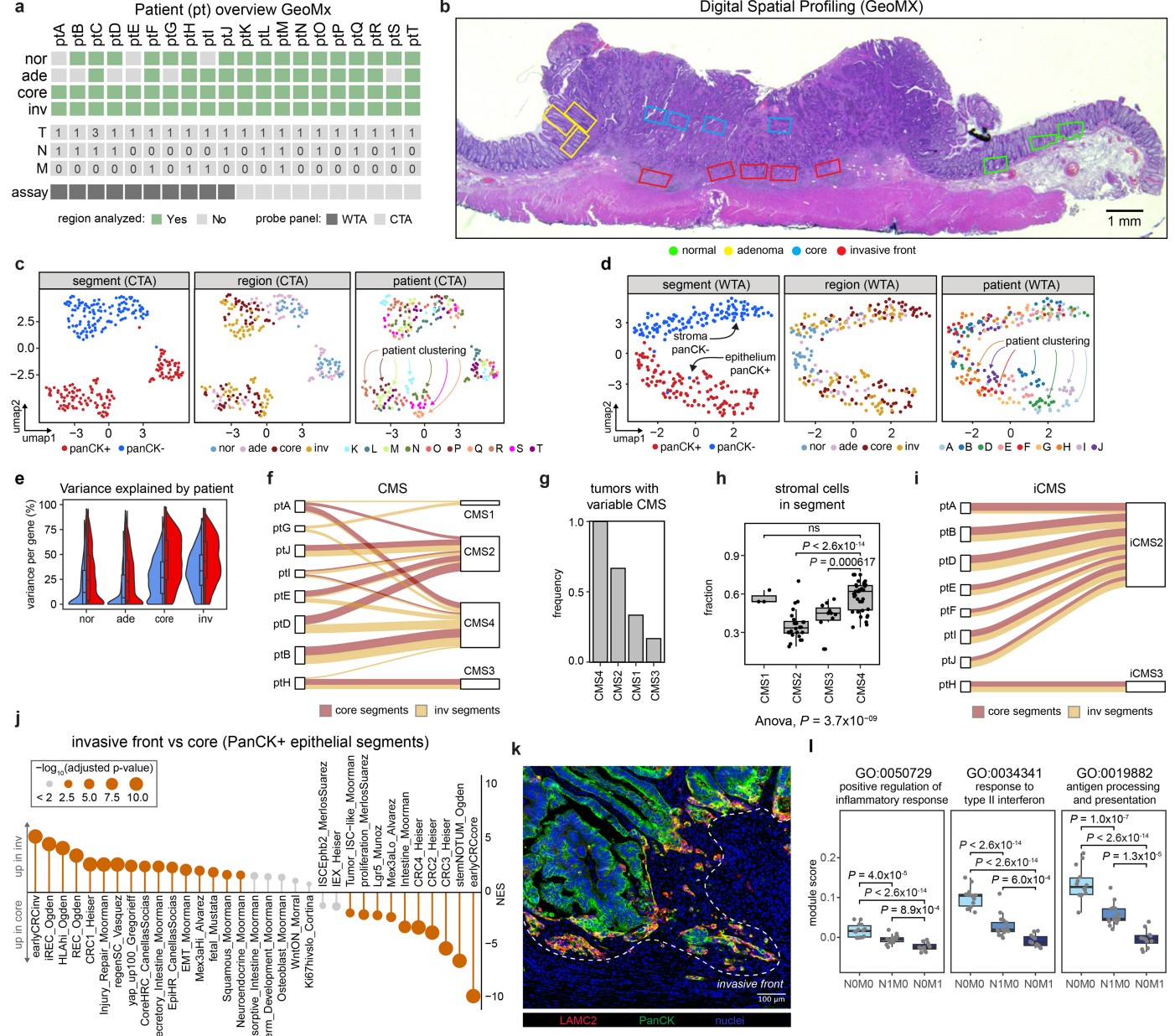

**Extended Data Fig. 1 | Bulk spatial transcriptomics of early-stage CRCs.**
a) Sample overview of spatial profiling experiment. ROI placement in pathologist-defined histopathological regions, tumor TNM status and probe panel are indicated per tumor. 19 T1 CRCs were analyzed (CTA: *n* = 10 and WTA: *n* = 9). b) H&E staining of T1 CRC specimen shown in Fig. 1b. Histopathological regions (nor, ade, core and inv) were annotated by a pathologist. c) UMAP of gene expression profiles of all segments (dots) in the CTA cohort. Dot colors represent segment type (panCK+ or panCK-), histopathological region or patient. d) UMAP of gene expression profiles of all segments (dots) in the WTA cohort. Dot colors represent segment type (panCK+ or panCK-), histopathological region or patient. e) Violin plot depicting variance explained by patient per gene (*n* = 16,000) in epithelial (red) and stromal (blue) segments for each histopathological region (WTA; boxes: interquartile range; black bars: median, whiskers: 1.5x interquartile range). f) Sankey flow diagram of CMS subtypes assigned to tumor core (red) and invasive front (yellow) ROIs. g) Distribution of CMS subtypes in heterogeneous tumors (*n* = 5). Note that all

multi-subtype tumors include a CMS4 component. h) Analysis of 94 ROIs from 9 patients shows high stromal cell density in the CMS4 subtype (boxes: interquartile range; black bars: median, whiskers: 1.5x interquartile range; ANOVA, Tukey HSD). i) Sankey flow diagram of iCMS subtypes assigned to tumor core (red) and invasive front (yellow) epithelial segments. j) GSEA of various intestinal stem cell, regenerative, oncofetal and metastasis-associated signatures between core (*n* = 42) and invasive front (*n* = 43) epithelial segments (PanCK + ). Dot size represent significance (adjusted p-values, gray: adjusted p-value > 0.01; permutation test with FDR). NES: Normalized enrichment score. k) Immunofluorescent staining showing local presence of HRC marker gene LAMC2 (red, *n* = 2 tumors) at the invasive front of early-stage CRCs (stage T2). Dashed line indicates tumor border. PanCK+ epithelial cells (green). SYTO13+ nuclei (blue). l) Module scores per metastatic status for indicated GO-terms (*n* = 43 PanCK+ segments; boxes: interquartile range; black bars: median, whiskers: 1.5x interquartile range; ANOVA, Tukey HSD).

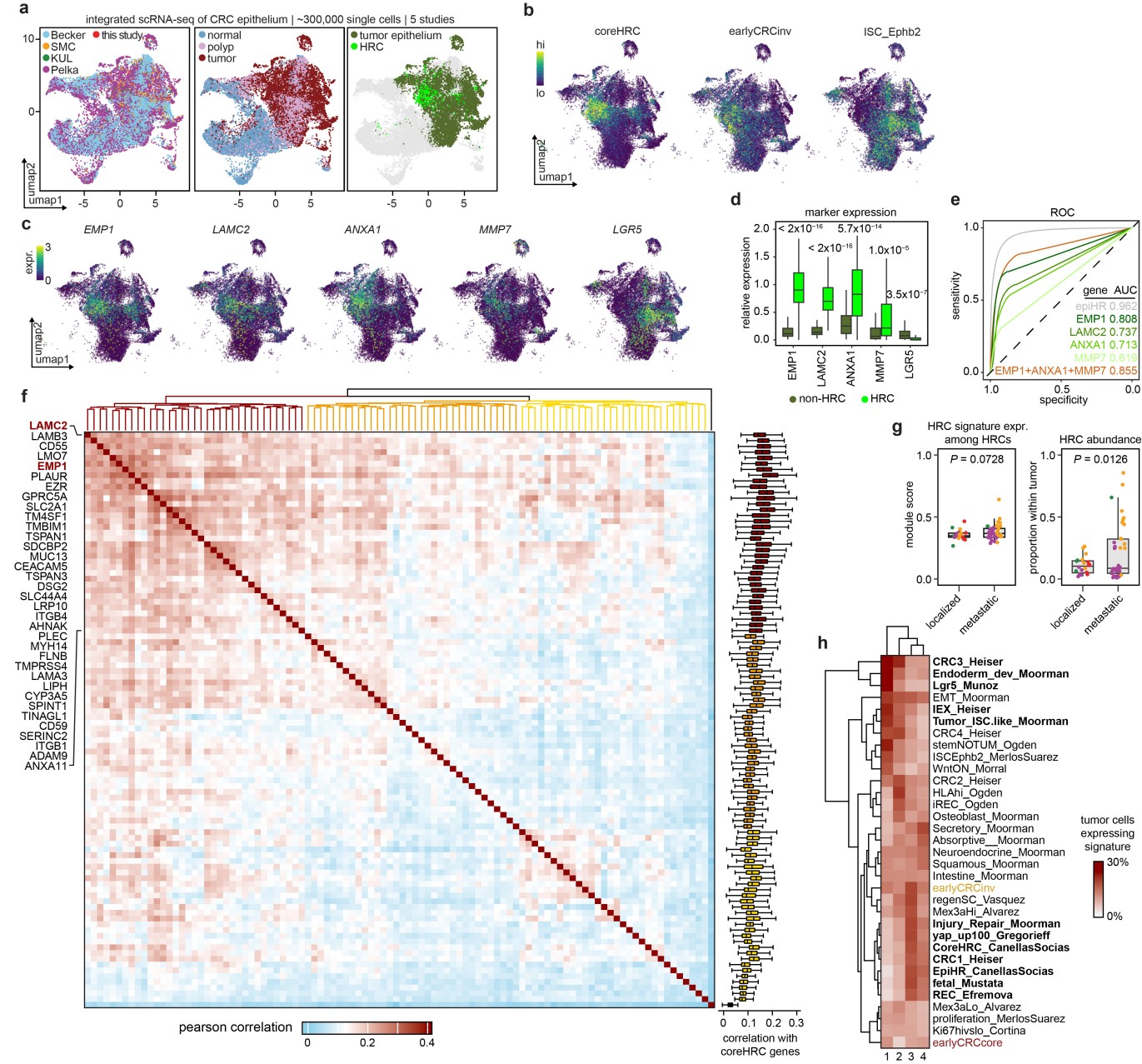

**Extended Data Fig. 2 | Integrated scRNA-seq analysis of colorectal cancer tumor cells.** a) UMAP of epithelial cells (>300,000) from integrated dataset of 5 scRNA-seq studies of CRC[45,54,55]. Colors represent study origin (left), histopathological region (middle) or tumor epithelial states (tumor epithelium: cluster contains <10% cells from normal tissue, HRC: High Relapse Cell (top 10 percent highest expressors of HRC program across tumor cells)). b) UMAP of tumor-specific clusters with color scale for coreHRC[14], earlyCRCinv (this study) and ISC-Ephb2[29] signature scores. c) UMAP of tumor-specific clusters with color scale for relative expression of indicated marker genes. d) Relative expression of indicated marker genes in High Relapse Cells (HRC) versus all other tumor epithelial cells (non-HRC). Boxplots: distribution of per patient mean expression of HRCs and non-HRCs (n = 57 patients; boxes: interquartile range; black bars: median, whiskers: 1.5x interquartile range; t-tests). e) Receiver Operating Characteristic (ROC) curves for indicated classifiers (genes and gene combinations) of HRC state. Dotted line denotes curve of random classifier.

f) Gene-gene correlation matrix of coreHRC signature (100 genes) within tumor cells (n = 195,356 single cells). Boxplot on the right: distribution of how each gene correlates with all other coreHRC genes (boxes: interquartile range; black bars: median, whiskers: 1.5x interquartile range). Genes in red cluster are annotated. LAMC2 and EMP1 are highlighted in red. g) Left: degree of HRC signature expression among HRCs from localized and metastatic primary CRCs. Points represent averages of cells from individual tumors (n = 57 tumors, t-test), colored by study as in left panel of **a**. Right: Relative frequency of HRCs within localized and metastatic primary CRCs. Points represent single tumors, colored by study as in **a** (n = 57 tumors, t-test). Boxes: interquartile range; black bars: median, whiskers: 1.5x interquartile range. h) Average proportion (%) of epithelial cells within tumors (MSI excluded) of indicated stages that express indicated transcriptional signatures (top 10th percentile among all tumor cells). Signatures significantly different between non-metastatic (stage I and II) and metastatic (stage III and IV) tumors are highlighted in bold (t-test, p < 0.05).

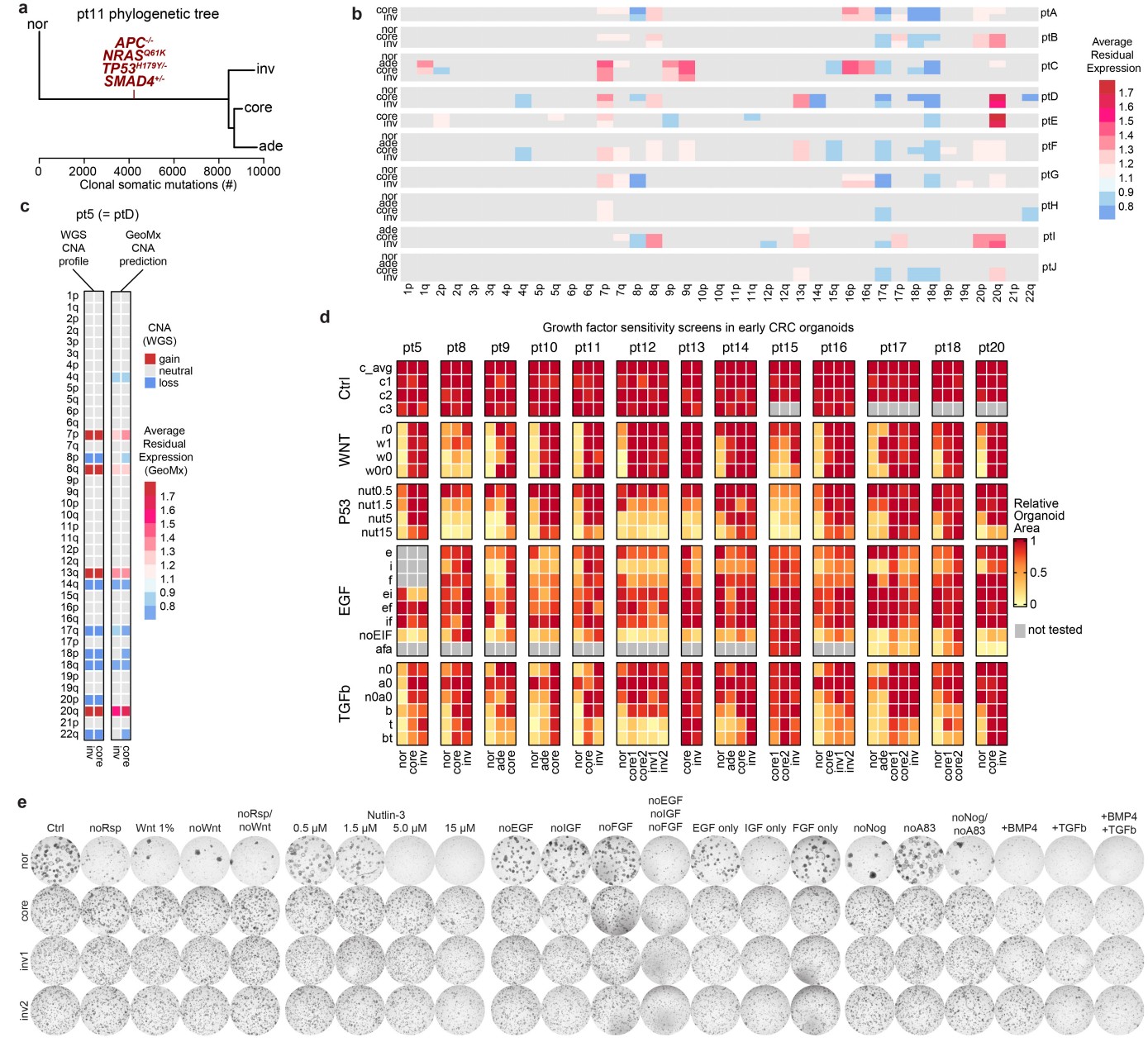

**Extended Data Fig. 3 | Genetic and functional characterization of tumor cells from tumor core and invasive front.** a) Phylogenetic lineage tree of patient 11 representing evolutionary relationship between regional biopsies based on WGS data. Mutations and stage of acquisition are annotated. b) Copy number alteration (CNA) profile predictions of epithelial segments from spatial transcriptomics based on average residual expression from inferCNV (Methods). c) pt5/ptD, for which both regional organoid WGS and spatial transcriptomics data was available, shows high concordance between WGS inferred CNA profile and expression-predicted CNA profile. d) Heatmap depicting organoid growth efficiency of all screened organoid lines in indicated medium compositions. c: control medium (standard organoid

culture medium (Methods)), c_avg: average of control conditions c1-c3, used for normalization (equal to maximum relative organoid area of 1), r0: no R-spondin, w1: 1% WNT (compared to WNT concentration in standard culture medium), w0: no WNT, nut0.5: 0.5 µM nutlin-3, nut1.5: 1.5 µM nutlin-3, nut5: 5 µM nutlin-3, nut15: 15 µM nutlin-3, e: only EGF (no IFG1 and FGF2), i: only IGF1 (no EGF and FGF2), f: only FGF2 (no EGF and IGF1), noEIF: no EGF, IGF1 and FGF2, afa: afatinib 1 µM (in noEIF medium), n0: no Noggin, a0: no A83, n0a0: no Noggin and no A83, b: 20 ng/ml BMP4 (in n0a0 medium), t: 20 ng/ml TGFβ1 (in n0a0 medium): bt: BMP4 and TGFβ1 (in n0a0 medium), pt: patient. e) Representative brightfield images of growth factor dependency screen (patient 16, day 9 after seeding as single cells).

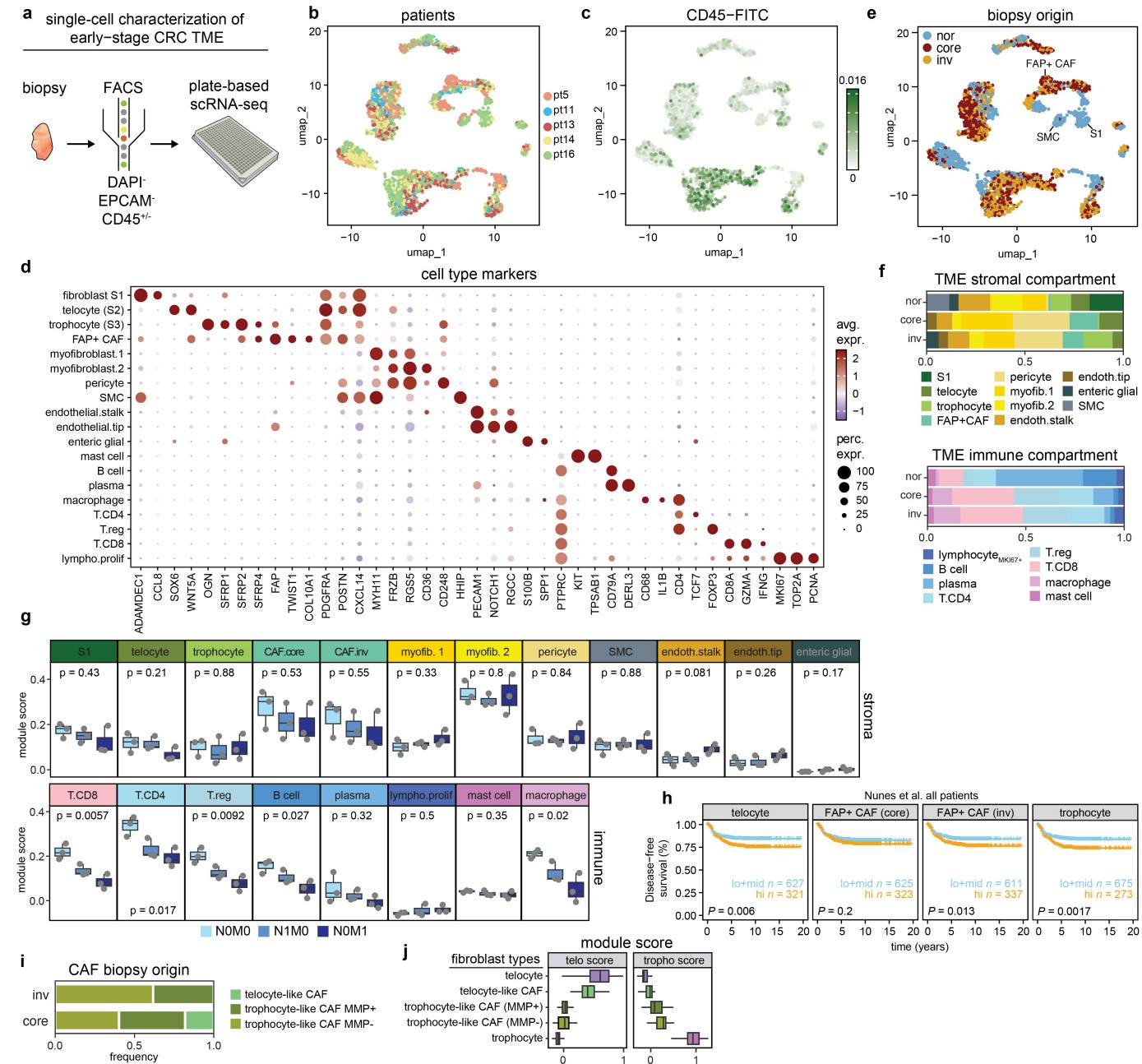

**Extended Data Fig. 4 | Single-cell RNA sequencing of multiregional punch biopsies of early-stage CRC.** a) Sorting strategy for plate-based scRNA-seq (Methods). Approximately equal numbers of immune cells and non-immune stromal cells were sorted (based on CD45 staining) and sequenced. Illustration from NIAID NIH BIOART (BIOART-000005). b) UMAP of early-stage CRC TME scRNA-seq overlayed with patient identity per cell. c) As in **b**, but overlayed with relative FITC-CD45 fluorescence signal measured during FACS. d) Normalized gene expression levels of cell type markers in early-stage CRC TME. Colors indicate expression level. Dot size represents percentage of cells in which indicated transcript was detected. e) As in **b**, but overlayed with regional biopsy identity of the single cells: normal (nor, blue); tumor core (core, red); and invasive front (inv, yellow). f) Frequency of TME cell types per biopsy origin. Top: stromal cell types; bottom: immune cell types. g) Boxplots showing relative signature scores for indicated stromal and immune cell types in non-epithelial segments of spatial transcriptomic dataset of early-stage CRC.

Cell type signatures were derived from the scRNA-seq dataset of early-stage CRC (Fig. 3). Points represent average signature score per patient ($n$ = 9) and are grouped by metastasis status (boxes: interquartile range; black bars: median, whiskers: 1.5x interquartile range; ANOVA p-values are shown). h) Prognostic value of CRC fibroblast subtypes in complete Nunes et al. cohort[49]. Kaplan-Meier curves of disease-free survival of all four CMS tumor subtypes combined, stratified by low/mid versus high expression of indicated fibroblast subtype signatures (top 100 region-specific marker genes, Supplementary Table 5, p values of log-rank test are shown in each plot). i) Frequency of CAF subtypes in total CAF population per biopsy origin. j) Trophocyte (top) and telocyte (bottom) signature scores in fibroblast subpopulations in early-stage CRC TME scRNA-seq dataset ($n$ = 222 fibroblasts from 5 patients; boxes: interquartile range; black bars: median, whiskers: 1.5x interquartile range). Illustration in **a** reproduced from NIH BioArt (https://bioart.niaid.nih.gov/bioart/5).

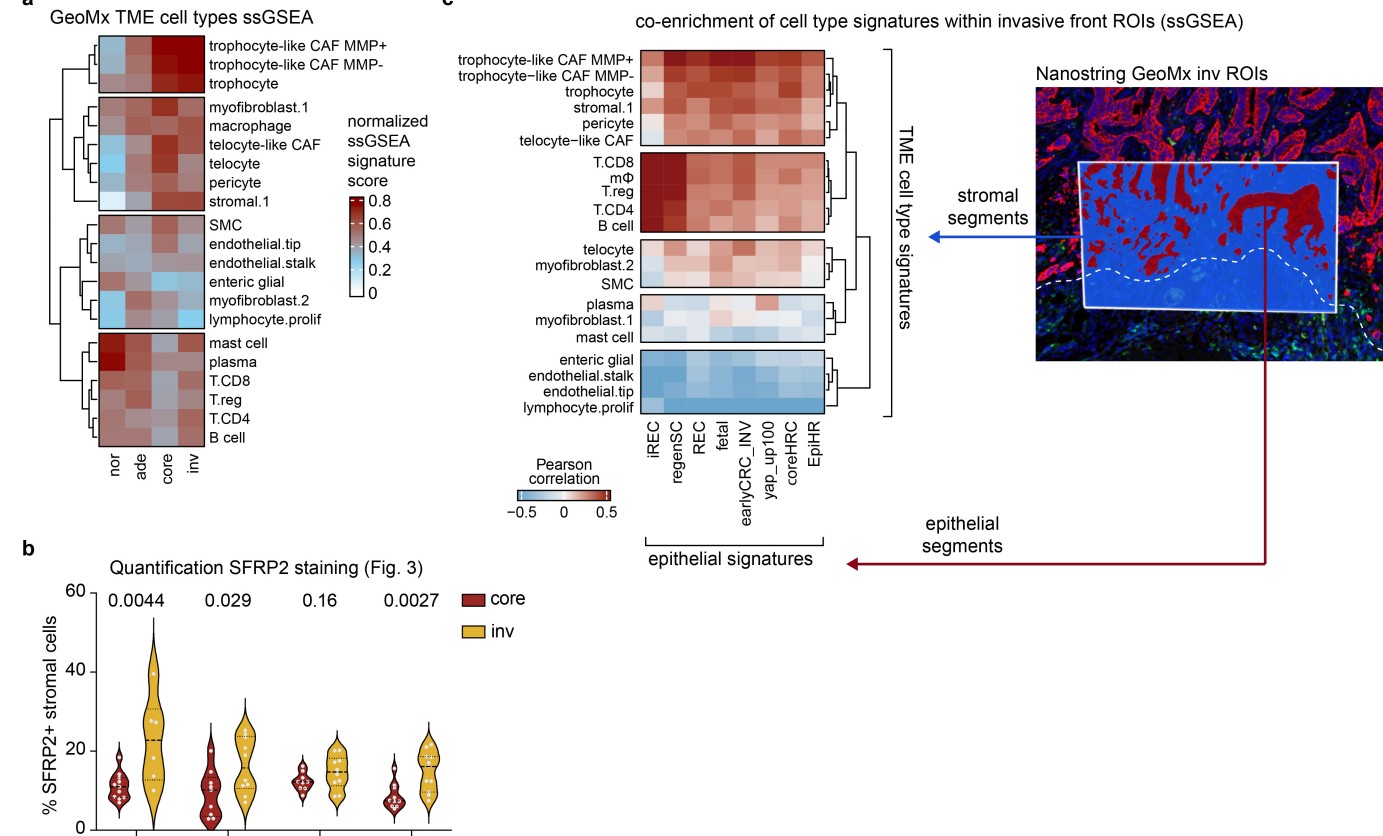

**a** GeoMx TME cell types ssGSEA

**c** co-enrichment of cell type signatures within invasive front ROIs (ssGSEA)

Nanostring GeoMx inv ROIs

stromal segments

epithelial segments

**b** Quantification SFRP2 staining (Fig. 3)

**Extended Data Fig. 5 | TME cell type enrichment in bulk spatial transcriptomics dataset of early-stage CRC.** a) Heatmap showing normalized ssGSEA signature scores for early-stage CRC TME cell type signatures (in rows) in histopathological regions (nor: normal, ade: adenoma, core: tumor core, inv: invasive front) analyzed in Nanostring GeoMx bulk spatial transcriptomics (WTA). For cell type signatures, the top 100 cell type-specific differentially expressed genes with the highest fold change were used (Supplementary Table 5). b) Violin plot showing distribution of SFRP2+ cells in 4 different

T1 CRCs. Percentage of SFRP2+ cells was calculated within the stromal compartment of the tumor core and the invasive front. Data points in the violin plot denote individual fields-of-view that were analyzed per indicated tumor region (red: tumor core; yellow: invasive front) (t-test). Relates to Fig. 3j. c) Heatmap depicting co-enrichment (Pearson correlation coefficient) of TME cell type signatures (within stromal segments) and epithelial cell signatures (within epithelial segments) in invasive front ROIs of WTA spatial transcriptomics dataset.

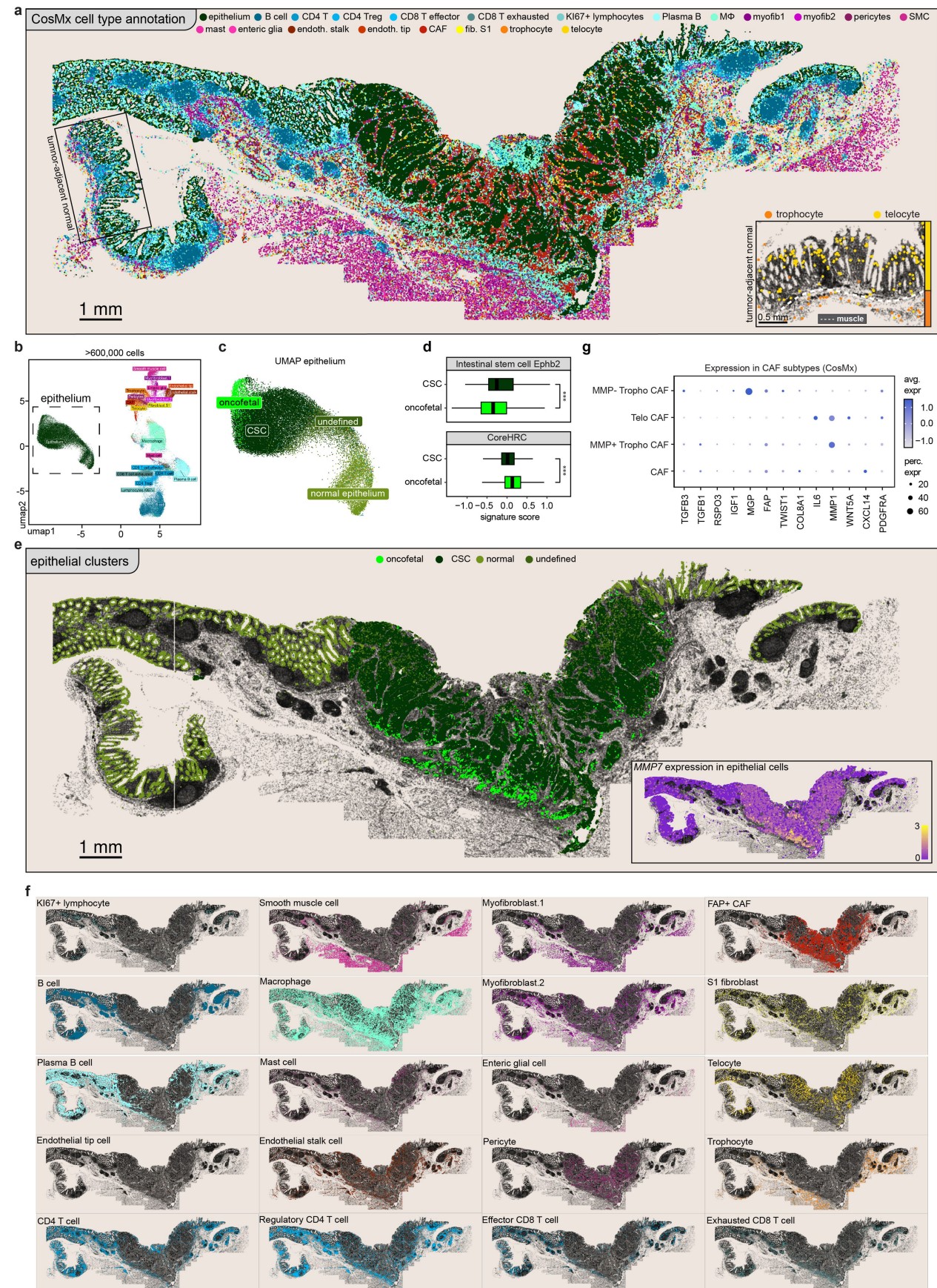

**Extended Data Fig. 6** | See next page for caption.

**Extended Data Fig. 6 | Single-cell spatial transcriptomics of early-stage CRC biobank specimen.** a) Single-cell spatial plot of tumor specimen (pt5; FFPE) profiled with CosMx single-cell spatial transcriptomics. Colors denote predicted cell types (prediction.score.max > 0.6, Methods) based on cell type clusters identified with scRNA-seq data of early-stage CRCs (reference dataset) (Fig. 3b). Inset shows stereotypical zonation of trophocytes and telocytes in indicated tumor-adjacent normal tissue (dashed line indicates mucosa/submucosa border). b) UMAP of CosMx-profiled cells annotated by predicted cell types. c) Subclustering of epithelial CosMx cells (from **b**) into oncofetal (high expressors of High Relapse Cells signature; $n = 5,078$), cancer stem cell (CSC; $n = 119,019$), undefined ($n = 194,491$) and normal epithelium ($n = 41,221$).

d) Boxplot of WNT/CSC[29] and coreHRC[14] signature scores among cells from CSC ($n = 119,019$) and oncofetal ($n = 5,078$) cell clusters shown in Fig. 3l and Extended Data Fig. 6c and e (boxes: interquartile range; black bars: median, whiskers: 1.5x interquartile range; *** $P < 0.001$, t-test). e) Single-cell spatial plot of epithelial subclusters from **c**. Inset shows relative expression (color gradient) of oncofetal marker *MMP7* at the invasive front (Fig. 1f). f) Single-cell spatial plots per indicated predicted cell type. g) Single-cell dot plot visualizing gene expression levels of CAF subtype marker genes across CAF subtypes identified in CosMx dataset. Colors indicate relative expression level. Dot size represents percentage of cells in which transcript was identified.

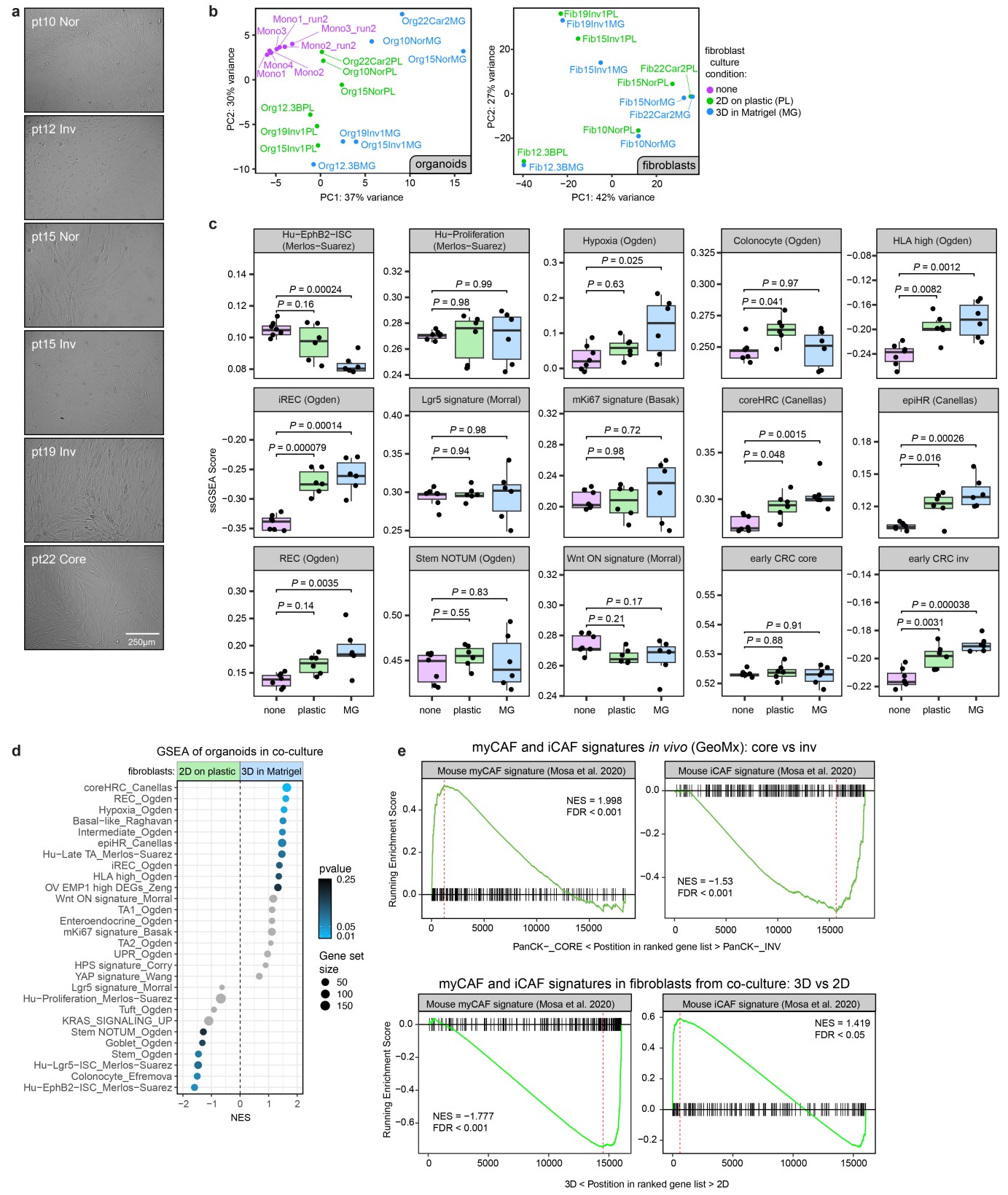

**Extended Data Fig. 7** | See next page for caption.

**Extended Data Fig. 7 | RNA sequencing of early-stage CRC organoid-fibroblast co-cultures.** a) Representative brightfield images of fibroblast lines derived from indicated patients and regions. Nor: normal, Inv: invasive front. b) Principal Component analysis of RNA-sequencing co-culture experiment. Left: organoid transcriptomes (pt5 invasive front organoids), right: fibroblast transcriptomes. Samples are colored by co-culture group: organoid monoculture (purple), co-culture with fibroblasts on plastic (green), and co-culture with fibroblasts in Matrigel (blue). Fibroblast annotation: 'Fib', followed by biobank patient number, and culture substrate (plastic (PL) or Matrigel (MG)). Principal component (PC) 1 and 2 are shown on x- and y-axis, respectively. c) Boxplot showing single-sample gene set enrichment analysis (ssGSEA) of various regenerative, metastatic, proliferative and stem cell signatures in organoid RNA-sequencing samples (none: $n = 7$; plastic: $n = 6$; MG: $n = 6$) grouped by co-culture condition (none: organoid monoculture; plastic: co-cultured with fibroblasts on plastic; and MG: co-cultured with fibroblasts in Matrigel). Boxes represent interquartile range, and black bars represent median values of all segments of indicated anatomical regions (ANOVA with Tukey's HSD, p-values are indicated). d) Dot plot depicting GSEA (permutation test with FDR) of epithelial regenerative, metastatic, proliferative and stem cell signatures in organoid RNA-sequencing samples from co-cultures with fibroblasts on plastic ($n = 6$) versus Matrigel (MG; $n = 6$). e) GSEA of myCAF and iCAF[50] signatures in vivo (WTA GeoMX: tumor core stromal segments versus invasive front stromal segments) and in vitro (3D co-culture (fibroblasts in Matrigel; $n = 6$) versus 2D co-culture (fibroblasts on plastic; $n = 6$)). Normalized enrichment score (NES) and false discovery rate (FDR) are shown as insets in each plot.

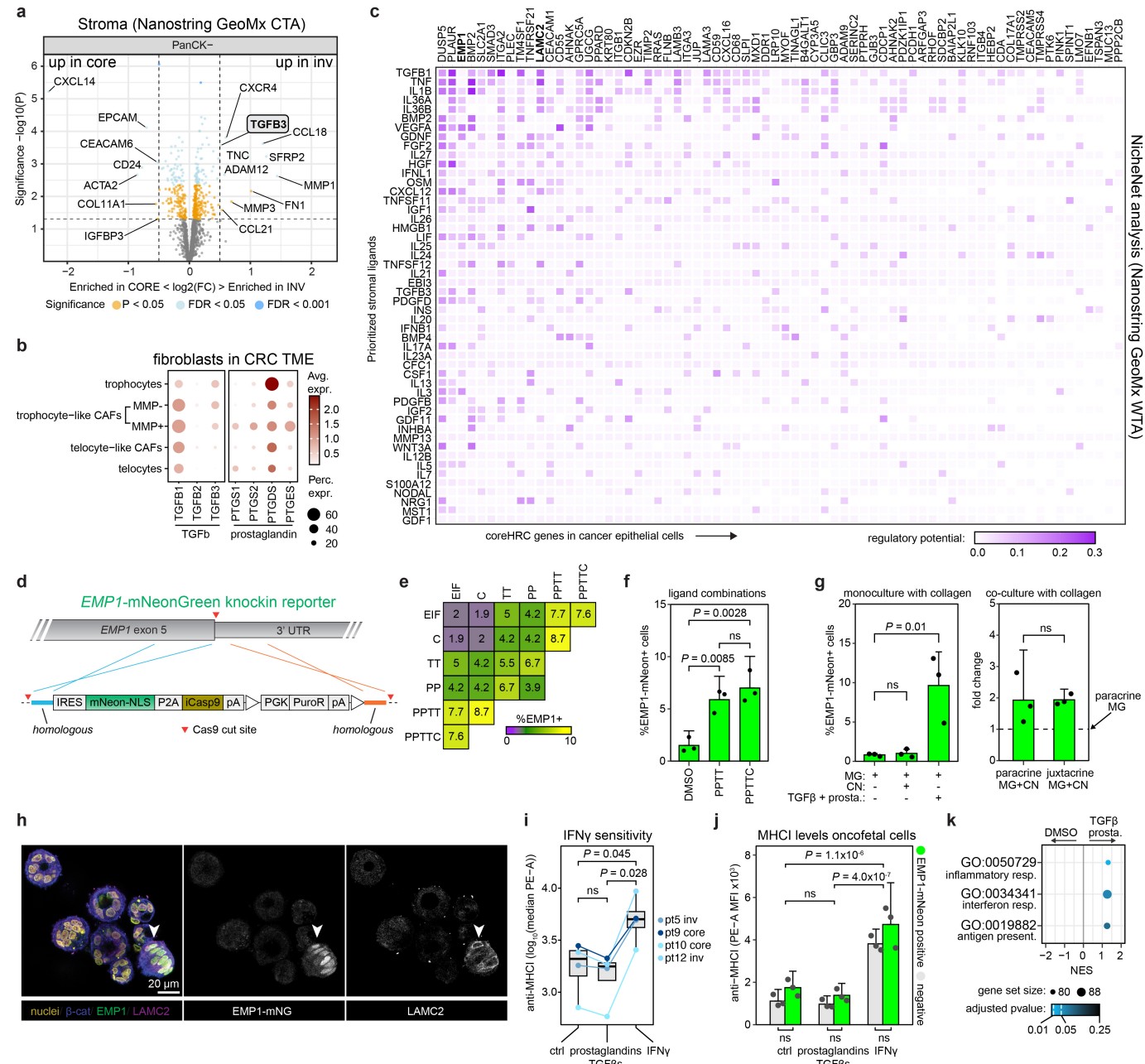

**Extended Data Fig. 8 | Organoid-based screening for inducers of oncofetal plasticity.** a) Differential expression analysis of invasive front (*n* = 40) versus tumor core (*n* = 38) stromal compartment (CD45-PanCK-) of CTA GeoMx bulk spatial transcriptomics dataset. Point color denotes significance level (Wald test, FDR). b) Relative expression (colors) of TGFβ- and prostaglandin-related genes in fibroblast subtypes (integrated CRC TME scRNA-seq). Dot size: percentage of cells expressing transcript. c) NicheNet heatmap showing top 50 ligands with highest predicted regulatory potential of coreHRC program (WTA dataset; *n* = 43 epithelial invasive front segments – receivers; *n* = 43 stromal invasive front segments – senders). Color gradient depicts regulatory potential. d) In-trans paired CRISPR knock-in strategy to label EMP1+ cells with nuclear mNeonGreen. IRES: internal ribosomal entry site, NLS: nuclear localization signal, pA: polyA, PuroR: Puromycin Resistance cassette. e) Heatmap showing induction of EMP1-mNeon+ cells using combinations of potent ligands identified in Fig. 4g,h. Color gradient EMP1+ cells (%) as measured by flow cytometry following 24 h induction. EIF: EGF + IGF1 + FGF2; C: CXCL12; TT: TGFβ1 + TGFβ3; PP: PGE2 + PGD2. f) EMP1-mNeon+ cells (%) upon most potent stimuli combinations in **e**. Data are presented as mean values + SD (*n* = 3, ANOVA, Tukey HSD). g) Left: EMP1-mNeon+ cells (%) in Matrigel (MG), 25%

Matrigel-25% collagen (CN) mixture, and upon treatment with TGFβ/ prostaglandins following 2 days exposure (*n* = 3, ANOVA *P* = 0.008, TukeyHSD). Right: EMP1-mNeon+ cells (%) in 25% Matrigel-25% collagen (CN) mixture in paracrine and juxtacrine cocultures. Dotted line represents indirect paracrine transwell coculture with organoids and fibroblasts in Matrigel (as in Fig. 4a, 3D co-culture; *n* = 3, t-test). Data are presented as mean values + SD h) Representative immunofluorescence for LAMC2 (purple) in EMP1-mNeonGreen-nls (green) knock-in organoids treated with PGD2, PGE2, TGFβ1, and TGFβ3. Cell membranes were visualized with β-catenin (β -cat, blue) staining. White arrow: organoid with high expression of LAMC2 and EMP1. i) MHCI levels in 4 patient-derived organoid lines upon treatment with TGFβ1 + 3 and prostaglandins (PGE2 + PGD2) or IFNγ (boxes: interquartile range; black bars: median, whiskers: 1.5x interquartile range; ANOVA, Tukey HSD). j) MHCI levels in pt5 inv EMP1-mNeon KI organoids upon treatment with TGFβ1 + 3 and prostaglandins (PGE2 + PGD2) or IFNγ. Data are presented as mean values + SD (*n* = 3, ANOVA, Tukey HSD). k) GSEA of immune-related GO biological processes (featured in Extended Data Fig. 1l) in RNA-seq of organoids treated with TGFβ and prostaglandins (PGD2 & PGE2). TGF-β: TGF-β1 & TGF-β3) (*n* = 2 patients; 3 replicates; permutation-based test, Benjamini-Hochberg corrected).

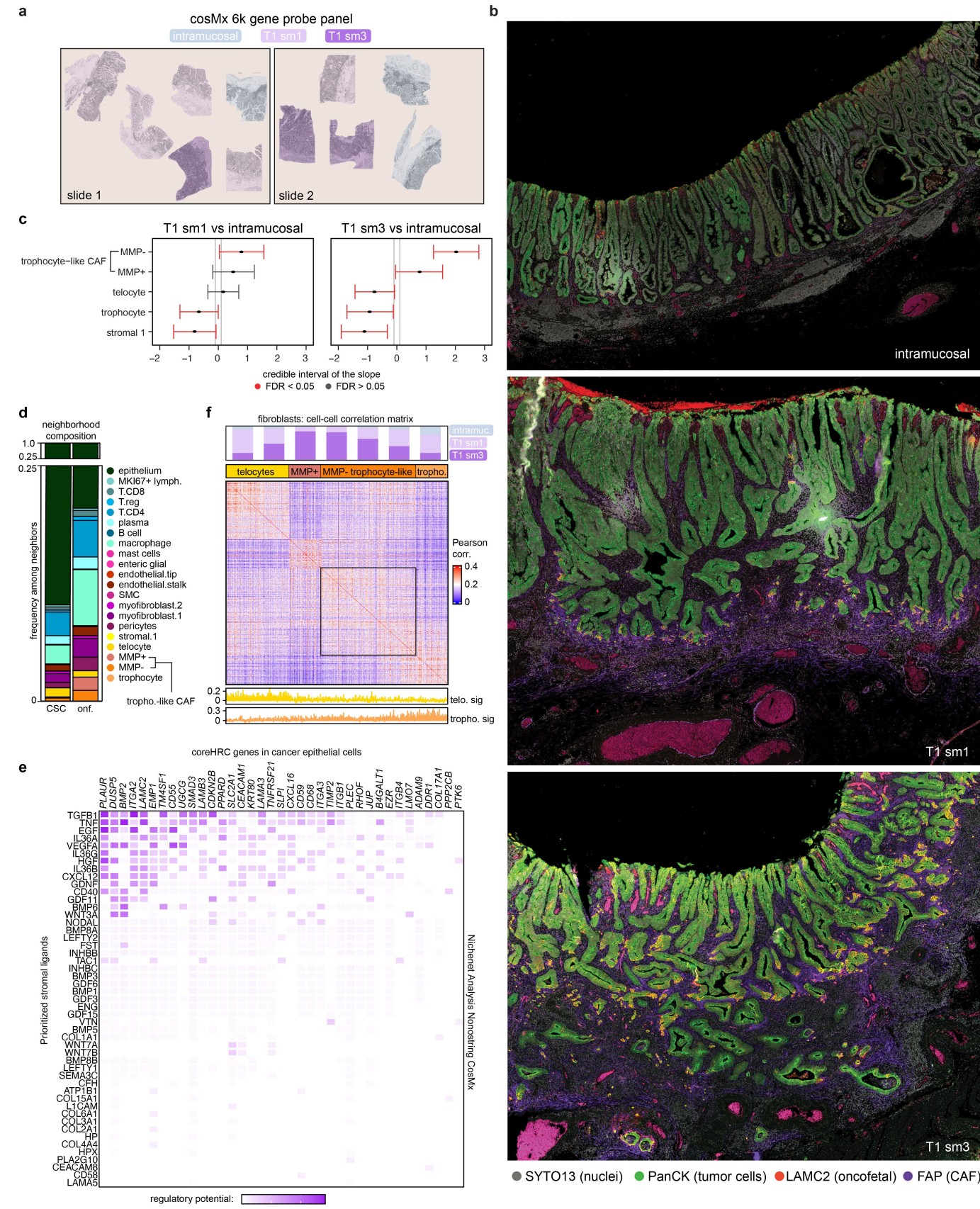

**Extended Data Fig. 9** | See next page for caption.

**Extended Data Fig. 9 | Single-cell spatial transcriptomics of early-stage CRC specimens temporally surrounding the onset of invasive tumor growth.** a) 11 tumor specimens distributed over 2 slides used for single cell spatial transcriptomics (CosMx, 6k gene probe panel). Colors indicate substage: intramucosal carcinoma (blue, $n = 3$); T1 sm1 (pink, $n = 5$) and T1 sm3 (purple, $n = 3$). b) Overview images of immunofluorescence of tumor specimens temporally surrounding the moment of malignant transformation, including an intramucosal carcinoma (top), a T1 sm1 (center) and T1 sm3 (bottom). Specimens were stained for SYTO13 (nuclei, gray), PanCK (green, epithelium), LAMC2 (red, regenerative tumor cells) and FAP (cancer-associated fibroblasts, purple). Regenerative tumor cells are yellow (PanCK+LAMC2 + ). c) Statistical analysis of fibroblast proportion shifts between T1 sm1 ($n = 5$) vs intramucosal ($n = 3$) and T1 sm3 ($n = 3$) vs intramucosal ($n = 3$). Whiskers denote 95% credible interval and are colored by significance as indicated. d) Barplot depicting cell type frequencies within the direct neighborhood (50 μm radius) of cancer stem cell (CSC) and oncofetal (onf, high expressors of High Relapse Cell signature) tumor cells. e) NicheNet heatmap showing top 50 stromal ligands with highest predicted regulatory potential of the coreHRC genes (Nanostring CosMx dataset, invasive front niche, $n = 11$ tumors). Color gradient depicts regulatory potential. f) Heatmap showing cell-cell Pearson correlations of fibroblasts of single-cell spatial transcriptomics (100 randomly selected fibroblasts per subgroup (hierarchical clustering) are shown). Bargraph at the top of the heatmap indicates proportion of cells in each cluster originating from intramucosal (intramuc.), T1 sm1 and T1 sm3 specimens. Trophocyte and telocyte signature ssGSEA scores are shown at the bottom of the heatmap. Black box annotates three subgroups of MMP- trophocyte-like CAFs that are intermediary to trophocytes and MMP+ trophocyte-like CAFs.

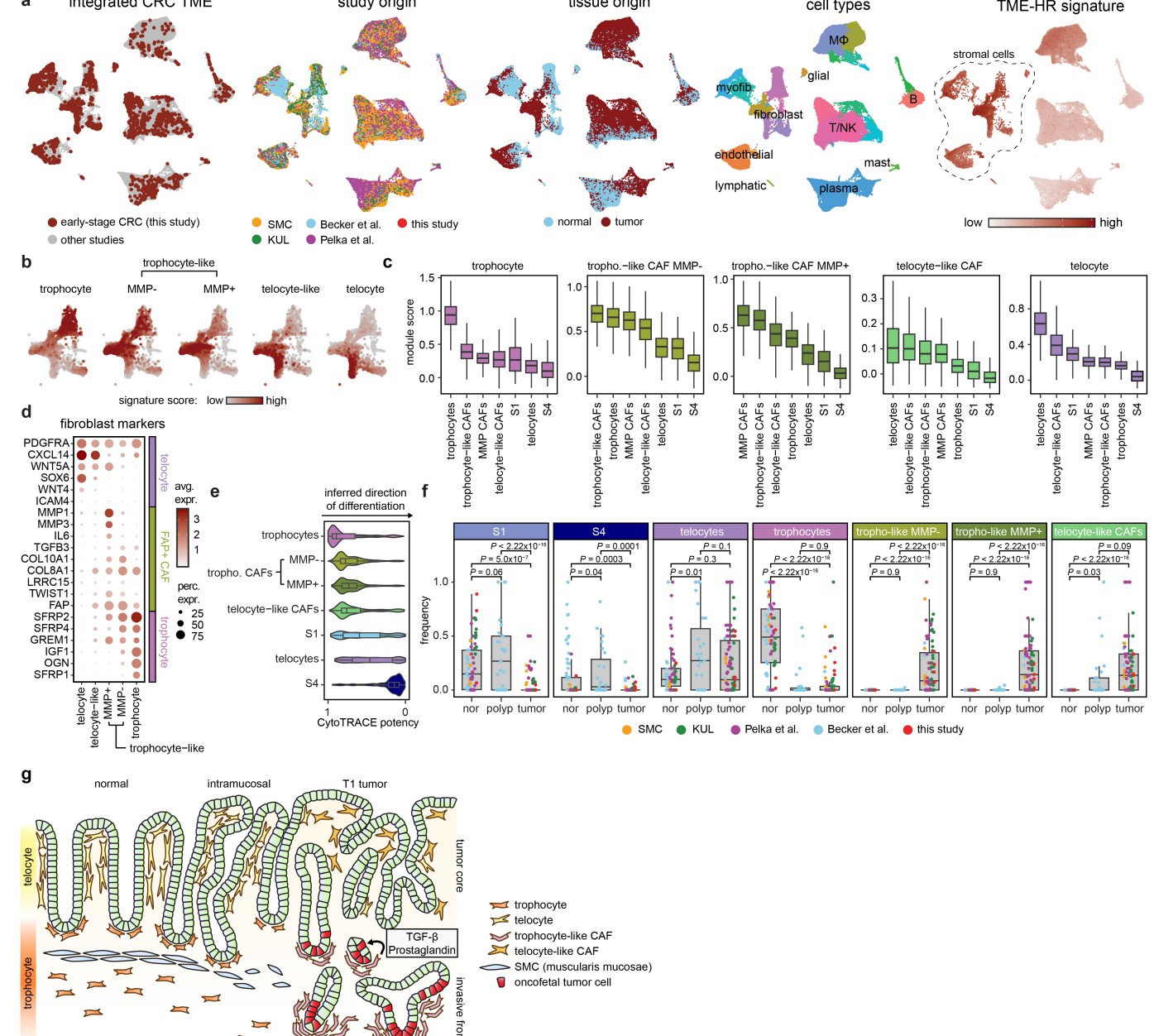

**Extended Data Fig. 10 | Integrated scRNA-seq dataset of CRC tumor microenvironment.** a) UMAP of integrated CRC TME scRNA-seq datasets containing data from Lee et al.[45] (GSE144735 and GSE132465), Becker et al.[55] (GSE201349), Pelka et al.[54] (GSE178341) and this study, overlayed with color coding highlighting cells from early-stage CRC (this study), indicating original study, tissue, cell type clusters, tumor stage and TME-High Relapse[14] signature score. b) Isolated fibroblast subcluster of integrated scRNA-seq UMAP of **a**, overlayed with fibroblast subtype signature scores. c) Fibroblast subtype signature score expression per fibroblast cluster of integrated CRC TME scRNA-seq dataset (*n* = 10,324 fibroblasts; boxes: interquartile range; black bars: median, whiskers: 1.5x interquartile range). d) Single-cell dot plot visualizing gene expression levels of CAF subtype markers in the integrated scRNA-seq dataset. Colors indicate expression level. Dot size represents

percentage of cells in which indicated transcript was detected. e) Violin plot depicting CytoTRACE potency scores per fibroblast subcluster (*n* = 10,324 fibroblasts; boxes: interquartile range; bars: median, whiskers: 1.5x interquartile range). Potency score represents developmental potential/stemness. High scores are attributed to less differentiated cells with capacity to develop into other cell types. f) Boxplot showing frequency of each fibroblast subtype in different CRC disease stages. Individual data points represent tumor samples (*n* = 147) colored by study (boxes: interquartile range; black bars: median, whiskers: 1.5x interquartile range; ANOVA, Tukey HSD). Note that cancer-associated fibroblasts are only found in tumors. g) Proposed model depicting emergence of trophocyte-like cancer-associated fibroblasts and oncofetal tumor cell states during earliest stages of invasive front formation in colorectal cancer.

# Reporting Summary

## Statistics

For all statistical analyses, confirm that the following items are present in the figure legend, table legend, main text, or Methods section.

| n/a | Confirmed | |
|---|---|---|
| ☐ | ☒ | The exact sample size (*n*) for each experimental group/condition, given as a discrete number and unit of measurement |
| ☐ | ☒ | A statement on whether measurements were taken from distinct samples or whether the same sample was measured repeatedly |
| ☐ | ☒ | The statistical test(s) used AND whether they are one- or two-sided<br>*Only common tests should be described solely by name; describe more complex techniques in the Methods section.* |
| ☒ | ☐ | A description of all covariates tested |
| ☐ | ☒ | A description of any assumptions or corrections, such as tests of normality and adjustment for multiple comparisons |
| ☐ | ☒ | A full description of the statistical parameters including central tendency (e.g. means) or other basic estimates (e.g. regression coefficient) AND variation (e.g. standard deviation) or associated estimates of uncertainty (e.g. confidence intervals) |
| ☐ | ☒ | For null hypothesis testing, the test statistic (e.g. *F*, *t*, *r*) with confidence intervals, effect sizes, degrees of freedom and *P* value noted<br>*Give P values as exact values whenever suitable.* |
| ☒ | ☐ | For Bayesian analysis, information on the choice of priors and Markov chain Monte Carlo settings |
| ☒ | ☐ | For hierarchical and complex designs, identification of the appropriate level for tests and full reporting of outcomes |
| ☐ | ☒ | Estimates of effect sizes (e.g. Cohen's *d*, Pearson's *r*), indicating how they were calculated |

*Our web collection on statistics for biologists contains articles on many of the points above.*

## Software and code

Policy information about availability of computer code

Data collection
- FACS Celesta (for EMP1-mNG organoid quantifications): BD FACS DIVA v9.2
- FACSaria III (for sorting cells for scRNA-seq): BD FACS DIVA v.8.0.1
- Microscopy Leica SP8 (for imaging EMP1-mNG organoids): LAS X v3.5.7.23225
- qPCR: CFX Manager Version 3.1.1517.0823.

Data analysis
Nanostring GeoMx:
- Fgsea v1.24.0
- ssGSEA (implemented in GSVA v1.46.0)
- CMScaller v2.0.1
- inferCNV v1.14.2
- NicheNet v2.0.0
- VariancePartition v1.38.1

WGS:
- oncoanalyser v1.0.0
- HaplotypeCaller v4.1.3
- SMuRF v3.02
- ape v5.8

scRNA-seq:

• nf-core scrnaseq pipeline v2.4.0
• STARsolo v2.7.10b
• STARsolo v2.7.10b
• Seurat v5.0.1
• Monocle3 v1.4.26
• CytoTRACE2 v1.1.0
• Slingshot v2.16.0
• CellRank v2.0.7

Nanostring CosMx:
• Seurat v5.0.1
• Sctransform

Bulk RNA-seq
• nf-core RNAseq pipeline v3.14.0
• FastQC v0.12.1
• Trim Galore! v0.6.7
• STAR v2.7.9a
• Salmon v1.10.1
• DESeq2 v1.38.3
• clusterProfiler (4.8.3)

Flow cytometry:
• Floreada v8/6/24
• DIVA version v9.2

Microscopy:
• OrganoSeg v1
• Stardist v0.3.0
• FIJI ImageJ 2.14.0/1.54f
• QuPath v0.6.0

General:
• Rstudio 2024.12.1+563
• R v4.2.0
• Excel 16.95
• Prism 10.4.1
• ggplot2 v3.5.1
• ggpubr v0.6.0

For manuscripts utilizing custom algorithms or software that are central to the research but not yet described in published literature, software must be made available to editors and reviewers. We strongly encourage code deposition in a community repository (e.g. GitHub). See the Nature Portfolio guidelines for submitting code & software for further information.

## Data

Policy information about availability of data

All manuscripts must include a data availability statement. This statement should provide the following information, where applicable:
- Accession codes, unique identifiers, or web links for publicly available datasets
- A description of any restrictions on data availability
- For clinical datasets or third party data, please ensure that the statement adheres to our policy

Whole genome sequencing data of patient-derived organoids (EGAD50000002204), RNA sequencing data of organoids and organoid-fibroblast co-cultures (EGAD50000002202) and scRNA-sequencing data of early-stage colorectal cancers (EGAD50000002203) are available through the European Genome-Phenome Archive (EGA) under accession number EGAS50000001532. Bulk (Nanostring GeoMx) and single-cell (Nanostring CosMx) spatial transcriptomic data of T1 colorectal cancers and processed expression data (RNA-seq of organoids and organoid-fibroblast co-cultures and scRNA-seq of early-stage colorectal cancer biopsies) are available at Zenodo (https://doi.org/10.5281/zenodo.17671259). Expression data (scRNA-seq, Nanostring GeoMx CTA, Nanostring CosMx Fig.3) can be accessed through an interactive dashboard (https://snippertlab.nl/resources). Published scRNA-seq data of human colorectal cancers were obtained from GSE144735 and GSE13246545; GSE20134955; and GSE17834154.

## Research involving human participants, their data, or biological material

Policy information about studies with human participants or human data. See also policy information about sex, gender (identity/presentation), and sexual orientation and race, ethnicity and racism.

| Reporting on sex and gender | Information on sex is reported in Supplementary Table S1. Out of the 16 patients included for organoid derivation, 8 (50%) were male and 8 (50%) were female. All patients gave informed consent to share this information. |
|---|---|
| Reporting on race, ethnicity, or other socially relevant groupings | Information on race and ethnicity has not been collected |

| Population characteristics | Relevant clinical information for the patients is listed in Supplementary Table 1. The organoid/fibroblast biobank consists of 16 early-stage colorectal cancer patients. Patients were aged between 60 and 81. Patients were not pre-treated. For GeoMx, 10 T1 colorectal cancers (5x with lymph node metastases and 5x without lymph node metastases) were analyzed using the GeoMx CTA (Cancer Transcriptome Atlas) panel and 9 T1 colorectal cancers (3x without metastases, 3x with lymph node metastases, and 3x with distant metastases) were analyzed using the GeoMx WTA (Whole Transcriptome Atlas) panel. For the tumor specimens analyzed by CTA, samples were selected such that risk factors, including lymphovasclular invasion and tumor budding and location and morphology were similar between metastatic and non-metastatic primary tumors within the CTA cohort. For Nanostring CosMx 1 T1 patient from the organoid biobank (pt5) was selected for single-cell spatial transcriptomics of a complete tumor cross section. Additionally, 11 CRC specimens capturing the moment of malignant transformation at the start of invasive tumor growth (3x tumor in situ, 5x T1 sm1 and 3x T1 sm3) were selected for Nanostring CosMx and sectioned to fit on 2 slides. |
|---|---|
| Recruitment | The biobank participants were 16 patients suspected of early-stage colorectal cancer who underwent surgery for removal of the primary tumor, instead of endoscopic removal due to inaccessibility of the tumor. This could skew the sampled tumors towards tumors that are difficult to access by endoscopy. Patients were recruited by the Utrecht Platform for Organoid Technology (UPORT) (https://uport.umcutrecht.nl/researcher/en/). |
| Ethics oversight | This study was approved by the University Medical Centre (UMC) Utrecht ethical committee |

Note that full information on the approval of the study protocol must also be provided in the manuscript.

# Field-specific reporting

Please select the one below that is the best fit for your research. If you are not sure, read the appropriate sections before making your selection.

☒ Life sciences          ☐ Behavioural & social sciences          ☐ Ecological, evolutionary & environmental sciences

For a reference copy of the document with all sections, see nature.com/documents/nr-reporting-summary-flat.pdf

# Life sciences study design

All studies must disclose on these points even when the disclosure is negative.

| Sample size | There was no statistical calculation of sample size upfront. To establish a biobank representative of early-stage CRC, we aimed to include all eligible patients during the study period, targeting a minimum of 10 participants. Ultimately, 16 patients were enrolled in the biobank. For the Nanostring GeoMx and CosMX, sample size (19 and 11 patients, respectively) was determined based on previous knowledge of the sequencing facility on experimental variation. Sample sizes of experiments are indicated throughout the manuscript. |
|---|---|
| Data exclusions | In scRNA-seq, epithelial cells were excluded due to low sequencing quality (based on the total number of transcripts per cell and mitochondrial transcripts per cell). In Nanostring GeoMx, 1 patient was excluded because it was classified as stage T3, whereas all other patients were stage T1. |
| Replication | All experiments were performed in multiple independent replicates, as described in the figure legends and Methods section. |
| Randomization | Experimental groups for the in vivo analyses were not based on randomization, but on pathological classification of normal, adenoma, tumor core, and invasive front. Organoids and fibroblasts used in the in vitro experiments were randomly assigned to an experimental group. |
| Blinding | The researchers were blinded to patient identities. In the experiments, no blinding of researchers for organoid identities or treatments was performed, to avoid sample swaps and as is according to common practice in the field. |

# Reporting for specific materials, systems and methods

We require information from authors about some types of materials, experimental systems and methods used in many studies. Here, indicate whether each material, system or method listed is relevant to your study. If you are not sure if a list item applies to your research, read the appropriate section before selecting a response.

## Materials & experimental systems

| n/a | Involved in the study |
|---|---|
| ☐ | ☒ Antibodies |
| ☐ | ☒ Eukaryotic cell lines |
| ☒ | ☐ Palaeontology and archaeology |
| ☒ | ☐ Animals and other organisms |
| ☒ | ☐ Clinical data |
| ☒ | ☐ Dual use research of concern |
| ☒ | ☐ Plants |

## Methods

| n/a | Involved in the study |
|---|---|
| ☒ | ☐ ChIP-seq |
| ☐ | ☒ Flow cytometry |
| ☒ | ☐ MRI-based neuroimaging |

# Antibodies

| | |
|---|---|
| Antibodies used | The following antibodies were used for immunohistochemistry: SFRP2 (PA5-29390, Invitrogen, 1:200), LAMC2 (AMAb91098, Atlas Antibodies, 1:500), PanCK (AlexaFluor 532 conjugated; NBP2-33200 Novus 1:500, and NBP3-08398 Novus 1:300), DNA SytoTM 13 (S7575, Invitrogen, 1:10k), Alexa 594 anti-rabbit (Invitrogen A11037; 2 μg/ml) and Alexa 594 anti-mouse (Invitrogen A11032; 2 μg/ml)<br><br>The following antibodies were used for Nanostring: Pan-Cytokeratin (PanCK, Novus Biologicals NBP2-33200AF532, 1 μg/ml) and CD45 (Novus Biologicals NBP2-34528AF594, 5 μg/ml) and DNA SytoTM 13 (S7575, Invitrogen, 500 nM).<br><br>The following antibodies were used for tissue staining for scRNA-seq: PE anti-human CD326 (EpCAM) (324205 9C4, Biolegend, 1:200) and FITC anti-CD45 (368507 2D1, Biolegend, 1:200)<br><br>The following antibodies were used for staining of organoids: PE anti-human HLA A/B/C (311405 W6/32, Biolegend, 1:400), PE anti-human CD326 (EpCAM) (324205 9C4, Biolegend, 1:400); beta-catenin (C2206, Sigma-Aldrich, 1:500), LAMC2 (AMAb91098, Atlas Antibodies, 1:500)., 647 anti-mouse (Invitrogen A21236; 1:500) and Alexa 568 anti-rabbit (Invitrogen A11011; 1:1,000). |
| Validation | All antibodies were validated for species reactivity and application by the manufacturer, as specified below:<br><br>SFRP2: verified for human immunohistochemistry and immunofluorescence<br>LAMC2: verified for human immunohistochemistry and immunofluorescence<br>PanCK: verified for human immunohistochemistry and immunofluorescence<br>CD45 (Novus Biologicals): verified for human immunohistochemistry<br>EpCAM: verified for human flow cytometry<br>CD45 (Biolegend): verified for human flow cytometry<br>HLA: verified for human flow cytometry<br>beta-catenin: verified for human immunofluorescence<br><br>Additional validation was performed by confirming the expected localization and expression patterns. |

# Eukaryotic cell lines

Policy information about cell lines and Sex and Gender in Research

| | |
|---|---|
| Cell line source(s) | All cell lines (organoids and fibroblasts) used were generated in this study as described in the methods. Patient information is listed in Supplementary Table 1. |
| Authentication | Organoids were subjected to whole-genome sequencing and functionally tested in growth factor dependency screens. This way, the most used organoids could be authenticated based on their specific mutation in driver genes (TP53, KRAS) and on their growth factor dependency. Additionally, organoid morphology (see Extended Data Fig. 3) was continuously monitored to prevent sample swaps. |
| Mycoplasma contamination | Cell lines were routinely tested for mycoplasma and results were always negative. |
| Commonly misidentified lines (See ICLAC register) | None |

# Plants

| | |
|---|---|
| Seed stocks | *Report on the source of all seed stocks or other plant material used. If applicable, state the seed stock centre and catalogue number. If plant specimens were collected from the field, describe the collection location, date and sampling procedures.* |
| Novel plant genotypes | *Describe the methods by which all novel plant genotypes were produced. This includes those generated by transgenic approaches, gene editing, chemical/radiation-based mutagenesis and hybridization. For transgenic lines, describe the transformation method, the number of independent lines analyzed and the generation upon which experiments were performed. For gene-edited lines, describe the editor used, the endogenous sequence targeted for editing, the targeting guide RNA sequence (if applicable) and how the editor was applied.* |
| Authentication | *Describe any authentication procedures for each seed stock used or novel genotype generated. Describe any experiments used to assess the effect of a mutation and, where applicable, how potential secondary effects (e.g. second site T-DNA insertions, mosiacism, off-target gene editing) were examined.* |

# Flow Cytometry

## Plots

Confirm that:

☒ The axis labels state the marker and fluorochrome used (e.g. CD4-FITC).

☒ The axis scales are clearly visible. Include numbers along axes only for bottom left plot of group (a 'group' is an analysis of identical markers).

☒ All plots are contour plots with outliers or pseudocolor plots.

☒ A numerical value for number of cells or percentage (with statistics) is provided.

## Methodology

| | |
|---|---|
| Sample preparation | Single-cell organoid suspensions were prepared by trypsinization with Trypsin-EDTA for 5 min at 37 °C.<br><br>Single-cell tissue suspensions used for scRNA-seq were prepared by mincing punch biopsies with scissors and subjecting them to enzymatic digestion at 37 °C for 15-25 min with 1 mg/ml collagenase and 1 mg/ml Dispase II. The resulting tissue fragments were washed 3 times by means of centrifugation, resuspended in Recovery Medium and then cryo-preserved. Later, tissue fragments were thawed, washed with basal medium, and trypsinized to single cell suspensions using TrypLE supplemented with 10 µM Y-27632 for 5 min at 37 °C. To distinguish epithelial, immune and stromal cell populations and sort equal amounts of these 3 populations, single cell suspensions were stained with DRAQ7, PE anti-human CD326 (EpCAM) and FITC anti-CD45 in advanced DMEM/F12 for 30 min on ice. |
| Instrument | Organoids: BD FACSCelesta<br>Primary tissue for scRNA-seq: BD FACSAria III |
| Software | Data collection:<br>• FACS Celesta: BD FACS DIVA v9.2<br>• FACSAria III: BD FACS DIVA v.8.0.1<br><br>Data analysis:<br>BD FACS Diva v9.2 and Floreada v8/6/24 |
| Cell population abundance | The CD45-positive and CD45-negative cell populations were subjected to scRNA-seq after sorting, which confirmed their identities as respectively immune cells and non-immune stroma. Moreover, the index sorting allowed us to confirm the expression of CD45-PE in the immune cell clusters, as shown in Extended Data Fig. 5c |
| Gating strategy | Organoids: SSC-A/FSC-A was used to select cells, FSC-H/FSC-A was used to select singlets. DAPI staining was used to mark dying cells. Single live cells (DAPI-) were gated in the BV421-A channel and organoid cells were separated from fibroblasts based on EpCAM-PE measured in the PE-A channel. mNeon fluorescence was measured in the FITC-A channel. Gates were set based on negative control samples.<br><br>Primary tissue for scRNA-seq: SSC-A/FSC-A was used to select cells, FSC-H/FSC-A was used to select singlets. DRAQ7 staining was used to mark dying cells, EpCam for epithelial cells, and CD45 for immune cells. Gates for all stainings were based on negative controls. |

☒ Tick this box to confirm that a figure exemplifying the gating strategy is provided in the Supplementary Information.

