## [Peer Review File · Nature]

Emergence of oncofetal plasticity is ubiquitous in early colorectal cancers

Corresponding Author: Dr Hugo Snippert

Version 0:

Reviewer comments:

Referee #1

(Remarks to the Author)

Through multi-region sampling of early-stage tumours, Buissant des Amorie et al., report an interesting study describing the early emergence of revival/regenerative/oncofetal stem cells at the invasive front of primary CRC tumours. The authors show how revival stem cell plasticity is likely non-genetic and can be driven by stromal fibroblasts in the tumour microenvironment. Stem cell plasticity has recently emerged as a hallmark of CRC and is under intense investigation by many labs in the field.

The paper is well written and the data support the conclusions. My main issue is that most of the finding in the paper are already known to the field. For example:

1. Revival stem cells can be activated by fibroblasts
(Efremova et al, preprint, 2024, Ramos et al., Cell, 2023, Qin et al., Cell, 2023)
2. Revival stem cells are important for CRC metastasis
(Efremova et al, preprint, 2024, Moorman et al., Nature, 2024, Ganesh, Nature Cancer 2020)
3. TGF- β and PGE2 can drive revival stem cells
(Qin et al., Cell, 2023, Roulis et al., Nature 2020)
4. CRC plasticity is largely non-genetic
(Rehman et al., Cell, 2021, Househam et al., Nature, 2022)

Some of these important references are also missing from the paper. Many of these topics have also been covered in recent reviews on the topic (Fey et al., Cell Reports 2024 and Tape, Trends in Cancer 2024).

The finding that revival stem cell activation can be seen very early in human tumours is interesting, but even then, the role of YAP signalling (that controls access revival stem cells) in CRC initiation via mesenchymal cells is known (Roulis et al., Nature 2020).

The finding that spatial patterning of CAFs lead to the revival state at invasive fronts is novel to my knowledge. This is the title of the paper and the most interesting part of the manuscript.

While this paper representants a substantial body of excellent work, is very robust, and I enjoyed reading it, I am not sure if the finding that fibroblast heterogeneity can regulate revival stem cells justifies publication in Nature.

Additional points:

1. By calling cells at the invasive front 'High Relapse Cells' (HRCs), there is a general presumption that revival stem cells = metastatic cells. It is entirely possible, and arguably more likely, that most of the revival stem cells that are induced in stromal

niches stay in the primary tumour, with no metastatic dissemination. Given that Stage 1 and 2 patients very rarely relapse with metastatic disease, I think it is unfair to call LAMC2+ or EMP1+ cells in early-stage tumours 'High Relapse Cells'. In my opinion, these cells in primary tumours should be called revival / regenerative / oncofetal-like stem cells, not HRC which is really a name given to revival stem cells found in liver mets that can lead to disease relapse.

2. The indirect co-culture model cannot account for matrix induced effects that are known to be important for revival stem cell activation. Why didn't the authors use direct co-cultures?

3. What is driving the fibroblast heterogeneity? Is it stochastic or do signals from cancer cells alter fibroblast populations?

I applaud the excellent science but from a publication perspective feel the work has re-discovered what was already known. The novelty of the paper lies in the fibroblast heterogeneity angle which only comprises x1 Figure in the paper.

Referee #2

(Remarks to the Author)

des Amorie et al present an eloquent, well-written and high quality characterisation of a novel tissue & organoid cohort from T1 CRCs, combining detailed single cell & spatial interrogation with patient-matched regional-specific organoid experimental validation. The cohort itself is highly novel, as while most similar collections have been developed from advanced cancers, the approach presented here in T1 tumours & methodologies used for patient-matched multi-regional organoid isolation from fresh T1 surgical resections are beyond any other approaches this reviewer has seen before. The authors utilise this cohort initially for discovery and identify a number of the previously published (Ref 17 – EMP1/HRC, Nature 2022) biomarkers for aggressive epithelial phenotypes are elevated in almost all T1 invasive front samples used in this study, which contrary to the previous publications these are elevated regardless of T1 outcome (this cohort contains 7N&M-, 10N+, 3M+). The authors also identify a series of previously published fibroblast subtypes, most of which reminiscent of normal crypt/tissue locations or cancer-associated states, which they refine further using their T1 cohort data to detail spatial arrangement of these during the early cancer development stages. Utilising the patient-matched organoid bank, the authors show that there is an absence of mutational drivers of these EMP1/HRC changes in patient samples, but the EMP1/HRC phenotype can be elevated in all samples from all regions (all those tested here) using TGFb and/or PTGS2 (which themselves are identified from the fibroblast populations)

My main concerns here are that while the sample set are novel, and the analyses are certainly state-of-the-art, the results are largely confirmatory of a previously identified (seen more widely than just the EMP1 study) set of mechanisms underpinning HRC/regenerative epithelial states. The authors discuss precisely (291-299) the two key points that that could be addressed that would push this beyond what has been published, which will be expanded on below.

I would again reiterate that I fully agree that the bias in previous studies has been towards later stage or more established disease, and little has been done to understand early signs of metastatic competence in CRC; points which this current paper addresses. The suggested main points aim to address the mechanisms underpinning the switch during tumour development for the widespread expression of this phenotype regardless of outcome to the point where expression of this phenotype is strongly associated with outcome.

Main points:

1 - Line 99-100: related to my main point above, these findings around how stromal/inflammatory factors can drive a regenerative state have been well described before, both in the Cañellas-Socias study but in others like Vasquez et al 2022 <https://pubmed.ncbi.nlm.nih.gov/35931031/> or Qing et al Cell 2024 [https://www.cell.com/cell/fulltext/S0092-8674\(23\)01221-7](https://www.cell.com/cell/fulltext/S0092-8674(23)01221-7) or Zapatero et al Cell 2024 [https://www.cell.com/cell/fulltext/S0092-8674\(23\)01220-5](https://www.cell.com/cell/fulltext/S0092-8674(23)01220-5) – some of these studies even use TGFb additives or fibroblast co-cultures to induce the same phenotypes (many not cited here).

In the context of these previous studies, the main findings to this reviewer is the one that describes that in T1 tumours, these signals are almost uniformly elevated across the invasive front of all lesions regardless of the metastatic status. So, to this reviewer it would appear that the findings of HRC cells in T1s are not so much a metastatic phenotype but rather a phenotype of epithelial cells that have a location closer to the invasive front (Authors reiterate this in line 266-269). So overall, this HRC signal is not driving metastatic phenotypes in T1, and only in later disease; a point that the authors discuss how to address and action on.

In lines 209-212, the authors also discuss this point in relation to the fibroblast lineages that drive this phenotype, but state that these signals "are associated with metastatic HRC tumour states, high TME-HR signatures and poor patient outcome" – however, again, this only appears to hold up in later stage disease, as detailed in previous studies and to some extent in Fig3e of this study, but most importantly not in the T1 cohort itself.

So this to me is the critical point, and as the authors clearly describe in lines 291-299, requires either of two approaches to answer the question they have posed from the findings of this current study.

A - How and when do these signals stay switched on during tumour progression in tumours that maintain metastatic phenotypes and ultimately relapse/progress, compared to how are these features lost in tumours that do not relapse? This would require assessment of samples at along the different stage of development/progression for the fibroblast & epithelial phenotypes described here.

B – If certain critical bottlenecks can be identified above, detailing on which of these are able to interfere/block phase-specific tumour progression would be needed (perhaps using the same models as the original EMP1 study showed) and ablation of the metastatic phenotype signalling.

While the current study certainly adds to the recent data in this area, inclusion of these (or similar) suggested analyses above in early-to-late-stage tumour data alongside follow-up of findings with in vivo experiments described in A/B above would elevate this study well above the current understanding of disease.

2 - Lines 68-71: This sampling approach presented here is very similar to the Heiser et al 2023 study in Cell (<https://pubmed.ncbi.nlm.nih.gov/38065082/>; not cited in this current study), which performed phylogenetic tumour evolution tracking across these sample regions and identified an immune exclusion phenotype driven by extracellular matrix signal in MSS tumours (predominantly stage I & II in this MSS subset) using many of the same profiling methodologies used here in this current study. It would be most useful if the author assessed the pseudotemporal assessments, the IEX markers and the spatial CNV inferences in their T1 dataset too, similar to how the authors have looked to confirm the findings of the Cañellas-Socias et al EMP1/HRC study.

Similarly, the markers and biologies used in the recent Moorman et al study 2024 Nature <https://www.nature.com/articles/s41586-024-08150-0> could also be assessed here, to provide clarity as to the T1 cellular states compared to those metastatic phenotypes.

Minor points:

1- The LAMC2 staining shown in Figure 1h for this single case does look to be associated with the invasive front epithelial cells acquiring this phenotypic marker, however I would argue that the staining presented for the single case with EMP1 in Supp Fig 1h is less than convincing. Ideally the authors would have presented these as multiplexed stains, or at the very least on sequential sections from a wider range of samples.

Related to this point, the choice of EMP1 for the fluorescent reporter seems unwarranted, at least based on the data for EMP1 staining that is presented.

2- Can the authors include a clearer table (perhaps alongside S1B) that details colonic location of these lesions.

3 - From UMAP on Fig1d, the point that these are clustering/non-clustering is quite subjective; can the authors perform some objective (clustering/alignment or correlative like in Fig2i) approach to determine this similarity?

4 – the paracrine studies (lines ~228) are exquisitely performed and fully validate this hypothesis.

Referee #3

(Remarks to the Author)

In this manuscript, Buissant des Amorie, Hageman, Brunner and colleagues explore the molecular mechanisms associated with the acquisition of metastatic traits in colorectal cancer (CRC). Metastases are the main cause of CRC deaths. So far, no genetic driver has been clearly associated with their emergence, and their origin has largely remained elusive.

Here, the authors approach this topic from a peculiar point of view, focusing on the in-depth characterization of a cohort of patients with early-stage disease displaying minimal sign of tumor invasiveness, if any. They produce a remarkable dataset, performing multiregional imaging-based spatial transcriptomics. They sample four region-of-interest (ROIs) across whole resected lesions, including the tumor core, the invasive front, as well as adjacent regions displaying adenoma features, or overtly normal, in approx. 20 patients, with or without locoregional metastases (T1-N0 or T1-N1, according to TNM staging), and using two distinct large scale probe panels (targeting either 1800 cancer-related genes with 5 probes each or a whole-transcriptome set against 18000 transcripts). This analysis revealed stereotypical tumor patterning, including an invasive front transcriptional signature (e.g., EMP1, LAMC1, ANXA1) previously associated (but not causally related yet) with invasive tumor cells in mouse models and relapse in a large clinical cohort (PMID: 36352230), and generally linked with fetal intestinal development, and injury-repair at adult stages, especially in the mouse. They complement their initial analysis by generating a biobank of ROI-specific organoids, which utilize to systematically track ROI-specific genetic alterations or test ROI-specific growth factor dependencies, focusing on tumor core vs. invasive edge, and finding no evidence for significant cancer cell-intrinsic differences between these two sites. They turn their attention to the tumor microenvironment by performing a whole-slide imaging-based spatial transcriptomic experiment spanning an entire tumor lesion and its margins, coupled with a lineage-balanced single-cell RNA-seq experiment, to map cell identities at single cell resolution. Co-culture of some ROI-specific stromal-derived lines with organoids leads to transcriptional changes reminiscent of those observed at the invasive front and led to the prioritization of some stromal-derived ligands, which partially mediate such response (e.g., TGFB3, PGE2).

The manuscript has several strengths, especially related to:

1. Multiregional spatial transcriptomics on T1 lesions across a 20-patient cohort reveal that stereotypical tumor patterning is present from the early stages of tumor development, markedly anticipating observations previously associated to more

advanced disease

2. Relying on pathology-guided multiregional sampling (a challenging task not commonly achievable across cancer centers), the authors establish a ROI-specific organoid biobank and systematically test the hypothesis that tumor core- vs. invasive front-derived organoid may have distinct growth factor dependencies, without finding supporting evidence
3. More broadly, the application of complementary spatial transcriptomic approaches to clinical stages unfrequently explored in this field

Conversely, other sections of the manuscript appear to be considerably less developed:

1. The authors find evidence of tumor patterning, as previously observed at more advanced stages of disease, associating an invasive-front specific transcriptional program to metastatic competence at very early stages, as stated in the title. Yet, most T1 patients (approx. 90%) do not relapse after surgery. Disease progression is found only in a minority of T1 cases (approx. 10%). While the study cohort is balanced between T1N0 and T1N1, the authors provide no insights towards the identification of biomarkers or gene expression programs to set apart patients with or without pathological evidence of metastasis, likely due to the small number of patient analyzed. Thus, in the present form, the manuscript title does not appear to be supported by the data.
2. When the authors shift their focus on the tumor microenvironment (TME), they largely disregard the Nanostring GeoMx-based study on a 20-patient cohort, by shifting their analytical approach to a combination of single-cell transcriptomics on dissociated cells coupled with a single whole-slide single-cell targeted transcriptomics experiment on an individual case. Higher spatial resolution is required to explore large cells with complex morphologies as found in the TME. Yet, spatial information is provided only for a single patient, limiting the ability to extrapolate conclusions at the cohort level. Expanding fibroblast characterization to multiple patients would strengthen the manuscript conclusions.
3. By primarily focusing on the comparison between the core-associated and front-associated TME, the authors overlooked that invasive tumor cells, by penetrating through the muscularis mucosa, become exposed to the submucosal TME characterized by peculiar stromal populations, which in turn reacts to the invading front. The authors interpret TME fibroblast patterning primarily by comparison with normal stromal mucosa cell identities, but do not explore the contribution from submucosal-specific cell types, as well as their response to the tumor edge.
4. The authors link the acquisition of metastatic competency to fibroblast-released extracellular factors, like TGF- β ligands and prostaglandins. While the role of these pathways is very well-known in the field, the identification of specific receptor-ligand interactions would represent a significant advance. However, the work to characterize and validate some of the ligands highlighted in the manuscript (e.g., TGFB3) appears to be very preliminary. For example, it's unclear which specific TGF- β ligand the authors use in their study (e.g., is it TGFB1 or TGFB3?). Also, the functional assays to test for the activation of the invasive front-specific transcriptional program (e.g., EMP-1 knock-in-based flow-cytometry and a qPCR assay) are poorly documented and display substantial variability. If the authors added TGFB1 to their organoid lines is unsurprising to see a downregulation of an intestinal stem cell marker, like LGR5, and an increase in some injury repair markers, as EMP1, MMP7 and ANXA1 (it should be noted that EMP1 became highly expressed also during intestinal differentiation). Yet, such observations say relatively little on the ability to recapitulate the invasive front specific program and even less on the acquisition of metastatic competency.

Version 2:

Reviewer comments:

Referee #1

(Remarks to the Author)

Fibroblast patterning instigates metastasis-prone oncofetal states at colorectal cancer onset

Buissant des Amorie et al.,

Nature 2025-03-05750B

I thank the authors for carefully considering the feedback from my first review.

In my opinion the major strength of this paper is showing how trophocytes induce revival stem cells in early colon cancer. I do not think the word metastasis should be in the title of the paper, when there are no mets in the paper and the focus is on early NON-metastatic disease. One of the major findings of the paper is that revival states are ubiquitous in NON-metastatic tumors. Revival/regenerative stem cells are also commonly induced during non-cancerous tissue damage and infection (that's how they got their name) so obviously these cells don't cause metastasis on their own (otherwise we'd all have liver mets after a bowel infection). Access to the revival stem cell state is a normal response of intestinal epithelia to stress and inflammation. In my opinion the message of this paper is that revival stem cells are common at the invasive fronts of early disease, being necessary but not sufficient for metastasis.

The new Figure 5 is much, much stronger and substantially strengthens the temporal and spatial claims of the paper. This for me is what this manuscript is all about.

In my opinion, a more appropriate title of the paper would therefore be 'Trophocytes induce revival stem (or oncofetal) cells at the invasive front of early colon cancer'.

If space is limited due to the Nature Letter format, Figure 4 could also be moved to sup. The Figure 4 title says 'Fibroblasts-derived TGF β and prostaglandins induce invasive front tumor cell phenotypes' but doesn't show any invasive fronts. Again a more appropriate title would be 'Fibroblasts-derived TGF β and prostaglandins induce revival stem cells' but that is already known (Qin et al., Cell, 2023, Roulis et al., Nature 2020) so I'm not sure it's needs to be a main figure in a Nature paper if space is limited. Replacing Figure 4 with the new Figure 5 would help focus on the novel aspects of the paper.

The authors say they have removed the term "High Relapse Cells" (HRCs) from the manuscript but there are still mentioned in several places, including main figures (e.g. Fig 1k, Fig 3m, n, q, Fig 4b, Fig 5c, d, f, g, h). As the authors show in Figure 1i the HRC signature is the same as the revival/regenerative signature. Even the new term "metastasis-associated fetal-like cell states" still mentions metastasis, when there is no metastasis. Why not just use a revival stem cell signature throughout the paper to avoid implying metastasis in non-metastatic tumours? Revival stem cells are interesting for lots of reasons beyond metastasis (e.g. chemo-radioresistance, tumour initiation etc) so calling them "High Relapse Cells" is unnecessary at best, and misleading at worse because these majority of these patients did not relapse.

There is some really great data in this paper and the find that trophocytes induce revival stem cells in early-CRC is novel and interesting to the field. If the authors can stop equating revival stem cells = metastasis (without showing metastasis) I think the fibroblast-patterning discovery message will be a lot clearer for readers.

All other points have been appropriately addressed and I congratulate the authors on a beautiful paper.

Referee #2

(Remarks to the Author)

The revised version of this study from Buissant des Amorie, Hageman, Brunner, van der Horst et al., presents a comprehensive and detailed series of new experimental data that directly address the points raised by this reviewer during the original round. As such, I have no hesitancy in recommending this for acceptance.

The revised version contains additional new fundamental understanding that very clearly demonstrates that the presence of the regenerative cellular state (identifiable by many overlapping, albeit uniquely named signatures) is not always a driver of metastasis in every tumour stage or setting. This was shown in the first submission via its ubiquitous expression in almost every pT1 which rarely metastasise, however in this revised version, the authors now unequivocally de-tangle and longitudinally track this paradoxical and dynamic relationship between "metastasis-prone" HRC/regenerative signalling and outcome. Additionally, the immune exclusion phenotype (repressed viral response signalling) are nicely decoupled from the timing of the regenerative signalling; a point that will no doubt open a therapeutic window for future investigation.

While these points will become very clear to those that read the paper in full, the title and abstract dont quite get across that these regenerative/onco-fetal/HRC.... states are almost ubiquitous from the moment of transformation and submucosal breach. The authors do say they emerge at these stages, but I guess the fact that these are uniformly reactive states to any/all submucosal interactions at this stage is really important, and not related to the metastasis potential of a tumour. The field has a habit of conflating a specific gene/biology with a fixed phenotyp (like being a good or bad driver); importantly, what this study does so well is to add the context, features, evolutionary stage and timings that all need to be present in order for these regenerative/onco-fetal/HRC.... states to actually mean what many in the field assume they always mean (metastasis).

Overall, I congratulate the authors on this excellent work. The authors have addressed my original concerns, added new data and revised analyses to reveal new understanding of the interplay between regenerative cell states, stromal populations, immune exclusion (via viral response signatures) and tumour evolution during developmental/progression phases.

Referee #4

(Remarks to the Author)

I have been asked to review this manuscript specifically with respect to the authors' responses to Reviewer #3, who has unfortunately withdrawn. I greatly enjoyed reading the revised manuscript, which makes a substantial contribution to the field by addressing important biological questions through contemporary techniques and the use of very novel early-stage CRC tissue datasets. The experimental work is scientifically rigorous, well executed, and clearly presented. The manuscript reads well and is significantly improved compared with the initial submission.

All three reviewers (including Reviewer #3) acknowledged the strength of the data and methodology but raised overlapping concerns around conceptual novelty, interpretation, and the lack of mechanistic insight.

I find the authors' responses to Reviewer #3 to be satisfactory in the sense that they have strengthened the manuscript and clarified points of confusion. However, despite these improvements, the work remains in my view an incremental advance.

The conceptual novelty lies largely in the temporal and spatial refinement of known principles of plasticity and CAF signalling, rather than in the discovery of new mechanisms. The immune evasion “bottleneck” proposed by the authors is an appealing explanatory model, but at present is supported only by correlative transcriptomics rather than causal evidence.

I also remain concerned about terminology. Even in the revised version, the use of “metastasis-prone oncofetal states” seems overstated. The cells described share transcriptomic similarity with metastatic states, but no functional data are presented to demonstrate that they are in fact metastasis-prone. Given the plasticity of revival/regenerative/fetal-like populations, a more neutral description would be more accurate.

In my opinion, to cross the bar for Nature, the following types of data (not necessarily all) would be required to demonstrate mechanistic breakthrough:

- a) Necessity and sufficiency of stromal cues: Move beyond the well-established TGF β /PGE2 to identify and test additional ligands. Application of purified gradients that mimic the invasive front could test whether these signals alone recreate the in vivo patterning.
- b) Causal link between fetal-like states and immune evasion: Tri-cultures of tumor organoids with matched trophocyte/CAF populations and autologous T cells to test whether stromal cues reduce immune killing via tumor state transitions. Additionally, determine whether CAF-conditioned invasive fronts recruit immunosuppressive myeloid states.
- c) In vivo impact on dissemination: Orthotopic, barcoded lineage-competition assays with manipulation of CAF cues to test whether induction of fetal-like states alters metastatic seeding potential.

Version 4:

Reviewer comments:

Referee #4

(Remarks to the Author)

I thank the authors for their careful and considered response to the queries raised in my previous review. Terminology has been refined appropriately, and the additional experimental work included in Extended Data Figures 9 and 10 strengthens the manuscript to a level consistent with publication in Nature.

While individual reviewers may naturally differ in their emphasis on novelty, I recognise this as an important and timely piece of work that is analysed rigorously and presented with exceptional clarity. It makes a significant contribution to the field, and I congratulate the authors on a splendid study.

Buissant des Amorie, Hageman, Brunner, van der Horst *et al.*, Rebuttal letter

We thank the reviewers for the time dedicated to evaluate our manuscript and for their overall enthusiasm with regards to the unique samples of early-stage CRC and depth of the methodology featured in our study.

We recognize that all three reviewers, to some degree, raise concerns about the novelty of the mechanism that we describe and therefore want to respond to this here. Indeed, regenerative/fetal-like stem cells are heavily studied across cancer types. However, what sets our manuscript beyond any other literature is that we add conceptual understanding regarding the *timing* and *restricted patterning* with which this intensively studied phenomenon, i.e. metastasis-relevant cell plasticity, first occurs during human colorectal cancer progression. Second, we embed our findings in a unique *in vivo* context that relates to the critical, yet for practical reasons understudied, malignant transformation of pre-cancers to cancers in human patients.

We appreciate the concrete suggestions of the reviewers on how to improve and/or substantiate the novel claims in our study. We are confident that the following additions, in line with the reviewers' key suggestions, will further increase the conceptual advancement:

1. **Regenerative cell states \neq metastatic success:**

Although regenerative stem cell states are associated with poor prognosis in late-stage CRC, we originally detected these tumor cell states in all invasive fronts of our early-stage CRC cohort (n=19 T1 CRCs). This uniform presence at the earliest stage, in particular in non-metastatic T1 CRCs, challenges prevailing concepts and indicates that these states are a prerequisite, but not the sole determinant of metastatic capacity. We now provide the following important additions to substantiate this claim, and explain its apparent contradiction:

- **232 T1 CRC cases with 5-year clinical follow-up:** We have now substantiated our discovery that regenerative stem cell states emerge at the invasive front immediately after malignant transformation, by documenting their presence in 232 cases of the earliest stage (T1) of human CRC (using tissue microarray stainings). Moreover, these samples include 5-year clinical follow-up information, which confirmed the mismatch between the almost uniform presence of regenerative stem cell states among T1 CRCs while only a small fraction of them are metastatic. With this large sample cohort, we can now make the most conclusive statement to date that the presence of regenerative stem cell states in primary cancers is not the sole determining factor for metastatic disease.

See *Figure 1L-m of the revised manuscript (lines 111-115)*

- **Immune evasion:** Further in-depth analyses of the datasets already present in the original version of our manuscript, in particular comparing expression patterns in metastatic vs non-metastatic T1 CRCs, indicate immune evasion to be an important secondary requirement for metastatic capacity. While immune evasion is a known phenomenon, the novelty is in the fact that the timelines by which two cancer hallmarks emerge, i.e. plasticity and immune evasion, do not align. Regenerative cell states generally emerge at the start of malignancy, while the moment by which cancers acquire immune evasion varies more extensively, with

few tumors having accomplished this feat at the T1 stage. This highlights immune evasion to be a major secondary bottleneck to control metastatic seeding. Moreover, it reconciles the seemingly paradoxical findings in our initial submission that showed metastasis-associated cell states to be widespread in T1 CRCs, of which only a minority is truly metastatic.

See *Figure 1n-o and Figure 3e of the revised manuscript (lines 119-127, 177-182)*

2. **Timing and spatial patterning of emerging plasticity:**

While regenerative stem cell states have been found in advanced CRCs (and are predominantly studied in that context), we elucidated that the moment at which regenerative stem cell states arise is in fact already early during CRC progression. Moreover, we identified an unexplored subtype of cancer-associated fibroblast (CAF), reminiscent of tissue-resident trophocytes and localized at the invasive front, as a major source of regenerative stem cell inducing factors. With our new inclusions, we now provide comprehensive understanding as to how fibroblast patterning orchestrates induction of tumor cell plasticity at that specific time and location:

- **Single-cell spatial transcriptomics on three critical substages of T1 CRCs:** Following the first submission of our manuscript, we embarked on an effort to precisely pinpoint the emergence of the regenerative/fetal-like stem cell states and better understand the origin of the various CAF populations. We generated single-cell spatial transcriptomics data of 11 tumor specimens (1.25M cells with 6k gene panel) at three critical substages of T1 that surround the exact moment of malignant transformation, i.e. before invasive front formation (intramucosal carcinoma), immediately after (T1sm1) and the latest T1 stage (T1sm3). These experiments precisely touch upon the specific suggestions of the reviewers to increase novelty, i.e.:
 - i. **Timing:** There are no regenerative cell states and CAF populations prior to rupture of the muscularis mucosae (intramucosal carcinomas, the earliest CRC substage that is still confined to the mucosa).
 - ii. **Spatial patterning:** Trophocyte-like CAFs emerge directly when the tumor penetrates into the submucosa (T1sm1) and line the invasive front where they co-localize with the first induced regenerative tumor cell states. The MMP+ trophocyte-like CAFs, which have shifted even further to the more classical CAFs, only become detectable in more advanced invasive fronts at the latest T1 stage (T1sm3).
 - iii. **Origin of CAFs:** Pseudo-timing of the three substages within T1 CRCs, together with the phenotypic similarity between tissue-resident trophocytes in the submucosa and trophocyte-like CAFs that emerge at the invasive front, provide the most robust evidence to date to identify submucosal trophocytes as the first cells-of-origin for the classical CAFs in human CRC. Moreover, also trajectory inference analyses of fibroblast subtypes supports such a differentiation trajectory.

See *completely new Figure 5 of the revised manuscript (lines 271-308)*

All in all, we are confident we have adequately addressed all concerns raised by the reviewers, both experimentally and conceptually, and have raised our manuscript well beyond the current state of literature. Please find our point-by-point responses to all reviewer comments and suggestions below. Main additions/changes to the manuscript are highlighted in yellow in the manuscript file.

Referees' comments:

Referee #1 (Remarks to the Author):

Through multi-region sampling of early-stage tumours, Buissant des Amorie et al., report an interesting study describing the early emergence of revival/regenerative/oncofetal stem cells at the invasive front of primary CRC tumours. The authors show how revival stem cell plasticity is likely non-genetic and can be driven by stromal fibroblasts in the tumour microenvironment. Stem cell plasticity has recently emerged as a hallmark of CRC and is under intense investigation by many labs in the field.

The paper is well written and the data support the conclusions. My main issue is that most of the finding in the paper are already known to the field. For example:

1. Revival stem cells can be activated by fibroblasts
(Efremova et al, preprint, 2024, Ramos et al., Cell, 2023, Qin et al., Cell, 2023)
2. Revival stem cells are important for CRC metastasis
(Efremova et al, preprint, 2024, Moorman et al., Nature, 2024, Ganesh, Nature Cancer 2020)
3. TGF- β and PGE2 can drive revival stem cells
(Qin et al., Cell, 2023, Roulis et al., Nature 2020)
4. CRC plasticity is largely non-genetic
(Rehman et al., Cell, 2021, Househam et al., Nature, 2022)

Some of these important references are also missing from the paper. Many of these topics have also been covered in recent reviews on the topic (Fey et al., Cell Reports 2024 and Tape, Trends in Cancer 2024). The finding that revival stem cell activation can be seen very early in human tumours is interesting, but even then, the role of YAP signalling (that controls access revival stem cells) in CRC initiation via mesenchymal cells is known (Roulis et al., Nature 2020).

The finding that spatial patterning of CAFs lead to the revival state at invasive fronts is novel to my knowledge. This is the title of the paper and the most interesting part of the manuscript.

While this paper representants a substantial body of excellent work, is very robust, and I enjoyed reading it, I am not sure if the finding that fibroblast heterogeneity can regulate revival stem cells justifies publication in Nature.

We thank the reviewer for their thoughtful evaluation and for the positive remarks regarding the robustness of our data and the overall quality of the work. We particularly appreciate their recognition of the most novel aspect of our manuscript, namely that the patterning of fibroblast populations orchestrates both the location and the timing by which regenerative cell states are first induced during tumor progression.

We recognize that several of the concepts explored in our manuscript are thematically related to mechanisms described in advanced CRC, particularly regarding stem cell plasticity and stromal-epithelial interactions. We acknowledge that we should have discussed this literature in the original version more thoroughly. In the revised manuscript, we included references to all the mentioned papers and improved clarity on how our findings extend beyond these prior works (lines 58-60; 329-338).

Critically, none of the mentioned studies had the ability to focus on the earliest stage in human CRC progression, the precise context in which we:

1. elucidate the first moment at which plasticity emerges during tumor progression.
2. show that the presence of regenerative cell states seems uniform across T1 CRCs, highlighting that its presence is not the sole determinant for metastatic seeding.
3. are the first to describe the spatial patterning of fibroblast subtypes that instigate regenerative stem cell states, providing conceptual understanding of the initiation and restricted patterning of metastasis-associated regenerative tumor cells states.
4. present evidence that submucosal trophocytes are the predominant cellular source from which early CAFs originate.

Below we briefly clarify how our work differs from the specific studies mentioned:

- **Efremova et al.** propose ligand-receptor interactions between fibroblasts and regenerative tumor cells, based on computational inference. However, these predictions were not functionally validated, nor were fibroblast subtypes resolved. In contrast, we experimentally validated key stromal-epithelial interactions using human fibroblast/organoid co-cultures and ligand perturbations, and we systematically characterized distinct fibroblast subtypes.
- **Ramos Zapatero et al.** examined the impact of fibroblasts on chemotherapy resistance in late-stage CRC, without spatial context, subtype discrimination, or specific investigation of invasion or metastatic initiation.
- **Qin et al.** demonstrated that TGF- β can induce revival-like states in mouse organoids co-cultured with normal fibroblasts. However, their system did not model early-stage human CRC or use spatially defined fibroblast subtypes, and their findings were not linked to the invasive front *in vivo*.
- **Moorman et al.** explored plasticity during liver metastasis outgrowth in advanced CRC, which is a biologically distinct setting from the invasive front at the earliest stages of primary tumors that we study.
- **Ganesh et al.** identified L1CAM⁺ regenerative cells as metastasis-initiating in advanced CRC. We do not claim novelty in identifying this state. Rather, our key contribution is to demonstrate the timing at which regenerative cells first emerge and that they are *not* sufficient for metastasis, as they are frequently observed in early-stage tumors that have not metastasized.
- **Roulis et al.** investigated fibroblast-induced tumor initiation via PGE2 in mice, providing important insights into the initiation phase of benign adenomas. However, their work does

not address the transitioning to malignancy, nor understanding on the spatial patterning and timing of regenerative cell states in human CRC.

- **Rehman et al.** describe a drug-tolerant persister (DTP) state arising during chemotherapy in PDX models. While this is an interesting example of non-genetic plasticity, it is unrelated to fibroblast-driven induction of regenerative programs at the invasive front and does not investigate early-stage CRC or spatial patterning to resolve its instigation.
- **Househam et al.** also conclude that transcriptional plasticity is often uncoupled from genetic variation in CRC. However, their study does not explore the timing nor the spatial organization of fibroblast-subtypes that orchestrate their formation at the invasive front.
- **Mzoughi et al.** studies the molecular mechanisms and targeting of regenerative oncofetal reprogramming in genetic mouse models. There is no focus at pre-cancer to cancer transition (timing) nor tumor-stroma spatial patterning and interactions as instigator.

In summary, while the cited studies touch on related concepts such as plasticity, regenerative stem cells, and fibroblast signaling, our manuscript offers a unique and novel perspective by focusing on early-stage human CRC, thereby providing conceptual understanding of the timing and spatial patterning by which these heavily studied processes first occur during human tumor evolution. We have clarified these distinctions more explicitly in the revised manuscript (lines 58-60 and lines 329-338).

Last, we appreciate that the reviewer identified more opportunities to increase the novelty and impact of our manuscript. In particular, to resolve the question of how fibroblast subtypes arise in cancers. Below, we outline how we have incorporated this point (response to Q3 and related comments from the other reviewers) and all other suggested ideas to improve our manuscript.

Additional points:

1. By calling cells at the invasive front ‘High Relapse Cells’ (HRCs), there is a general presumption that revival stem cells = metastatic cells. It is entirely possible, and arguably more likely, that most of the revival stem cells that are induced in stromal niches stay in the primary tumour, with no metastatic dissemination. Given that Stage 1 and 2 patients very rarely relapse with metastatic disease, I think it is unfair to call LAMC2+ or EMP1+ cells in early-stage tumours ‘High Relapse Cells’. In my opinion, these cells in primary tumours should be called revival / regenerative / oncofetal-like stem cells, not HRC which is really a name given to revival stem cells found in liver mets that can lead to disease relapse.

We thank the reviewer for pointing out the potential confusion around our use of the term “High Relapse Cells” (HRCs). We fully agree that the regenerative cells observed at the invasive front in early-stage CRC are, in the vast majority of cases, unlikely to progress to overt metastasis. We used the term HRCs in reference to the original *Nature* publication that introduced this nomenclature in relation to EMP1+ cells at the invasive front of primary tumors (not exclusively in metastatic lesions) and most conclusively demonstrated that these cells gave rise to metastatic seeding. However, we

now realize that reusing this term outside its original metastatic context may lead to misinterpretation.

To avoid further confusion, throughout the revised manuscript we now use a more neutral term, such as metastasis-associated fetal-like cell states, better reflecting the observed phenotype without implying successful metastatic seeding. Importantly, one of the central insights of our study is that these regenerative states alone are not sufficient for metastasis formation. While such states may be a prerequisite for metastatic potential (as shown in previous literature), our findings demonstrate their almost uniform presence in early-stage tumors of which the majority is not metastatic (yet), underscoring that other factors are involved as well.

To substantiate on the latter statements:

- we now document the presence of regenerative cell states in 232 cases of the earliest stage (T1) of human CRC using tissue microarrays (Figure 1l-m). These samples include 5-year clinical follow-up information, substantiating the apparent mismatch between the presence of regenerative cell states in patients without metastasis. Furthermore, analysis of published scRNA-seq data shows that the expression of the HRC program in regenerative tumor cells remains largely stable during disease progression, while the relative frequency of cells expressing this program increases (Extended Data Fig. 2g-h). Therefore, we can now make the most conclusive statement to date that the mere presence of regenerative cell states in primary colon cancers is not the sole determining factor for metastatic disease.
- further in-depth analyses of the datasets already present in the original version of our manuscript, in particular comparing expression patterns in metastatic vs non-metastatic T1 CRCs, indicate immune evasion to be an important secondary requirement for successful metastatic seeding (Figure 1n-o and Figure 3e, lines 119-127, 177-182). While immune evasion is a known phenomenon, the novelty lies in the fact that the timelines by which two cancer hallmarks emerge during tumor evolution, i.e. plasticity and immune evasion, do not align. Regenerative cell states generally emerge at the start of malignancy, while the moment by which cancers acquire immune evasion varies more extensively with few tumors having accomplished this feat at the T1 stage.

2. The indirect co-culture model cannot account for matrix induced effects that are known to be important for revival stem cell activation. Why didn't the authors use direct co-cultures?

We agree with the reviewer that direct co-cultures would be an appropriate approach to investigate if matrix-dependent effects play a role in addition to the paracrine effects that we already observed in the co-cultures and have therefore incorporated such experiments in the revised version of the manuscript.

We initially chose to employ indirect co-culture systems primarily because they allow for straightforward independent expression analysis of the fibroblast and organoid compartments. Not only did this enable the identification of soluble signaling factors mediating paracrine interactions, it

also provided the opportunity to validate their expression in the *in vivo* context to be predominantly restricted to trophocyte-like CAFs (Figure 4f).

In the revised manuscript, we performed new experiments to test additional matrix-dependent effects. Organoids were cultured in Matrigel-Collagen (type I) gel mixtures with fibroblasts in the same matrix either in transwells (indirect co-culture, paracrine) or embedded within the same droplets as the organoids (direct co-culture, juxtacrine). We used our EMP1-mNeon knock-in reporter organoids in combination with flow cytometry to measure induction of regenerative tumor cell states (Extended Data Fig. 9c). In organoid monocultures, we did not observe a direct effect of the change in matrix (Extended Data Fig. 9e). However, we did observe an effect of the matrix change in the co-culture setups, as both the indirect (transwell) and direct co-culture in Matrigel-Collagen mixtures induced regenerative tumor cell states more potently (Extended Data Fig. 9e) than the indirect (transwell) co-culture in Matrigel alone. While tension/stiffness or contact-dependent signaling could potentially affect epithelial tumor cell states, we did not observe clear differences between the juxtacrine and paracrine setups with the Matrigel-Collagen matrix, suggesting that the matrix change mostly effects the fibroblasts.

3. What is driving the fibroblast heterogeneity? Is it stochastic or do signals from cancer cells alter fibroblast populations?

The reviewer highlights an important question, that relates not only to identifying the cell-of-origin of CAFs, but indirectly also provides conceptual understanding of the timing and patterning by which CAF subtypes emerge and in turn orchestrate cellular plasticity during tumor progression.

To improve the conceptual advancement provided by our manuscript and maximize the power of analyzing spatial patterning in early-stage cancers, we embarked on an effort to better understand the origin of the various CAF populations and to pinpoint the emergence of the regenerative/oncofetal-like stem cell states. To this end, we generated single-cell spatial transcriptomics data of 11 tumor specimens (1.25M cells with 6k gene panel) at three critical substages of T1 that surround the exact moment of malignant transformation, i.e. before invasive front formation (intramucosal carcinoma), immediately after (T1sm1) and the latest T1 stage (T1sm3) (new Figure 5). These experiments precisely touch upon the key question raised by the reviewer, namely:

- a. There are no regenerative cell states and CAF-like subpopulations prior to rupture of the muscularis mucosae (intramucosal carcinoma, confined to mucosa) (Figure 5a-f).
- b. Trophocyte-like CAFs emerge directly when the tumor penetrates into the submucosa (T1sm1, Figure 5c and Extended Data Fig. 10c), and these CAFs line the invasive front (Figure 5d and f) where they co-localize with the first induced regenerative cell states (Figure 5g and Extended Data Fig. 10d). The more classical MMP+ trophocyte-like CAFs only become detectable in the more advanced invasive fronts at the latest T1 stage (T1sm3).
- c. Pseudo-timing of the three substages within T1 CRCs, together with the phenotypic similarity between tissue-resident trophocytes in the submucosa and trophocyte-like CAFs that

emerge at the invasive front (Figure 5i and Figure 3gh), provide the most robust evidence to date to identify submucosal trophocytes as the predominant cellular source from which early CAFs originate in human CRC.

- d. Moreover, trajectory inference analyses identify plausible differentiation trajectories from trophocytes to trophocyte-like CAFs (Figure 5j-k) and CytoTRACE analysis positions trophocytes at the apex of the fibroblast differentiation hierarchy (subtype with the highest CytoTRACE potency score, Extended Data Fig. 11e). Thus, these independent and unbiased analyses support our findings that trophocytes are the dominant cell-of-origin of trophocyte-like CAFs.

The transitioning of submucosal trophocytes to trophocyte-like CAFs during invasive front formation explains how the original fibroblast population diversifies, and why the newly identified CAF subtypes strongly resemble tissue-resident counterparts from which they originated. Likewise, telocyte-like CAFs could emerge by transitioning from telocytes in the tumor core (lines: 353-355). How the trophocyte-telocyte axis as seen in normal mucosa is maintained in the tumor, how the transitioning of the tissue-resident fibroblast population to CAFs is triggered, and how tumor cells influence their states, are interesting future research direction for which this manuscript lays the foundation (lines: 355-361).

I applaud the excellent science but from a publication perspective feel the work has re-discovered what was already known. The novelty of the paper lies in the fibroblast heterogeneity angle which only comprises x1 Figure in the paper.

We thank the reviewer again for complimenting the type of science that we conduct. Respectfully, we would also like to reiterate the important distinction and novelty in our work in relation to published work that predominantly involves advanced cancers; we provide conceptual understanding and *in vivo* context of the timing and restricted patterning at which two intensively studied phenomena, i.e. stem cell plasticity and the origin of various CAF populations, first occur during human tumor progression.

Referee #2 (Remarks to the Author):

des Amorie et al present an eloquent, well-written and high quality characterisation of a novel tissue & organoid cohort from T1 CRCs, combining detailed single cell & spatial interrogation with patient-matched regional-specific organoid experimental validation. The cohort itself is highly novel, as while most similar collections have been developed from advanced cancers, the approach presented here in T1 tumours & and methodologies used for patient-matched multiregional organoid isolation from fresh T1 surgical resections are beyond any other approaches this reviewer has seen before. The authors utilise this cohort initially for discovery and identify a number of the previously published (Ref 17 – EMP1/HRC, Nature 2022) biomarkers for aggressive epithelial phenotypes are elevated in almost all T1 invasive front samples used in this study, which contrary to the previous publications these are elevated regardless of T1 outcome (this cohort contains 7N&M-, 10N+, 3M+). The authors also identify a series of previously published fibroblast subtypes, most of which reminiscent of normal crypt/tissue locations or cancer-associated states, which they refine further using their T1 cohort data to detail spatial arrangement of these during the early cancer development stages. Utilising the patient-matched organoid bank, the authors show that there is an absence of mutational drivers of these EMP1/HRC changes in patient samples, but the EMP1/HRC phenotype can be elevated in all samples from all regions (all those tested here) using TGFb and/or PTGS2 (which themselves are identified from the fibroblast populations)

My main concerns here are that while the sampleset are novel, and the analyses are certainly state-of-the-art, the results are largely confirmatory of a previously identified (seen more widely than just the EMP1 study) set of mechanisms underpinning HRC/regenerative epithelial states. The authors discuss precisely (291-299) the two key points that that could be addressed that would push this beyond what has been published, which will be expanded on below.

I would again reiterate that I fully agree that the bias in previous studies has been towards later stage or more established disease, and little has been done to understand early signs of metastatic competence in CRC; points which this current paper addresses. The suggested main points aim to address the mechanisms underpinning the switch during tumour development for the widespread expression of this phenotype regardless of outcome to the point where expression of this phenotype is strongly associated with outcome.

First, we thank the reviewer for their kind remarks and for recognizing the strength and uniqueness of the early-stage CRC cohort we developed. We fully agree that providing additional insight into the induction and origin of CAF phenotypes (lines 291-299 of our initial submission) would further elevate the novelty and impact of the study. Therefore, we generated additional spatial transcriptomics data capturing the critical moments of invasive front formation to pinpoint the exact time at which CAFs emerge and elucidate their cells-of-origin. Please see the first two introductory pages of this rebuttal letter for a more elaborate explanation of these experiments.

Second, we realize that several of the concepts explored in our manuscript are thematically related to mechanisms described in advanced CRC, particularly regarding stem cell plasticity and stromal-

epithelial interactions (also see our more elaborate response to comment 1 of reviewer 1). The important distinction is that we provide conceptual understanding and *in vivo* context of the timing and local patterning at which these phenomena first occur during human tumor progression, which we now state more explicitly in the revised version of the manuscript (lines 58-60 and lines 329-338).

Last, the reviewer raises an important point in the end that relates to the seemingly incompatible observation between the widespread expression of the metastasis-prone regenerative stem cell state in early-stage tumors with generally good prognosis, versus its association with poor prognosis in late-stage tumors. This apparent mismatch is one of the cardinal points identified by all reviewers and we have now resolved this issue in the revised version of the manuscript (lines: 111-127). In short, we first corroborated our finding of widespread expression of the metastatic-prone cell states in a large T1 sample cohort that include 5-year clinical follow-up information (n=232). Next, we performed additional in-depth analyses on our spatial transcriptomics dataset comparing metastatic and non-metastatic T1 tumors. We find clear evidence that the regenerative tumor cells at the invasive front of the metastatic T1 tumors had acquired immune evasive properties, highlighting immune evasion as an important secondary bottleneck in human CRC.

Main points:

1 - Line 99-100: related to my main point above, these findings around how stromal/inflammatory factors can drive a regenerative state have been well described before, both in the Cañellas-Socias study but in others like Vasquez et al 2022 <https://pubmed.ncbi.nlm.nih.gov/35931031/> or Qing et al Cell 2024 [https://www.cell.com/cell/fulltext/S0092-8674\(23\)01221-7](https://www.cell.com/cell/fulltext/S0092-8674(23)01221-7) or Zapatero et al Cell 2024 [https://www.cell.com/cell/fulltext/S0092-8674\(23\)01220-5](https://www.cell.com/cell/fulltext/S0092-8674(23)01220-5) – some of these studies even use TGFb additives or fibroblast co-cultures to induce the same phenotypes (many not cited here). In the context of these previous studies, the main findings to this reviewer is the one that describes that in T1 tumours, these signals are almost uniformly elevated across the invasive front of all lesions regardless of the metastatic status. So, to this reviewer it would appear that the findings of HRC cells in T1s are not so much a metastatic phenotype but rather a phenotype of epithelial cells that have a location closer to the invasive front (Authors reiterate this in line 266-269). So overall, this HRC signal is not driving metastatic phenotypes in T1, and only in later disease; a point that the authors discuss how to address and action on.

We agree that the studies mentioned by the reviewer do not examine the stage and spatial pattern at which regenerative states emerge, which represents a central and novel aspect of our manuscript. In addition, these studies do not provide detailed insight into CAF subtypes or their spatial organization, another key contribution of our work. That said, we acknowledge the omission of some of these relevant references and have ensured they are properly cited in the revised manuscript.

With regards to the apparent mismatch between the uniform presence of metastasis-associated HRC cell states at T1 invasive fronts regardless of metastatic status, we kindly refer to our more elaborate response to comment 1A and 1B below.

In lines 209-212, the authors also discuss this point in relation to the fibroblast lineages that drive this phenotype, but state that these signals “are associated with metastatic HRC tumour states, high

TME-HR signatures and poor patient outcome” – however, again, this only appears to hold up in later stage disease, as detailed in previous studies and to some extent in Fig3e of this study, but most importantly not in the T1 cohort itself.

So this to me is the critical point, and as the authors clearly describe in lines 291-299, requires either of two approaches to answer the question they have posed from the findings of this current study.

A - How and when do these signals stay switched on during tumour progression in tumours that maintain metastatic phenotypes and ultimately relapse/progress, compared to how are these features lost in tumours that do not relapse? This would require assessment of samples at along the different stage of development/progression for the fibroblast & epithelial phenotypes described here.

Regarding the emergence, dynamics and stability of the regenerative cell states during tumor progression, we have added the following experimental data and insights:

First emergence of metastasis-prone regenerative tumor cell states

To support our finding that regenerative tumor cell states are uniformly present at the invasive fronts of early-stage cancers, we now documented their presence in 232 cases of the earliest stage (T1) of human CRC using tissue microarrays (Figure 1l-m, lines: 111-115). These samples include 5-year clinical follow-up information, substantiating the apparent mismatch between the widespread presence of regenerative cell states in early cancers that are mostly non-metastatic. Therefore, we can now make the most conclusive statement to date that the mere presence of regenerative cell states in primary colon cancers is not the sole determining factor for metastatic disease.

Second, we embarked on an effort to pinpoint the exact emergence of the HRC/regenerative stem cell states during tumor evolution and to better understand the origin of the various CAF populations. For this, we generated single-cell spatial transcriptomics data of 11 tumor specimens (1.25M cells, 6k gene panel) at three critical substages of invasive front formation that surround the exact moment of malignant transformation, i.e. prior to penetration through the muscularis mucosae (intramucosal carcinoma), immediately after (T1sm1) and the latest T1 stage (T1sm3). These experiments (Figure 5, lines 271-308) precisely touch upon the key question raised by the reviewer namely:

- a) There are no regenerative tumor cell states and CAF-like subpopulations prior to rupture of the muscularis mucosae (intramucosal carcinoma, confined to mucosa) (Figure 5a-f).
- b) Trophocyte-like CAFs emerge directly when the tumor penetrates into the submucosa (T1sm1, Figure 5c and Extended Data Fig. 10c), and these CAFs line the invasive front (Figure 5d and f) where they co-localize with the first induced regenerative cell states (Figure 5g and Extended Data Fig. 10d). The more classical MMP+ trophocyte-like CAFs only become detectable in the more advanced invasive fronts at the latest T1 stage (T1sm3).

Pseudo-timing of the three substages of T1 CRC, together with the similarity between tissue-resident trophocytes in the submucosa and trophocyte-like CAFs that emerge at the invasive front (Figure 3g-h and Figure 5h), provide the most robust *in vivo* evidence in human cancer to pinpoint submucosal trophocytes as the cells-of-origin that transform into trophocyte-like CAFs (and later the more

advanced MMP+ CAF) when tumor cells of the intramucosal carcinoma breach the muscularis mucosae. This is further supported by trajectory inference analysis using a large number of fibroblast transcriptomic profiles (~5000) of published scRNA-seq datasets (Figure 5j-k and Extended Data Fig. 11e).

Stability of regenerative tumor cell state during disease progression

Investigating changes in HRC state during disease progression, we integrated scRNA-seq data of tumor cells from different studies that include all tumor stages (including ours to enhance T1 coverage) to assess whether the HRC/regenerative cell state changes (Extended Data Fig. 2a and g-h, lines 115-119). Contrary to the number of HRCs that increase during tumor progression (as was also reported by the Cañellas-Socias study), we find that the expression level of HRC/regenerative signature genes remains largely stable in HRC clusters across the different tumor stages. In combination with our experimental data that shows that regenerative tumor cell states are widespread among invasive fronts of early-stage CRCs, this indicates that the regenerative phenotype of this tumor cell subpopulation does not change substantially during disease progression. Likewise, analysis of the fibroblast cell population in early and late-stage CRCs using published scRNA-seq data, reveals the same diversity of fibroblast subtypes to be present in early and late-stage disease. However, the relative proportion of cancer-associated fibroblasts that can induce HRC states, like the number of HRCs itself, increases with disease progression (Figure 5l).

Together, our analyses indicate that the early emergence of regenerative cell states, the stability of this phenotype during tumor progression and its widespread presence among T1 cancers that are mostly non-metastatic, point to a secondary bottleneck that determines metastatic capacity (see our response to point B).

B – If certain critical bottlenecks can be identified above, detailing on which of these are able to interfere/block phase-specific tumour progression would be needed (perhaps using the same models as the original EMP1 study showed) and ablation of the metastatic phenotype signalling. While the current study certainly adds to the recent data in this area, inclusion of these (or similar) suggested analyses above in early-to-late-stage tumour data alongside follow-up of findings with in vivo experiments described in A/B above would elevate this study well above the current understanding of disease.

We now followed up on the apparent mismatch between widespread regenerative cell states in T1 cancers, while only a minor fraction shows metastatic disease.

Immune evasion is an important second bottleneck

Further in-depth analyses of the datasets already present in the previous version of our manuscript, in particular comparing expression patterns in metastatic vs non-metastatic T1 CRCs, indicate immune evasion to be an important secondary requirement for metastatic capacity (Figure 1n-o and Figure 3e). While immune evasion is a known phenomenon, the novelty lies in the fact that the timelines by which two cancer hallmarks emerge during tumor evolution, i.e. plasticity and immune evasion, do not align. Regenerative cell states generally emerge at the start of malignancy, while the

moment by which these cells acquire immune evasion varies more extensively with few tumors having accomplished this feat at the T1 stage.

As we articulated in the introduction of the manuscript, there are to date no experimental model systems, including genetic mouse models, that faithfully capture malignant transformation where benign adenomas transition into invasive carcinomas. Moreover, xenotransplantation of organoid models requires immune compromised mice which eliminates an important factor influencing metastatic disease. For that reason, we decided to focus our research effort on the patient-relevant *in vivo* context and generate unprecedented datasets of malignant transformation in patients, including assessment of HRC/regenerative states across >200 T1 CRCs with 5-year clinical follow-up information, and state-of-the-art spatial analysis (~1,25M single cells, 6k genes per cell). Related to the latter, arguably the most valuable feature of this 11-patient dataset is its high resolution in terms of three pseudo-timed substages that capture the most critical events during malignant transformation of CRCs.

2 - Lines 68-71: This sampling approach presented here is very similar to the Heiser et al 2023 study in Cell (<https://pubmed.ncbi.nlm.nih.gov/38065082/>; not cited in this current study), which performed phylogenetic tumour evolution tracking across these sample regions and identified an immune exclusion phenotype driven by extracellular matrix signal in MSS tumours (predominantly stage I & II in this MSS subset) using many of the same profiling methodologies used here in this current study. It would be most useful if the author assessed the pseudotemporal assessments, the IEX markers and the spatial CNV inferences in their T1 dataset too, similar to how the authors have looked to confirm the findings of the Cañellas-Sociés et al EMP1/HRC study.

Heiser et al. performed an interesting study with multiregional sampling of CRCs focusing on pseudotime trajectories and immune exclusion scores, although they did not address fibroblasts, regenerative cells or include functional *in vitro* validations. We agree that comparing our findings to theirs is valuable. To that end, we have analyzed the expression of IEX markers in our dataset. While we did observe higher expression of the IEX markers in tumor tissue compared to normal (Rebuttal Figure 1), we did not see higher expression of these markers at the invasive front (Extended Data Fig. 1h) or in metastatic T1 tumors (Rebuttal Figure 1).

Regarding the spatial CNV inference, we had already included such analyses in the initial submission of our manuscript (see Fig. 2f and Extended Data Fig. 4b-c of the revised manuscript). While we cannot replicate their pseudo-time analysis exactly (as it relies on WES data from matched samples), our regions were histologically validated as normal, adenomatous, tumor core, and invasive front. This provides an alternative approach for pseudotemporal ordering based on spatially and morphologically defined tumor progression. This approach allows us to compare CNV scores in a similar way as the pseudotemporal progression trajectory used by Heiser et al. As shown in Extended Data Fig. 4b-c and Figure. 2f of the revised manuscript, the number of CNVs increase from normal to

adenomatous to core and invasive front, supporting the observations by Heiser et al. that the CNV score increases along the pseudotime trajectory.

Similarly, the markers and biologies used in the recent Moorman et al study 2024 Nature <https://www.nature.com/articles/s41586-024-08150-0> could also be assessed here, to provide clarity as to the T1 cellular states compared to those metastatic phenotypes.

This is a great suggestion and in the revised version of the manuscript we added the transcriptional states described by Moorman *et al.* to our analyses to determine their overlap with early-stage CRC cell states. In **Figure 1i** the tumor intestinal stem cell signature from Moorman *et al.* correlated with more classical intestinal stem cell signatures and the tumor core signature from our T1 CRCs. In contrast, the EMT and injury repair signatures from Moorman *et al.* correlate with other regenerative signatures, among which is our invasive front signature of the T1 invasive fronts. Moreover, the non-canonical states described in the Moorman paper were mildly enriched at the invasive front compared to the tumor core (**Extended Data Fig. 1h**). We did not see higher expression of the non-canonical states in the metastatic T1 tumors (**Rebuttal Figure 2**), but these states were also predominantly attributed in Moorman et al. to disseminated cells at metastatic locations.

Minor points:

1- The LAMC2 staining shown in Figure 1h for this single case does look to be associated with the invasive front epithelial cells acquiring this phenotypic marker, however I would argue that the staining presented for the single case with EMP1 in Supp Fig 1h is less than convincing. Ideally the authors would have presented these as multiplexed stains, or at the very least on sequential sections from a wider range of samples.

Related to this point, the choice of EMP1 for the fluorescent reporter seems unwarranted, at least based on the data for EMP1 staining that is presented.

We agree with the reviewer that the EMP1 staining is less robust than that of LAMC2. From a practical point of view EMP1 is a technically very challenging target to stain (including background staining in stroma). It is likely that only high levels of EMP1 expression yield detectable signal by immunostaining, potentially leading to an underestimation of the youngest HRC cells.

In the revised manuscript we have now included new data to solidify the concept that both EMP1 and LAMC2 are good markers for HRC cells (**Extended data Fig. 2a-f**) in the invasive front of early-stage CRCs: a) Using published scRNAseq data we confirm that LAMC2 belongs, just like EMP1, to the set of marker genes whose expression correlates best with the HRC phenotype (**Extended data Fig. 2f**). b) LAMC2 is within the top10 upregulated genes within invasive fronts of T1 CRCs (**Fig. 1h; GeoMx analysis using 5 probes per gene**).

Subsequently, after establishing that EMP1 and LAMC2 are equally good markers on the conceptual level, we have decided to remove the EMP1 immunohistochemistry from the revised version for practical reasons and solely use LAMC2 as a validation marker per IHC in T1 and T2 stage invasive

fronts. We hope this will stimulate colleagues to take advantage of LAMC2 stainings for future HRC detection, as it represents a reliable, robust, but mostly a practical marker for HRC identification with conventional IHC.

Regarding the choice of EMP1 as the fluorescent reporter, both *EMP1* and *LAMC2* are equally good marker genes as we describe above. The one downside of EMP1 is that it is technically difficult to detect with IHC, in contrast to LAMC2. As far as we understand, and as was reported by the study of Canellas-Socias, *EMP1* expression is the best-known marker for the HRC state and robustly reports on this state in organoids.

2- Can the authors include a clearer table (perhaps alongside S1B) that details colonic location of these lesions.

This information is now included in Supplementary table 1.

3 - From UMAP on Fig1d, the point that these are clustering/non-clustering is quite subjective; can the authors perform some objective (clustering/alignment or correlative like in Fig2i) approach to determine this similarity?

We apologize for the omission of objective correlatives of the cluster quality in Figure. 1. In the revised version of our manuscript, we have included variance partition analyses and correlations to substantiate our claims like the reviewer suggested (Figure 1e-g; lines 78-82).

4 – the paracrine studies (lines ~228) are exquisitely performed and fully validate this hypothesis.

We thank the reviewer for their compliments regarding the technical quality of our analysis and experimental set-ups.

Referee #3 (Remarks to the Author):

In this manuscript, Buissant des Amorie, Hageman, Brunner and colleagues explore the molecular mechanisms associated with the acquisition of metastatic traits in colorectal cancer (CRC). Metastases are the main cause of CRC deaths. So far, no genetic driver has been clearly associated with their emergence, and their origin has largely remained elusive.

Here, the authors approach this topic from a peculiar point of view, focusing on the in-depth characterization of a cohort of patients with early-stage disease displaying minimal sign of tumor invasiveness, if any. They produce a remarkable dataset, performing multiregional imaging-based spatial transcriptomics. They sample four region-of-interest (ROIs) across whole resected lesions, including the tumor core, the invasive front, as well as adjacent regions displaying adenoma features, or overtly normal, in approx. 20 patients, with or without locoregional metastases (T1-N0 or T1-N1, according to TNM staging), and using two distinct large scale probe panels (targeting either 1800 cancer-related genes with 5 probes each or a whole-transcriptome set against 18000 transcripts). This analysis revealed stereotypical tumor patterning, including an invasive front transcriptional signature (e.g., EMP1, LAMC1, ANXA1) previously associated (but not causally related yet) with invasive tumor cells in mouse models and relapse in a large clinical cohort (PMID: 36352230), and generally linked with fetal intestinal development, and injury-repair at adult stages, especially in the mouse. They complement their initial analysis by generating a biobank of ROI-specific organoids, which utilize to systematically track ROI-specific genetic alterations or test ROI-specific growth factor dependencies, focusing on tumor core vs. invasive edge, and finding no evidence for significant cancer cell-intrinsic differences between these two sites. They turn their attention to the tumor microenvironment by performing a whole-slide imaging-based spatial transcriptomic experiment spanning an entire tumor lesion and its margins, coupled with a lineage-balanced single-cell RNA-seq experiment, to map cell identities at single cell resolution. Co-culture of some ROI-specific stromal-derived lines with organoids leads to transcriptional changes reminiscent of those observed at the invasive front and led to the prioritization of some stromal-derived ligands, which partially mediate such response (e.g., TGFB3, PGE2).

The manuscript has several strengths, especially related to:

1. Multiregional spatial transcriptomics on T1 lesions across a 20-patient cohort reveal that stereotypical tumor patterning is present from the early stages of tumor development, markedly anticipating observations previously associated to more advanced disease
2. Relying on pathology-guided multiregional sampling (a challenging task not commonly achievable across cancer centers), the authors establish a ROI-specific organoid biobank and systematically test the hypothesis that tumor core- vs. invasive front-derived organoid may have distinct growth factor dependencies, without finding supporting evidence
3. More broadly, the application of complementary spatial transcriptomic approaches to clinical stages unfrequently explored in this field

We thank the reviewer for highlighting the strengths related to the technologies and experimental setups used, and our exploration into a critical, yet relative unexplored stage of human tumor

progression. Foremost, we have taken the reviewer's comments to heart in order to improve and further substantiate our findings and claims in the revised manuscript. Our main additions are summarized in the two introductory pages of this rebuttal letter. Please find below our responses to the specific questions raised.

Conversely, other sections of the manuscript appear to be considerably less developed:

1. The authors find evidence of tumor patterning, as previously observed at more advanced stages of disease, associating an invasive-front specific transcriptional program to metastatic competence at very early stages, as stated in the title. Yet, most T1 patients (approx. 90%) do not relapse after surgery. Disease progression is found only in a minority of T1 cases (approx. 10%). While the study cohort is balanced between T1N0 and T1N1, the authors provide no insights towards the identification of biomarkers or gene expression programs to set apart patients with or without pathological evidence of metastasis, likely due to the small number of patient analyzed. Thus, in the present form, the manuscript title does not appear to be supported by the data.

The reviewer raises a valid point that was shared by reviewer 1 and 2. It partially reflects confusing nomenclature that we used in the initial submission, as well as the striking but puzzling finding that regenerative cell states, associated with poor outcome and metastatic disease, emerge in virtually all CRCs immediately after malignant transformation, while only a minority of these early T1 CRCs are in fact metastatic.

We acknowledge that substantiating our findings and providing more insights into what discriminates T1 CRC patients with -and without metastasis, will enhance context and conceptual understanding of our findings. Therefore, we now documented the presence of regenerative cell states in 232 cases of the earliest stage (T1) of human CRC using tissue microarrays (Figure 1l-m, lines: 111-115). These samples include 5-year clinical follow-up information, enabling us to substantiate the apparent mismatch between the presence of regenerative cell states in patients without metastatic disease. Consequently, we can now make the most conclusive statement to date that the mere presence of regenerative cell states in primary colon cancers is not the sole determining factor for metastatic disease.

In order to understand this finding, we compared the regenerative cell states between early and late-stage CRCs. We integrated scRNA-seq data from multiple studies, including ours, to cover all tumor stages. Analysis of the regenerative cell population shows that the transcriptional programs that drive their regenerative phenotype remain largely stable during disease progression, while the relative abundance of this cell population increases (Extended Data Fig. 2a and g-h).

Next, to investigate determinants of metastatic disease in our early-stage T1 CRCs, we performed further in-depth analyses of the datasets already present in the initial version of our manuscript, in particular comparing expression patterns in metastatic vs non-metastatic T1 CRCs in the spatial transcriptomics dataset. This revealed that invasive fronts in metastatic T1 CRCs had already acquired immune evasive properties, highlighting this known hallmark as an important secondary requirement for successful metastatic seeding (Figure 1n-o and Figure 3e, lines 119-127, 177-182).

While immune evasion is a known phenomenon, the novelty is found in the fact that the timelines by which two cancer hallmarks emerge during tumor evolution, i.e. plasticity and immune evasion, do not align. Regenerative cell states generally emerge at the start of malignancy, while the moment by which these cells acquire immune evasion varies more extensively with few tumors having accomplished this feat at the T1 stage. Moreover, it reconciles the seemingly paradoxical findings in our initial submission that showed metastasis-associated cell states to be widespread in T1 CRCs, of which only a minority is truly metastatic.

Regarding the title and the use of terms like metastatic cell states and/or 'high relapse cells' (HRCs). We fully agree that the transformed cells observed at the invasive front in early-stage CRC are, in the vast majority of cases, unlikely to progress to overt metastasis. We now realize that reusing above mentioned terms based on the similarity of these cells to metastatic states, but outside their original metastatic context, may lead to misinterpretation.

To avoid further confusion, throughout the revised manuscript we now use a more neutral term, such as metastasis-associated fetal-like cell states, better reflecting the observed phenotype without implying metastatic outcome.

2. When the authors shift their focus on the tumor microenvironment (TME), they largely disregard the Nanostring GeoMx-based study on a 20-patient cohort, by shifting their analytical approach to a combination of single-cell transcriptomics on dissociated cells coupled with a single whole-slide single-cell targeted transcriptomics experiment on an individual case. Higher spatial resolution is required to explore large cells with complex morphologies as found in the TME. Yet, spatial information is provided only for a single patient, limiting the ability to extrapolate conclusions at the cohort level. Expanding fibroblast characterization to multiple patients would strengthen the manuscript conclusions.

We apologize if our use of the Nanostring GeoMx dataset beyond figure 1 was not clearly described. In fact, we did use it to analyze the patterning of the tumor microenvironment (TME) (Figure 3i and Extended Data Fig. 6a and c (=Figure 3h and Extended Data Fig. 6a-b of the initial submission)). Specifically, the regional scRNA-seq data that we generated first was instrumental to then deconvolve cell type enrichment within the pseudobulk GeoMx datasets, subsequently generating spatial patterning information for cell types across multiple T1 CRCs (n=19).

Having said that, we acknowledge that spatial information at single-cell resolution was initially only provided for one patient (pt5, also included in the organoid biobank, GeoMx experiment and regional biopsy scRNAseq). To extrapolate our findings, we performed a new single-cell spatial transcriptomics experiment to chart the spatial patterning of fibroblast subtypes across an additional 11 tumor specimens (1.25 cells, with 6k gene panel). Arguably most valuable of this experiment is that these specimens capture three critical substages of T1 that surround the exact moment of malignant transformation, i.e. prior to penetration through the muscularis mucosae (intramucosal carcinoma), immediately after (T1sm1) and the latest T1 stage (T1sm3). These new

experimental datasets (Figure 5, lines 271-308) precisely touch upon key question raised by the reviewers, namely:

- a) There are no regenerative cell states and CAF-like subpopulations prior to rupture of the muscularis mucosae (intramucosal carcinoma, confined to mucosa) (Figure 5a-f).
- b) Trophocyte-like CAFs emerge directly when the tumor penetrates into submucosa (T1sm1), and these CAFs line the invasive front where they co-localize with the first induced regenerative tumor cell states. The more advanced MMP+ trophocyte-like CAFs only become detectable in invasive fronts of the latest T1 stage (T1sm3) that penetrate deeper into the supportive tissue (Figure 5a-f and Extended Data Fig. 10c).

Pseudo-timing of the three substages of T1 CRC, together with the transcriptomic similarity between tissue-resident trophocytes in the submucosa and trophocyte-like CAFs that emerge at the invasive front (Figure 3g-h and Figure 5h), provide the most robust *in vivo* evidence in human cancer to pinpoint submucosal trophocytes as the elusive cells-of-origin that transform into trophocyte-like CAFs (and later the more advanced MMP+ CAF) when tumor cells of the intramucosal carcinoma breach the muscularis mucosae. This is further supported by trajectory inference analysis using a large number of fibroblast transcriptomic profiles (~5000) of published scRNA-seq datasets (Figure 5j-k and Extended Data Fig. 11e).

3. By primarily focusing on the comparison between the core-associated and front-associated TME, the authors overlooked that invasive tumor cells, by penetrating through the muscularis mucosa, become exposed to the submucosal TME characterized by peculiar stromal populations, which in turn reacts to the invading front. The authors interpret TME fibroblast patterning primarily by comparison with normal stromal mucosa cell identities, but do not explore the contribution from submucosal-specific cell types, as well as their response to the tumor edge.

The reviewer is spot on with the insights regarding the possible involvement of submucosal cell populations. We apologize for the lack of clarity regarding submucosal stromal cells in the previous version of our manuscript. To clarify, trophocytes themselves are a subset of stromal cells also found in the submucosa, as described in Kraiczy et al. (*Cell Stem Cell* 2023; PMID 37028407) and Figure 3o-p and lines 31, 297 and 307 of our manuscript. In that sense, our study indirectly focused primarily on submucosal fibroblasts, in alignment with the reviewer's remark.

Moreover, as we elaborately responded to the previous point of the reviewer, we have now substantial new evidence using spatial information on pseudo-timed tumor specimen surrounding the exact moment of malignant transformation. Our current analyses give rise to the following conceptual framework (Figure 5m): as soon as tumor cells from intramucosal carcinomas breach the muscularis mucosae, they enter the submucosa and encounter a large reservoir of submucosal trophocytes. These trophocytes are triggered to transition into trophocyte-like CAFs, which in turn induce the regenerative oncofetal reprogramming in the invading tumor cells via TGF- β and prostaglandin signaling. Ultimately, our extensive amount of patient-related *in vivo* data provide the most robust evidence to date to identify the submucosal trophocytes as the cell-of-origin for the early CAFs in human CRC.

4. The authors link the acquisition of metastatic competency to fibroblast-released extracellular factors, like TGF- β ligands and prostaglandins. While the role of these pathways is very well-known in the field, the identification of specific receptor-ligand interactions would represent a significant advance. However, the work to characterize and validate some of the ligands highlighted in the manuscript (e.g., TGFB3) appears to be very preliminary. For example, it's unclear which specific TGF- β ligand the authors use in their study (e.g., is it TGFB1 or TGFB3?). Also, the functional assays to test for the activation of the invasive front-specific transcriptional program (e.g., EMP-1 knock-in-based flow-cytometry and a qPCR assay) are poorly documented and display substantial variability. If the authors added TGFB1 to their organoid lines is unsurprising to see a downregulation of an intestinal stem cell marker, like LGR5, and an increase in some injury repair markers, as EMP1, MMP7 and ANXA1 (it should be noted that EMP1 became highly expressed also during intestinal differentiation). Yet, such observations say relatively little on the ability to recapitulate the invasive front specific program and even less on the acquisition of metastatic competency.

We agree with the reviewer that the set of candidate marker genes assessed in our TGF- β and prostaglandin treatment experiments is more limited compared to the co-culture experiments where we performed RNA sequencing to evaluate the full HRC/regenerative gene signature. That said, we now performed analysis of >300,000 tumor epithelial cells from 5 scRNA-seq datasets of CRC to evaluate the robustness of the chosen markers (EMP1, MMP7 and ANXA1) and found their expression to be very representative for the full regenerative/HRC state (Extended Data Fig. 2a-f) within the tumor cell population.

We thank the reviewer for the insights regarding EMP1 expression during intestinal differentiation. In our manuscript, we rely on EMP1 expression as a specific marker for HRC/regenerative stem cells within tumor cell populations (i.e. within our CRC derived organoids, not the normal tissue-derived organoids).

We apologize for the confusion regarding the exact ligands that were used in our *in vitro* experiments, which we referred to in the figure legends and methods section of our manuscript. Here it is stated that a combination of TGF β 1 and 3 are used in the experiments. Both TGF β 1 and 3 are expressed by fibroblasts at the invasive front of early-stage CRC (see Fig. 4f) and for this reason we tested them in combination. We hope the reviewer recognizes that we do not intend to claim the mechanistic discovery that TGF β can induce regenerative/oncofetal states, rather we use the experimental data with T1 CRC organoids to provide functional support for the interactions that we observe between trophocyte-like CAFs and HRC/regenerative/ oncofetal-like cells at the transitioning from pre-cancer to cancer.

We initially used the fluorescent EMP1 knock-in reporter organoids to screen different conditions for their ability to induce EMP1 expression (Fig. 4h and Extended Data Fig. 9d). The most consistent response was observed with prostaglandins and the TGF β combinations, which we then validated across multiple organoid models (qPCR, Fig. 4j). In these assays, robust transcriptional induction of the three marker genes *EMP1*, *MMP7*, and *ANXA1* in organoids is very representative for activity of

the metastasis-associated oncofetal-like program in general (Extended Data Fig. 2, and as elaborated on above).

We would like to close by reiterating how technically challenging it is to truly model *in vivo* complexity *in vitro*, including its multicellularity, heterogeneity, spacing, endogenous levels of signaling gradients, and more. Hence, we value our comprehensive approach that not only includes functional assays with early cancer organoids, but in particular also their embedding within the various single-cell reference atlases, as the cornerstone in our strategy.

Rebuttal Figure 1 | Heiser *et al.* signature expression in T1 CRC spatial transcriptomics dataset
 Boxplots depicting expression of indicated immune exclusion (IEX) marker genes from Heiser *et al.* per histopathological region (left column) or metastatic status (right column) within spatial transcriptomics dataset of T1 CRCs. Points represent individual PanCK+ epithelial segments. ANOVA p-values are shown.

Moorman *et al.* signatures in T1 CRCs

Rebuttal Figure 2 | Moorman et al signature expression in T1 CRC spatial transcriptomics dataset
 Boxplots depicting module scores of indicated gene expression signatures from Moorman *et al.* per histopathological region or metastatic status within spatial transcriptomics dataset of T1 CRCs. Points represent individual PanCK+ epithelial segments. ANOVA p-values are shown.

Buissant des Amorie, Hageman, Brunner, van der Horst et al., Rebuttal letter 2

We thank the reviewers for their positive evaluation of our revised manuscript, including the strength of the new experimental additions and overall improvement of our manuscript. Please find below our response to the final requests of reviewer 1 and 2 regarding finetuning of the text to improve clarity of the message.

In addition, guided by suggestions made by reviewer 4, we have generated new experimental data to elevate our study regarding the plasticity-inducing stromal cues. We agree that there is an opportunity to further leverage our integrated approach of (spatial) single-cell atlases combined with experimental validations and acknowledge that our analysis should be comprehensive. Hence, we now included new experiments to tackle this point (see below).

However, we oppose the notion that molecular insights into *how* trophocyte-like CAFs induce oncofetal plasticity need to be novel. Our paper is the first to provide a conceptual framework on the time and place at which intensively studied oncofetal cells first appear during the evolutionary timeline of human cancer. This is the main message of our work, as also reiterated and emphasized by the other referees. Second, we *do* provide mechanistic understanding for the surprisingly early time and place at which oncofetal tumor cells arise by identifying submucosal trophocytes as the first cell-of-origin for CAFs and validating trophocyte-like CAFs as inducers of oncofetal states. Finally, our novel finding that oncofetal plasticity emerges much earlier than widely anticipated (as in conjunction with metastatic disease at late stages) will decouple the model that oncofetal cell states are the dominant drivers of metastases. They are necessary, but not sufficient, a finding that will open and redirect future investigations into the additional mechanisms that prevent tumor cells from disseminating.

Regarding new experimental additions:

- Inspired by cell-cell interaction analyses from our *in vivo* reference datasets (Extended Data Fig. 9c and the new Extended Data Fig. 10e), we now tested an extensive list of candidate stromal cues. These experiments reinforce our earlier observation that trophocyte-like CAFs induce oncofetal plasticity, unlike telocyte-derived ligands, collagen or various inflammatory cytokines that were also predicted to be upregulated in the cellular neighborhood of oncofetal cells. TGF β and prostaglandins, derived from trophocyte-like CAFs, represent the strongest inducers of oncofetal plasticity. We generated new figure panels regarding this data which highlight our integrated approach between (spatial) single-cell atlases and experimental validations (manuscript lines 246-254; Fig. 4g-h; Extended Data Fig. 9e-g).

- Moreover, to underscore sufficiency of these factors, we validated induction of the full EMP1-related oncofetal transcriptional program using RNA-seq on multiple patient-derived organoid lines (manuscript lines 254-256; Fig. 4j-k).

- Lastly, we find that induction of oncofetal plasticity is functionally decoupled from the acquisition of cell-intrinsic immune evasive properties like MHC1 downregulation (manuscript lines 258-260; flow cytometry: Extended Data Fig. 9i-j) or downregulation of immune-related programs (RNA-seq, Extended data Fig. 9k).

Regarding textual changes to our manuscript:

Important changes related to new content are highlighted in yellow. Additional textual edits include shortening of main text and legends to comply with editorial Nature formatting requests. Likewise, various figure panels from the previous version of our manuscript (Fig. 1d,f,o; Fig. 3l,n,p; Fig. 4f,g,l; and Fig. 5i,m) were moved from the main figures to extended data to comply with Nature figure size requirements. All changes can be found in the manuscript version that includes tracked-changes in full.

Referee #1 (Remarks to the Author):

Fibroblast patterning instigates metastasis-prone oncofetal states at colorectal cancer onset

Buisant des Amorie et al.,

Nature 2025-03-05750B

I thank the authors for carefully considering the feedback from my first review.

In my opinion the major strength of this paper is showing how trophocytes induce revival stem cells in early colon cancer. I do not think the word metastasis should be in the title of the paper, when there are no mets in the paper and the focus is on early NON-metastatic disease. One of the major findings of the paper is that revival states are ubiquitous in NON-metastatic tumors. Revival/regenerative stem cells are also commonly induced during non-cancerous tissue damage and infection (that's how they got their name) so obviously these cells don't cause metastasis on their own (otherwise we'd all have liver mets after a bowel infection). Access to the revival stem cell state is a normal response of intestinal epithelia to stress and inflammation. In my opinion the message of this paper is that revival stem cells are common at the invasive fronts of early disease, being necessary but not sufficient for metastasis.

We thank the reviewer for pointing out that the word 'metastasis' in our previous title could be confusing and may blur clarity of the manuscript's message. To prevent confusion, we now mention 'oncofetal plasticity' in the title rather than '*metastasis-prone* oncofetal states'. Moreover, we improved alignment of the title with our major finding as identified by all referees.

New title: "*Emergence of oncofetal plasticity is ubiquitous in early colorectal cancers*"

Moreover, in our updated abstract (and throughout the new manuscript) we carefully refined our wording and now explicitly mention that oncofetal states are ubiquitous in early non-metastatic cancers, being necessary but not sufficient for metastatic seeding.

Abstract lines 22-25: "*Here, we show that metastasis-associated oncofetal cell states already emerge at the earliest stages of CRC, concurrent with invasive front formation. However, while necessary for metastasis, we detect them ubiquitously among early non-metastatic cancers, highlighting additional bottlenecks like immune evasion.*"

We do, however, mention upon introducing oncofetal states that these are classically metastases-associated phenotypes. In fact, it is in this manuscript that we present the surprising finding that oncofetal cell states are ubiquitous among early non-metastatic cancers, thereby demonstrating that these states by themselves are not sufficient drivers of metastatic capacity. To ensure that we study and discuss the same oncofetal phenotype as is studied in late-stage cancer, which is experimentally shown to be the source of metastatic relapse in mice, and predicts risk for metastatic relapse in patients, we retain the use of metastases-associated transcriptional profiles (HRC) to computationally assign oncofetal phenotypes in our early-stage CRCs.

Importantly, to accommodate the suggestions of the reviewer with which we agree, we now use the term 'oncofetal cell states' throughout the manuscript when we directly refer to those cells. This is in particular after Figure 1, in which we decouple oncofetal cell states (HRC transcriptional phenotypes) and metastasis (i.e. oncofetal cell states are necessary but not sufficient).

The new Figure 5 is much, much stronger and substantially strengthens the temporal and spatial claims of the paper. This for me is what this manuscript is all about.

In my opinion, a more appropriate title of the paper would therefore be ‘Trophocytes induce revival stem (or oncofetal) cells at the invasive front of early colon cancer’.

We thank the reviewer for extracting the key finding of our work and ensure that this is reflected in the title. Inspired by the initial comments made by this reviewer, as well as the suggestions from referee 2 and 4, we changed the title, which now also complies with Nature’s maximum of 75 characters including spaces:

New title: “Emergence of oncofetal plasticity is ubiquitous in early colorectal cancers”

If space is limited due to the Nature Letter format, Figure 4 could also be moved to sup. The Figure 4 title says ‘Fibroblasts-derived TGFβ and prostaglandins induce invasive front tumor cell phenotypes’ but doesn’t show any invasive fronts. Again a more appropriate title would be ‘Fibroblasts-derived TGFβ and prostaglandins induce revival stem cells’ but that is already known (Qin et al., Cell, 2023, Roulis et al., Nature 2020) so I’m not sure it’s needs to be a main figure in a Nature paper if space is limited. Replacing Figure 4 with the new Figure 5 would help focus on the novel aspects of the paper.

We agree with the reasoning of the reviewer regarding Figure 4. However, our argument to keep it as a main figure, is to highlight our unique integrative approach where state-of-the-art, but descriptive, spatial single-cell atlases are accompanied with validations in experimental human *in vitro* models. Moreover, along the suggestions made by referee 4, we now elevated the content of this figure after comprehensively testing many stromal cues that were predicted with our datasets to be present near and of potential influence on oncofetal plasticity. Of interest, these extended experiments exclude major influence from ligand sources other than trophocyte-like CAFs, including immune-related inflammatory cytokines. We changed the title of the figure accordingly: “*Trophocyte-like CAFs induce oncofetal plasticity to EMP1+ tumor cell states*”.

The authors say they have removed the term “High Relapse Cells” (HRCs) from the manuscript but there are still mentioned in several places, including main figures (e.g. Fig 1k, Fig 3m, n, q, Fig 4b, Fig 5c, d, f, g, h). As the authors show in Figure 1i the HRC signature is the same as the revival/regenerative signature. Even the new term “metastasis-associated fetal-like cell states” still mentions metastasis, when there is no metastasis. Why not just use a revival stem cell signature throughout the paper to avoid implying metastasis in non-metastatic tumours? Revival stem cells are interesting for lots of reasons beyond metastasis (e.g. chemo-radioresistance, tumour initiation etc) so calling them “High Relapse Cells” is unnecessary at best, and misleading at worse because these majority of these patients did not relapse.

We kindly refer to our earlier answers regarding our textual refinements and the new use of oncofetal cell states.

In short, we now use the term ‘oncofetal cell states’ to refer to the cells with revival/regenerative fetal-like signatures. In particular after Figure 1, we refrain from using the association between metastasis and oncofetal phenotypes when not necessary, as it is primarily in Figure 1 where we decouple both phenomena. In few instances where oncofetal-like transcriptional profiles were computationally assigned based on the HRC signature, this is mentioned in the legends. We kept using HRC signatures to ensure that we discuss and study the same phenotypes as studied in late-stage cancer, which are experimentally shown to be the source of metastatic relapse in mice, and predict risk for metastatic relapse in patients.

We agree with the sentiment of the reviewer regarding nomenclature, including the referee’s statement made below that oncofetal cell states do not equate to metastasis per se. In fact, it is our

manuscript that delivers significant amounts of scientific evidence to substantiate that statement. We hope that our revised text improves clarity of that message.

There is some really great data in this paper and the find that trophocytes induce revival stem cells in early-CRC is novel and interesting to the field. If the authors can stop equating revival stem cells = metastasis (without showing metastasis) I think the fibroblast-patterning discovery message will be a lot clearer for readers.

We thank the reviewer for the time and energy to help us improve our manuscript, both experimentally as well as clarity in communicating our discovery.

All other points have been appropriately addressed and I congratulate the authors on a beautiful paper.

Referee #2 (Remarks to the Author):

The revised version of this study from Buissant des Amorie, Hageman, Brunner, van der Horst et al., presents a comprehensive and detailed series of new experimental data that directly address the points raised by this reviewer during the original round. As such, I have no hesitancy in recommending this for acceptance.

The revised version contains additional new fundamental understanding that very clearly demonstrates that the presence of the regenerative cellular state (identifiable by many overlapping, albeit uniquely named signatures) is not always a driver of metastasis in every tumour stage or setting. This was shown in the first submission via its ubiquitous expression in almost every pT1 which rarely metastasise, however in this revised version, the authors now unequivocally de-tangle and longitudinally track this paradoxical and dynamic relationship between "metastasis-prone" HRC/regenerative signalling and outcome. Additionally, the immune exclusion phenotype (repressed viral response signalling) are nicely decoupled from the timing of the regenerative signalling; a point that will no doubt open a therapeutic window for future investigation.

While these points will become very clear to those that read the paper in full, the title and abstract don't quite get across that these regenerative/onco-fetal/HRC.... states are almost ubiquitous from the moment of transformation and submucosal breach. The authors do say they emerge at these stages, but I guess the fact that these are uniformly reactive states to any/all submucosal interactions at this stage is really important, and not related to the metastasis potential of a tumour. The field has a habit of conflating a specific gene/biology with a fixed phenotype (like being a good or bad driver); importantly, what this study does so well is to add the context, features, evolutionary stage and timings that all need to be present in order for these regenerative/onco-fetal/HRC.... states to actually mean what many in the field assume they always mean (metastasis).

We thank the reviewer for the time and energy devoted to helping us improve our manuscript, as well as the latest round of suggestions to improve clarity in communicating the main findings.

In the revised version of our manuscript (in particular the new title and updated abstract), we now explicitly mention that oncofetal cells are ubiquitous in early-stage non-metastatic colorectal cancers, implying they are necessary but not sufficient for metastasis.

New title: "Emergence of oncofetal plasticity is ubiquitous in early colorectal cancers"

Abstract lines 22-25: "Here, we show that metastasis-associated oncofetal cell states already emerge at the earliest stages of CRC, concurrent with invasive front formation. However, while necessary for metastasis, we detect them ubiquitously among early non-metastatic cancers, highlighting additional bottlenecks like immune evasion."

Overall, I congratulate the authors on this excellent work. The authors have addressed my original concerns, added new data and revised analyses to reveal new understanding of the interplay between regenerative cell states, stromal populations, immune exclusion (via viral response signatures) and tumour evolution during developmental/progression phases.

Once again, we thank the reviewer for their time and energy devoted to helping us improve our manuscript.

Referee #4 (Remarks to the Author):

I have been asked to review this manuscript specifically with respect to the authors' responses to Reviewer #3, who has unfortunately withdrawn. I greatly enjoyed reading the revised manuscript, which makes a substantial contribution to the field by addressing important biological questions through contemporary techniques and the use of very novel early-stage CRC tissue datasets. The experimental work is scientifically rigorous, well executed, and clearly presented. The manuscript reads well and is significantly improved compared with the initial submission.

All three reviewers (including Reviewer #3) acknowledged the strength of the data and methodology but raised overlapping concerns around conceptual novelty, interpretation, and the lack of mechanistic insight.

I find the authors' responses to Reviewer #3 to be satisfactory in the sense that they have strengthened the manuscript and clarified points of confusion. However, despite these improvements, the work remains in my view an incremental advance. The conceptual novelty lies largely in the temporal and spatial refinement of known principles of plasticity and CAF signalling, rather than in the discovery of new mechanisms. The immune evasion "bottleneck" proposed by the authors is an appealing explanatory model, but at present is supported only by correlative transcriptomics rather than causal evidence.

We thank the reviewer for the compliments on our revised manuscript, as well as assessing our responses to reviewer 3.

As mentioned on the introductory page of this rebuttal letter, we disagree with the notion that the work is an incremental advance. We feel supported by comments from the other referees, that also reiterate and emphasize that our paper is the first to provide a conceptual framework on the time and place at which intensively studied oncofetal cells first appear during the evolutionary timeline of human cancer. Second, by identifying submucosal trophocytes as the first cell-of-origin for CAFs and validating trophocyte-like CAFs as inducers of oncofetal states, we provide mechanistic understanding for the surprisingly early time and place at which oncofetal tumor cell states first arise. Finally, our new finding that oncofetal plasticity emerges much earlier than anticipated (unexpectedly not in conjunction with metastatic disease at late stages) debunks the widely held assumption that oncofetal cell states are sufficient drivers of metastases. They are necessary, but not sufficient.

I also remain concerned about terminology. Even in the revised version, the use of "metastasis-prone oncofetal states" seems overstated. The cells described share transcriptomic similarity with metastatic states, but no functional data are presented to demonstrate that they are in fact metastasis-prone. Given the plasticity of revival/regenerative/fetal-like populations, a more neutral description would be more accurate.

We acknowledge the sentiment regarding the delicacy of terminology and the reviewer's suggestion to revise to a more neutral term. As also answered to similar suggestions by the other referees, we now revised our wording to improve clarity and avoid overstatements (predominantly referring to oncofetal cell states only, as opposed to 'metastasis-prone' or 'metastasis-associated' cell states).

In my opinion, to cross the bar for Nature, the following types of data (not necessarily all) would be required to demonstrate mechanistic breakthrough:

a) Necessity and sufficiency of stromal cues: Move beyond the well-established TGF β /PGE2 to identify and test additional ligands. Application of purified gradients that mimic the invasive front could test whether these signals alone recreate the in vivo patterning.

We appreciate the suggestions by referee 4 to elevate our study regarding the plasticity-inducing stromal cues.

We do emphasize that substantial mechanistic insight already lies in the identification of submucosal trophocytes as the first cell-of-origins for CAFs, which provides conceptual understanding on the time and place at which oncofetal plasticity first occurs during the evolution of human colorectal cancers. Moreover, using unprecedented datasets in our manuscript, we unequivocally demonstrate that oncofetal cell states do not automatically equate metastatic disease. These findings will have major impact on the field, as the widely held assumption is that oncofetal cells mean metastasis (by the words of referee 2).

We do, however, acknowledge that our analysis should be as comprehensive as possible, by harnessing the unique opportunity to further leverage our integrated approach between (spatial) single-cell atlases and experimental validations in human *in vitro* models of early-stage CRC.

Consequently, we now tested an extended list of candidate stromal cues, inspired by cell-cell interaction analyses from our *in vivo* reference datasets (Extended Data Fig. 9c and the new Extended Data Fig. 10e). These experiments reinforce our earlier observation that trophocyte-like CAFs induce oncofetal plasticity, unlike telocyte-derived ligands, collagen or various inflammatory cytokines that were also predicted to be upregulated in the cellular neighborhood of oncofetal cells. TGF β and prostaglandins, derived from trophocyte-like CAFs, represent the strongest inducers of oncofetal plasticity. We generated new figure panels regarding this data which highlight our integrated approach between (spatial) single-cell atlases and experimental validations (manuscript lines 246-254; Fig. 4g-h; Extended Data Fig. 9e-g).

Furthermore, to address sufficiency of these factors, we validated induction of the full EMP1-related oncofetal transcriptional program using RNA-seq on multiple patient-derived organoid lines (manuscript lines 254-256; Fig. 4j-k).

b) Causal link between fetal-like states and immune evasion: Tri-cultures of tumor organoids with matched trophocyte/CAF populations and autologous T cells to test whether stromal cues reduce immune killing via tumor state transitions. Additionally, determine whether CAF-conditioned invasive fronts recruit immunosuppressive myeloid states.

We appreciate the suggestion to investigate potential links between fetal-like states and immune evasion.

While complex tri-cultures with autologous T cells fall beyond the scope of the current work, we have been able to test direct co-acquisition of cell-intrinsic immune evasion, in conjunction with our screen to test for stromal ligands that induce EMP1+ oncofetal cell states. Specifically, we observed susceptibility to IFN γ -induced upregulation of MHC1 in early-stage CRC organoids (manuscript lines: 258-260; Extended Data Fig. 9i) and found minimal to no effects on MHC1 levels when cells transition to oncofetal cell states (Extended Data Fig. 9j). Similarly, we did not observe downregulation of immune-related programs per RNA analysis (Extended data Fig. 9k). Thus, in alignment with our observation that oncofetal plasticity comes early and generally decoupled of immune evasion, our functional data also suggest no causal link between the plastic transitioning to oncofetal cell states and the acquisition of cell-intrinsic immune evasive properties.

c) In vivo impact on dissemination: Orthotopic, barcoded lineage-competition assays with manipulation of CAF cues to test whether induction of fetal-like states alters metastatic seeding potential.

To date, there are unfortunately no experimental model systems, including genetic mouse models, that faithfully capture malignant transformation where benign adenomas transition into invasive carcinomas (see also the introductory paragraph of our manuscript). Moreover, xenotransplantation of patient-derived organoid models requires immunocompromised mice which eliminates an important factor influencing metastatic disease. For that reason, we decided to focus our current research efforts directly on the patient material, including the generation of unprecedented *in vivo*-relevant datasets that are accompanied by experimental organoid models.

Nevertheless, we acknowledge and share the ambition to explore experimental conditions to study the cause and consequences of malignant transformation *in vivo*. In fact, the requirement of fetal-like cell states for successful metastatic seeding has been demonstrated using elegant experiments in mouse models from the Battle lab (Cañellas-Socias et al., *Nature* 2022). Moreover, and in alignment with the reviewer's interest, we are exploring strategies to model spontaneous malignant transformation in mice, for future studies on the role of various cell-cell interactions during that transitioning, but we deem the development and study of such models well beyond the scope of our current manuscript as this cannot be executed within a reasonable timeframe.

Response to referee

Referee #4 (Remarks to the Author):

I thank the authors for their careful and considered response to the queries raised in my previous review. Terminology has been refined appropriately, and the additional experimental work included in Extended Data Figures 9 and 10 strengthens the manuscript to a level consistent with publication in Nature.

While individual reviewers may naturally differ in their emphasis on novelty, I recognise this as an important and timely piece of work that is analysed rigorously and presented with exceptional clarity. It makes a significant contribution to the field, and I congratulate the authors on a splendid study.

We thank the reviewer for the compliments on the technical and conceptual level of our work.